# FROM CHARTS TO CODE: A HIERARCHICAL BENCHMARK FOR MULTIMODAL MODELS

## ABSTRACT

We introduce Chart2Code, a new benchmark for evaluating the chart understanding and code generation capabilities of large multimodal models (LMMs). Chart2Code is explicitly designed from a user-driven perspective, capturing diverse real-world scenarios and progressively increasing task difficulty. It consists of three levels: **Level 1 (Chart Reproduction)** reproduces charts from a reference figure and user query; **Level 2 (Chart Editing)** involves complex modifications such as changing chart types or adding elements; and **Level 3 (Long-Table to Chart Generation)** requires models to transform long, information-dense tables into faithful charts following user instructions. To our knowledge, this is the first hierarchical benchmark that reflects practical chart2code usage while systematically scaling task complexity. In total, Chart2Code contains 1,947 tasks across 22 chart types, paired with multi-level evaluation metrics that assess both code correctness and the visual fidelity of rendered charts. We benchmark 25 state-of-the-art LMMs, including both proprietary and the latest open-source models such as GPT-5, Qwen2.5-VL, InternVL3/3.5, MiMo-VL, and Seed-1.6-VL. Experimental results demonstrate that even the strongest models struggle to generalize across levels and chart types, highlighting the significant challenges posed by Chart2Code. We anticipate this benchmark will drive advances in multimodal reasoning and foster the development of more robust and general-purpose LMMs.

## 1 INTRODUCTION

Charts are one of the most powerful tools for communicating complex ideas. From scientific publications to business reports, they distill large amounts of structured data into clear and persuasive visuals. With the rapid progress of large multimodal models (LMMs) (OpenAI, 2025; Anthropic, 2025), it becomes increasingly realistic to envision AI systems that not only interpret visual charts (Wang et al., 2024b) but also generate executable plotting code, a task we refer to as chart-to-code (chart2code). Such capabilities can significantly enhance productivity by automating visualization creation, enabling reproducibility.

Yet, the reality of how people use charts tells a different story. Users rarely stop at simple chart reproduction—they need to edit figures by changing chart types, merging datasets, or adding new elements; they often work with long tables that must be distilled into interpretable plots; and they expect precise control over layout and style to ensure clarity. On the other hand, current LMMs (OpenAI, 2025; Anthropic, 2025; Deitke et al., 2024) achieve impressively high scores on existing chart2code benchmarks Yang et al. (2025a); Zhao et al. (2025b), suggesting that the problem is close to being solved. However, when applied to these more common and demanding scenarios, the very same models often struggle, revealing substantial gaps in their practical ability (refer to Appendix B for examples). This discrepancy *creates a mismatch between reported benchmark performance and real-world utility, highlighting the need for a benchmark that more comprehensively reflects everyday chart2code challenges.*

Motivated by this observation, we introduce Chart2Code (Figure 1), a new benchmark designed to rigorously evaluate chart generation capabilities of LMMs under progressively challenging conditions. Chart2Code consists of three levels: **Level 1 (Chart Reproduction)** targets mimicking a reference figure and instruction; **Level 2 (Chart Editing)** requires complex and precise editing, such as changing chart types or adding new elements; **Level 3 (Long-Table to Chart Generation)** presents

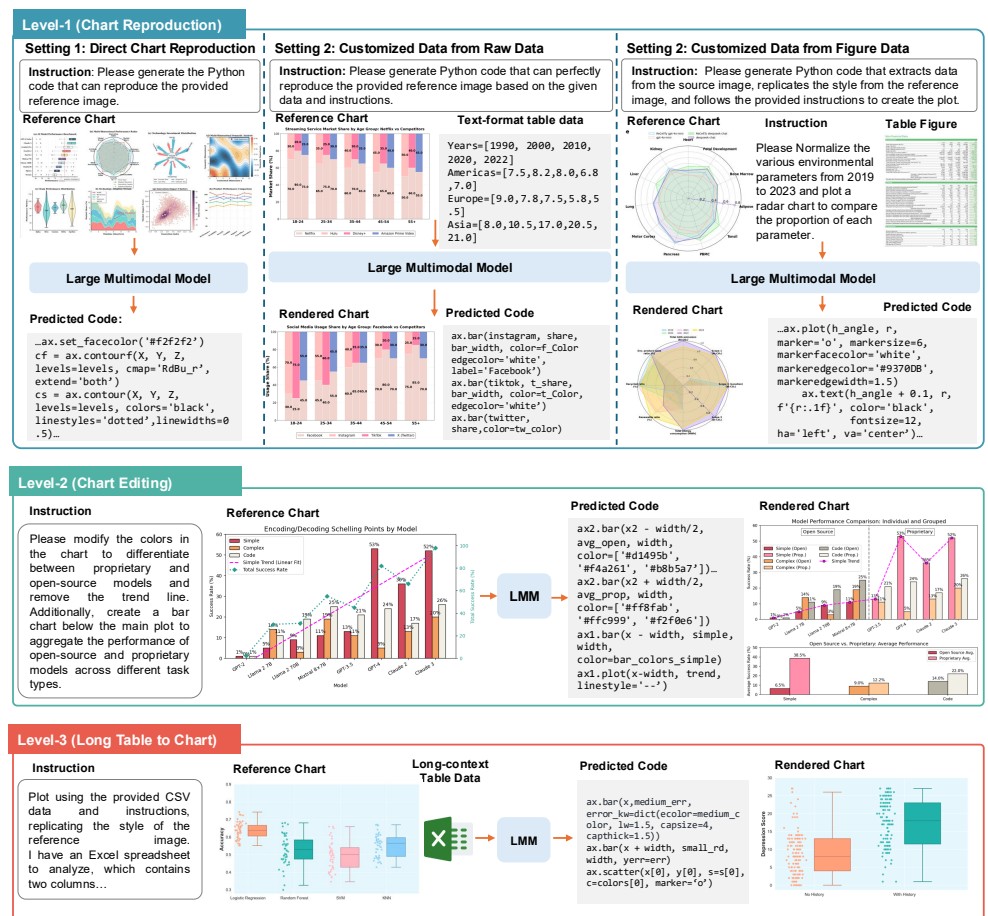

Figure 1: **Chart2Code covers three progressively challenging levels**: reproduction, editing, and long-table to chart generation. It provides a user-driven and diverse benchmark that better reflects real-world chart2code demands.

the most demanding setting, where models must convert long, unprocessed data tables into faithful charts from user instructions. This hierarchical design reflects real-world usage while progressively increasing difficulty, and its distinctions from prior benchmarks are highlighted in Table 1.

We comprehensively benchmark 25 state-of-the-art LMMs, including both proprietary and open-weight models, across the three levels of Chart2Code. Our results show that while LMMs demonstrate promising capabilities on simple reproduction tasks, their **performance deteriorates sharply on complex editing and long-context data-to-chart generation**. Together, these findings reveal the *unsolved challenges of chart2code generation and point to future directions for building more reliable visualization assistants*. In summary, our contributions are threefold:① We present Chart2Code, the first hierarchical benchmark targeting chart2code generation with progressively more challenging tasks. ② We propose multi-level evaluation protocols that jointly assess code executability and visual fidelity, offering a comprehensive lens on model performance. ③ We provide an extensive empirical study across 25 mainstream LMMs, yielding new insights into their strengths, weaknesses, and design trade-offs for chart generation.

## 2 RELATED WORK

**Large Multimodal Models.** Thanks to the success of proprietary LMMs such as GPT-5 (OpenAI, 2025), Gemini-2.5-Pro (Comanici et al., 2025), and Claude-Sonnet-4 (Anthropic, 2025), we see the dawn of building AI agents for addressing realistic applications. In the academic community, we see enormous excellent open-source models: MiMo-VL (Xiaomi & Team, 2025), QwenVL-series (Bai

Table 1: **Comparison of existing chart-to-code benchmarks.** Ref. Fig.: Reference Figure; Instr.: Instruction; Text Data: Text-format data; Fig. Data: Figure-format data; L1: Chart reproduction; L2: Chart editing; L3: Long-table-to-chart generation; NL: Natural language.

| Benchmark | Input Type | | | | Task Cat. | | | Output | Metric | |
|---|---|---|---|---|---|---|---|---|---|---|
| | Ref. Fig. | Instr. | Text Data | Fig. Data | L1 | L2 | L3 | | Rule-based | GPT-score |
| CharXiv (Wang et al., 2024b) | ✗ | ✓ | ✗ | ✗ | ✗ | ✗ | ✗ | NL | ✓ | ✗ |
| Plot2Code (Wu et al., 2025) | ✓ | ✓ | ✗ | ✗ | ✓ | ✗ | ✗ | Code | ✗ | ✓ |
| AcademiaChart (Zhang et al., 2024) | ✓ | ✓ | ✗ | ✗ | ✓ | ✗ | ✗ | Code | ✓ | ✓ |
| Chartmimic (Yang et al., 2025a) | ✓ | ✓ | ✗ | ✗ | ✓ | ✗ | ✗ | Code | ✓ | ✓ |
| ChartEdit (Zhao et al., 2025b) | ✓ | ✓ | ✗ | ✗ | ✗ | ✓ | ✗ | Code | ✗ | ✓ |
| Chart2Code (Ours) | ✓ | ✓ | ✓ | ✓ | ✓ | ✓ | ✓ | Code | ✓ | ✓ |

et al., 2025; Wang et al., 2024a), and InternVL-series (Wang et al., 2025; Zhu et al., 2025), MolMo (Deitke et al., 2024), Kimi-VL (Team et al., 2025) LLaVA-series (Li et al., 2024a; Liu et al., 2024; Li et al., 2024b), Deepseek-VL (Lu et al., 2024), and GLM-4V (GLM et al., 2024).

**Agentic Benchmarks.** The rapid progress of foundation LLMs and LMMs has motivated the creation of diverse agentic benchmarks, spanning GUI automation (Xie et al., 2024; Zhao et al., 2025a; Lin et al., 2024; Koh et al., 2024), agentic coding (Jimenez et al., 2024; Yang et al., 2025b), tool use (Yao et al., 2025), AI research assistance (Nathani et al., 2025), and chart reasoning (Wang et al., 2024b). We focus on chart2code, a practical task central to everyday workflows for researchers and professionals. Despite progress, even the best proprietary LMMs still fail to generate faithful charts from long, raw tables, underscoring the need for future modeling advances.

**Chart Understanding to Code Generation.** Chart understanding has evolved through a series of benchmarks that progressively expand task complexity. ChartQA (Masry et al., 2022) first established large-scale visual question answering over charts, combining queries with logical and visual reasoning. ChartXiv (Wang et al., 2024b) advanced this line by introducing scientific charts with expert-designed questions, further exposing the gap between multimodal models and human performance. Moving beyond QA, Chart2Code benchmarks address faithful chart generation. ChartMimic (Yang et al., 2025a) formalized this by requiring code synthesis from chart images and instructions, while ChartEdit (Zhao et al., 2025b) emphasized interactive modification, where models must edit chart-rendering code following natural-language instructions. Extending chart generation more generally, StarVector (Rodriguez et al., 2025) proposed a vision-language approach to directly produce scalable vector graphics from visual or textual inputs. Although GPT-4o achieves high scores on ChartMimic (83.2) and ChartEdit (93.6), it still struggles with realistic chart2code tasks, motivating a new, more challenging benchmark for reliable evaluation.

## 3 CHART2CODE: FROM VISUAL CHARTS TO CODE

### 3.1 TASK DEFINITION OF CHART2CODE

Chart2Code can be represented as: $C = f(R, I, D)$ where, $R$ is the reference chart (e.g., screenshot), $I$ is the instruction and $C$ is the Python code generated by LMM ($f$). $D$ represents optional input data types, Chart2Code supports three kinds of data formats: textual data, image data (e.g., screenshot), and Excel files. To ensure rigor and comprehensiveness, we designed three tasks of increasing difficulty.

**Level 1 (Chart Reproduction):** This task consists of two subsettings. The first setting requires the LMM to directly generate the executable code that can reproduce the reference chart ($R$). This task primarily explores the model's visual understanding capabilities. The second setting requires the LMM to extract the required table data from the data file $D$ and generate Python code based on the style and format of the given reference chart ($R$). It is closely aligned with real-world chart creation needs and not included in previous studies (Yang et al., 2025a; Wu et al., 2025; Zhang et al., 2024).

**Level 2 (Chart Editing):** At this level, the LMM edits the reference chart ($R$) as instructed, with operations like style changes, type swaps, data edits, or multi-subplot generation. The LMM is expected to generate code that meets the editing requirements and adheres to the style and format of chart.

**Level 3 (Long-Table to Chart Generation):** The final level asks the LMM to accurately gather the target data points from the extremely long data and unprocessed sheet and then produce the executable code, referencing the style and format of the given reference chart ($R$). It is the hardest task, which targets the most realistic scenario in data visualization or business presentations, assuming the user is not a data visualization expert.

## 3.2 DATA CURATION AND ANNOTATION

### 3.2.1 DATA CURATION

**Chart Data:** Our chart figure sources primarily consist of three aspects. First, we collected approximately 5,000 paper charts from Arxiv, spanning from January 2024 to July 2025, covering various fields such as CSEE, Physics, Statistics, and Economics, to ensure diversity and modernity in the chart types. Second, we gathered 1,000 example charts from function libraries such as Matplotlib, Seaborn, WordCloudX, Scipy, as well as Matlab plotting example tutorials. Finally, we filtered 300 difficult charts from the ChartMimic (Yang et al., 2025a) dataset.

**Raw Data:** Our benchmark collects raw data from sources such as Kaggle, Annual Reports, and publicly available data from various company websites. The raw data includes Excel spreadsheets, figures, text, and other formats, covering multiple domains such as corporate financial reports, flight route data, weather data, GDP data, and car sales figures. Additionally, we have intentionally selected data of varying lengths to test the LLM's ability to analyze and process long text data.

### 3.2.2 DATA FILTERING

**Chart Data:** We propose a "gathering-distribution" data selection process. First, we gather data from various sources into a chart pool, which is then roughly filtered by 10 undergraduate computer science students based on chart type and information complexity. Based on this initial selection, we reduce the data to 3,000 charts to ensure that the resulting data contains a diverse range of visual elements and chart types. Next, the gathered data is distributed by category to 5 experts with many years of experience in Python plotting for independent evaluation. The evaluation criteria are refined into three dimensions: data complexity, visual complexity, and programming complexity. Each dimension is independently assessed to select more valuable charts as part of the benchmark data. Finally, the charts from various categories are aggregated to form the 719 reference figures in the benchmark.

**Raw Data:** Since the raw data we collected contains various data formats, we first use automated scripts to filter out the raw data that exhibits rich numerical performance and is suitable for plotting. After that, we conduct manual checks to preserve the diversity of the raw data as much as possible. The final selection includes 39 Excel files, 80 raw data figures, and 36 raw data text files.

### 3.2.3 DATA ANNOTATION

During the data annotation process for the three-level tasks, we employed an interactive data annotation method based on Python scripts and agents, which we refer to as the human-AI interactive annotation process. Specifically, in the level 1 data annotation process, annotators, with the assistance of the LMM, recreate the selected data by writing Python code. The data generated here directly serves as the first setting of the Level 1 task. Subsequently, based on the 719 scripts, annotators select and modify suitable chart types using the data from the 80 raw table figures and 36 raw table text files, resulting in 108+36 customized entries for the second setting of the task.

In the Level 2 annotation process, annotators first categorize and summarize chart editing operations commonly encountered in real-world scenarios. They then modify the code with the help of prompt engineering and Python code injection, leveraging the programming capabilities of LLM. While the LLM may lack proficiency in the chart2code task, its programming ability is exceptional. Through this process, we obtained over 4,700 edited and modified scripts, which were further filtered through the data selection process, ultimately yielding 925 high-quality Level 2 data entries.

For Level 3 data annotation, annotators first analyzed the content of the 39 diverse data tables, formulated statistical data requirements, and extracted and processed the data from the tables. This process resulted in 150 Level 3 data entries.

Figure 2: **Collected charts distribution.**

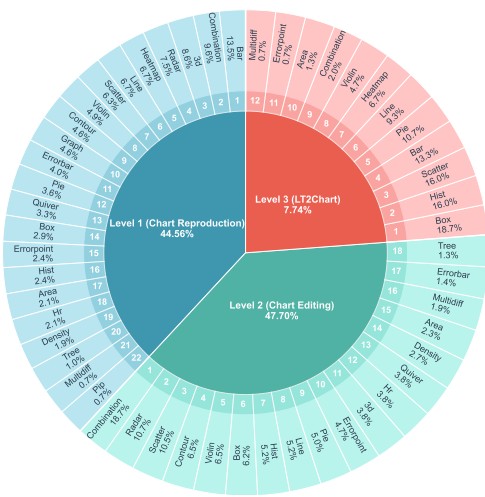

Table 2: **Deatiled data statistic.**

| Statistic | Number |
| --- | --- |
| **GT Charts** | |
| Total charts | 1,911 |
| - Level 1 / 2 / 3 charts | 836 / 925 / 150 |
| Unique charts | 804 |
| - Unique Level 1 / 2 / 3 charts | 719 / 0 / 85 |
| **Instructions** | |
| Total instructions | 1,947 |
| - Level 1 / 2 / 3 instructions | 872 / 925 / 150 |
| Unique instructions | 1,220 |
| - Unique instructions - Level 1 / 2 / 3 | 145 / 925 / 150 |
| Maximum instruction length - Level 1/ 2 / 3 | 224 / 544 / 390 |
| Average instruction length - Level 1 / 2 / 3 | 137.8 / 307.6 / 178.9 |
| **GT Code (Lengths/Lines)** | |
| Maximum code length - Level 1 / 2 / 3 | 96,563 / 7,855 / 790,130 |
| Average code length - Level 1 / 2 / 3 | 2,621.6 / 2,880.6 / 29,899.8 |
| Maximum code lines - Level 1 / 2 / 3 | 842 / 219 / 388 |
| Average code lines - Level 1 / 2 / 3 | 69.9 / 82.9 / 51.3 |
| **Extremely Long-Table Data** | |
| Total Excel files | 37 |
| Average lines per file | 606.7 |
| Maximum lines | 3,023 |
| Average data entries | 8,329.3 |
| Maximum data entries | 51,391 |

### 3.3 DATA STATISTICS AND ANALYSIS

Chart2Code comprises 1,947 tasks across three levels–872/925/150 for L1/L2/L3–spanning 22/18/12 chart families (e.g., radar, heatmap, scatter, box, tree, error-bar, pie, multidiff; see Fig. 2). To maximize diversity, Level 1 emphasizes unique charts (719 unique). Level 2 reuses Level 1 charts with at least one edit instruction per chart, resulting in 925 unique, non-duplicated edits. Level 3 (LT2Chart) includes 85 charts and 150 instructions derived from web-sourced long tables, making annotation and ground-truth code especially challenging. As summarized in Tab. 2, the instruction scale and substantial code lengths highlight the breadth and difficulty of Chart2Code.

### 3.4 EVALUATION

To comprehensively evaluate the performance of various models on the Chart2Code benchmark, we first establish the code **executability rate** as the primary evaluation metric. This directly measures the model's ability to generate functional visualization code, and its calculation is detailed in equation 1. Secondly, we introduce a multi-level, multi-dimensional evaluation method to assess model performance at both the code-level and the chart-level.

At the code-level, we propose a 'base evaluation' method that calculates the similarity of visual outcomes by parsing and matching `matplotlib.Figure` objects across eight dimensions. Our 'base evaluation' method offers faster assessment, more comprehensive dimensions, and superior evaluation performance (see Appendix E.2 for details). Similarly, to provide a broader code assessment, we employ GPT-5-mini (OpenAI, 2025) to score the code without execution, assessing its prospective visual output to derive a comprehensive **LLM-score** (see Appendix E.3 for details).

At the chart-level, we similarly use GPT-5-mini to assess the predicted charts, yielding an **LMM-score**. Although LLMs like GPT-5 may not excel at the Chart2Code generation task itself, they possess a keen ability to judge the similarity between both code and charts. The direct evaluation of charts is most aligned with human intuition, making it more suitable as the final evaluation score.

## 4 EXPERIMENTS

### 4.1 EXPERIMENTS SETUP

**Models.** We conducted tests on 25 widely-used open-source models and proprietary models to evaluate their performance on our benchmark. For the open-source models, we selected 12 representative vision-language models, with total parameters ranging from 7B to 72B, including: Qwen2-VL (7B, 72B), Qwen2.5-VL (7B, 72B), Deepseek-VL (7B), Kimi-VL (7B), MiMo-VL-SFT (7B), MiMo-VL-RL (7B), InternVL-2.5 (8B, 38B), InternVL-3 (8B, 38B), InternVL-3.5 (8B, 38B), GLM-4V (9B),

Table 3: **Evaluation results on Chart Reproduction (Level 1) with various LMMs.** Each task includes a reference chart as input. DR: input without the table data. CRD: input with customized text-format table data. CFD: input with customized figure-format table data. Exec. Rate: execution rate; We use GPT-5-mini as the base model for both LLM-score and LMM-score;

| Model | Direct Mimic(DR) | | | Customize Raw Data(CRD) | | | Customize Figure Data(CFD) | | |
|---|---|---|---|---|---|---|---|---|---|
| | Exec.Rate | LLM-Score | LMM-Score | Exec.Rate | LLM-Score | LMM-Score | Exec.Rate | LLM-Score | LMM-Score |
| *Proprietary* | | | | | | | | | |
| Gemini-2.5-Pro | 90.4 | 0.6286 | 0.3807 | **100** | **0.6763** | 0.2661 | 87.04 | **0.6145** | 0.2214 |
| Claude-Sonnet-4 | **96.38** | 0.5629 | 0.2553 | 97.2 | 0.4878 | 0.236 | 88.89 | 0.5538 | 0.2273 |
| GPT-5 | 87.48 | 0.6334 | 0.3575 | 94.4 | 0.6070 | 0.2238 | 85.19 | 0.6082 | **0.2382** |
| Seed-1.5-VL | 85.81 | 0.5536 | 0.2341 | 97.2 | 0.6325 | **0.2662** | 65.74 | 0.5756 | 0.1962 |
| Seed-1.6-VL | 84.70 | 0.5237 | **0.8117** | 94.4 | 0.6525 | 0.2503 | 83.96 | 0.5978 | 0.2075 |
| *Open-Source LMMs (non-thinking)* | | | | | | | | | |
| LLaVA-OV-Qwen2-7B-SI | 32.82 | 0.1820 | 0.0154 | 11.11 | 0.4225 | 0.1550 | 0 | - | - |
| LLaVA-OV-Qwen2-7B-OV | 11.13 | 0.2651 | 0.0376 | 5.56 | 0.4213 | 0.0825 | 0 | - | - |
| DeepSeek-VL-7B | 48.68 | 0.2854 | 0.0431 | 61.11 | 0.5374 | 0.1114 | 10.19 | 0.2539 | 0.0145 |
| kimi-VL-A3B | 68.85 | 0.4409 | 0.1374 | 72.22 | 0.5887 | 0.2081 | 61.11 | 0.4641 | 0.1379 |
| Qwen2-VL-7B | 64.39 | 0.3364 | 0.0664 | 75.00 | 0.5950 | 0.1367 | 30.56 | 0.4235 | 0.0519 |
| Qwen2-VL-72B | 75.66 | 0.4368 | 0.1207 | 80.56 | 0.6082 | 0.1628 | 51.85 | 0.5518 | 0.1373 |
| InternVL-2.5-8B | 66.89 | 0.3348 | 0.0723 | 80.56 | 0.5712 | 0.1183 | 37.74 | 0.5715 | 0.0568 |
| InternVL-2.5-38B | 86.23 | 0.4577 | 0.1463 | 0 | - | - | 0 | - | - |
| InternVL-3-8B | 66.34 | 0.4371 | 0.1389 | 86.11 | 0.6169 | 0.1732 | 57.41 | 0.4450 | 0.1028 |
| GLM-4V-9B | 72.18 | 0.2881 | 0.0459 | 66.67 | 0.5628 | 0.1183 | 44.74 | 0.2904 | 0.0130 |
| Intern-VL-3.5-8B | 66.34 | 0.4371 | 0.1389 | 86.11 | 0.6169 | 0.1732 | 57.41 | 0.4450 | 0.1028 |
| MiMo-VL-7B-RL | 37.83 | 0.5439 | 0.2316 | 69.44 | 0.6068 | 0.2421 | 41.67 | 0.4962 | 0.1407 |
| MiMo-VL-7B-SFT | 44.65 | 0.4959 | 0.1983 | 69.44 | 0.6237 | 0.1852 | 46.30 | 0.5155 | 0.1732 |
| Qwen2.5-VL-7B | 65.64 | 0.4197 | 0.0994 | 75.00 | 0.5952 | 0.1515 | 44.44 | 0.5952 | 0.091 |
| Qwen2.5-VL-72B | 65.36 | 0.5118 | 0.1893 | 100 | 0.6273 | 0.1989 | 37.96 | 0.5532 | 0.1688 |
| *Open-Source LMMs (thinking)* | | | | | | | | | |
| MiMo-VL-7B-RL | 55.77 | 0.5261 | 0.2294 | 69.44 | 0.6053 | 0.2582 | 33.33 | 0.5807 | 0.2172 |
| MiMo-VL-7B-SFT | 50.35 | **0.6555** | 0.2130 | 86.11 | 0.6644 | 0.2248 | 38.89 | 0.5578 | 0.1455 |

LLAVA-onevision-si (7B), LLAVA-onevision-ov (7B), Molmo (7B). For proprietary models, we selected the five most popular multimodal large models, including: Gemini-2.5-pro, Claude-sonnet-4, GPT-5, Seed-1.5-VL, and Seed-1.6-VL.

**Configuration.** All experiments were conducted on NVIDIA V100 GPUs. Qwen2-VL-7B and Qwen2.5-VL-7B models were executed on a single GPU. MiMo-VL-SFT, MiMo-VL-RL, and LLaVA-OneVision (LLaVA-OV) required two GPUs, with inference parallelized across devices due to memory constraints. Similarly, the InternVL series (2.5-VL-8B, 3-VL-8B, 3.5-VL-8B), Kimi-VL, DeepSeek-VL, and GLM-4V models were evaluated using two GPUs with model parallelism. We set the maximum output length to 8,192 tokens for Level 1 and 2, and 32,768 tokens for Level 3. Empirically, non-thinking models required only 4,096 tokens, with negligible truncation except for the largest InternVL-3.5-38B model. The decoding temperature was fixed at 0.1 across all models. To preserve visual fidelity, we fed images at their native resolution and used the maximum input pixel setting supported by each model to ensure complete processing of chart details.

## 4.2 MAIN EXPERIMENTAL RESULTS

### 4.2.1 LEVEL-WISE COMPARISON OF MODELS

**Level 1.** As shown in Tab. 3, proprietary models lead across Direct Mimic (DM), Customize Raw Data (CRD), and Customize Figure Data (CFD), achieving high executability but only moderate visual fidelity—for example, Gemini-2.5-Pro reaches 90.4/100/87.04% ER on DM/CRD/CFD while LMM-Scores stay around 0.22–0.38. CRD is "easy to run" (e.g., Gemini and Qwen2.5-VL-72B at ≈100% ER) yet still low-fidelity (≈0.15–0.27), confirming execution ≠ fidelity. CFD is the hardest: top proprietary models keep ≥85% ER but LMM-Scores remain ≈0.22–0.24, and many open-source models drop sharply (some 0 ER). Larger open-source backbones (Qwen2/2.5-VL-72B, InternVL-3-8B/38B) close part of the execution gap but not the fidelity gap. A notable outlier is Seed-1.6-VL with DM LMM-Score ≈0.812, suggesting evaluator/model calibration effects.

**Level 2.** The results are presented in Tab. 4. Proprietary models sustain ∼90% ER (Gemini 90.49, Claude 90.92, GPT-5 90.59) and excel on code-level subskills—especially Layout/Type ≈0.95–0.96—yet figure-level remains modest (∼0.18–0.22), evidencing a persistent gap between syntactic compliance and rendered-image fidelity. Strong open-source systems improve executability (e.g., Qwen2.5-VL-72B 71.89%) with solid code-level scores (Layout ≈0.94, Type ≈0.92), but figure-level still lags (0.12–0.14). Smaller backbones struggle (e.g., LLaVA-OV-Qwen2-7B variants ≤2.71%

Table 4: **Evaluation results on Chart Editing (Level 2) with various LMMs.**

| Model | Exec. Rate | Color | Grid | Layout | Legend | Visual | Data | Text | Type | LLM-Score | Chart-Level LMM-Score |
|---|---|---|---|---|---|---|---|---|---|---|---|
| | | | | | | Code-Level | | | | | |
| *Proprietary* | | | | | | | | | | | |
| Gemini-2.5-Pro | 90.49 | **0.6284** | **0.8958** | **0.9606** | 0.5269 | **0.4988** | 0.7564 | 0.6195 | **0.9638** | **0.5725** | 0.2134 |
| Claude-Sonnet-4 | **90.92** | 0.5871 | 0.8330 | 0.9591 | 0.4878 | 0.4640 | 0.6782 | 0.5724 | 0.9575 | 0.5318 | 0.1844 |
| GPT-5 | 90.59 | 0.5898 | 0.8548 | 0.9509 | 0.4939 | 0.4643 | 0.7040 | 0.5962 | 0.9602 | 0.5658 | 0.2201 |
| Seed-1.5-VL | 63.46 | 0.5213 | 0.8418 | 0.9530 | 0.4599 | 0.4400 | 0.7013 | 0.7175 | 0.9433 | 0.5148 | 0.1547 |
| Seed-1.6-VL | 72.22 | 0.5359 | 0.8117 | 0.9485 | 0.4926 | 0.4275 | 0.6888 | **0.7324** | 0.9441 | 0.5179 | 0.1634 |
| *Open-Source LMMs (non-thinking)* | | | | | | | | | | | |
| LLaVA-OV-Qwen2-7B-SI | 1.30 | 0.3507 | 0.6964 | 0.7833 | 0.4074 | 0.3002 | 0.5249 | 0.4871 | 0.7889 | 0.3157 | 0.0875 |
| LLaVA-OV-Qwen2-7B-OV | 2.71 | 0.3216 | 0.5933 | 0.7138 | 0.4667 | 0.2111 | 0.5041 | 0.5080 | 0.3607 | 0.2975 | 0.0284 |
| DeepSeek-VL-7B | 22.51 | 0.2625 | 0.6403 | 0.7273 | 0.2541 | 0.1797 | 0.4121 | 0.4572 | 0.8048 | 0.2600 | 0.0322 |
| kimi-VL-A3B | 49.73 | 0.4055 | 0.7376 | 0.9069 | 0.3633 | 0.3176 | 0.5876 | 0.5915 | 0.9131 | 0.3776 | 0.0838 |
| Qwen2-VL-7B | 24.86 | 0.2859 | 0.6116 | 0.7736 | 0.2900 | 0.2221 | 0.4602 | 0.4881 | 0.8124 | 0.3215 | 0.0519 |
| Qwen2-VL-72B | 57.73 | 0.4161 | 0.7972 | 0.9044 | 0.3581 | 0.3276 | 0.6149 | 0.5748 | 0.9129 | 0.3949 | 0.0898 |
| InternVL-2.5-8B | 21.08 | 0.3343 | 0.7165 | 0.8388 | 0.3213 | 0.2741 | 0.5378 | 0.5488 | 0.8423 | 0.3391 | 0.0611 |
| InternVL-2.5-38B | 69.47 | 0.2625 | 0.6403 | 0.7273 | 0.2541 | 0.1797 | 0.4121 | 0.4572 | 0.8048 | 0.2600 | 0.0322 |
| InternVL-3-8B | 4.65 | 0.3609 | 0.6094 | 0.9408 | 0.3393 | 0.3454 | 0.5581 | 0.5313 | 0.8533 | 0.3504 | 0.073 |
| InternVL-3-38B | 61.51 | 0.4818 | 0.7954 | 0.9406 | 0.4281 | 0.3841 | 0.6476 | 0.6544 | 0.9216 | 0.4543 | 0.1205 |
| GLM-4V-9B | 10.49 | 0.2085 | 0.6869 | 0.7771 | 0.2470 | 0.2016 | 0.4616 | 0.4904 | 0.7598 | 0.2975 | 0.0533 |
| Intern-VL-3.5-8B | 25.62 | 0.4218 | 0.7590 | 0.8975 | 0.3849 | 0.3670 | 0.6290 | 0.6530 | 0.9181 | 0.4072 | 0.1062 |
| MiMo-VL-7B-RL | 16.54 | 0.4454 | 0.8706 | 0.9260 | 0.4376 | 0.4014 | 0.6421 | 0.6530 | 0.9172 | 0.4461 | **0.4713** |
| MiMo-VL-7B-SFT | 22.27 | 0.4435 | 0.7581 | 0.8888 | 0.3982 | 0.3891 | 0.6335 | 0.6558 | 0.9371 | 0.4510 | 0.1203 |
| Qwen2.5-VL-7B | 33.84 | 0.286 | 0.612 | 0.774 | 0.290 | 0.222 | 0.460 | 0.488 | 0.81 | 0.3651 | 0.0759 |
| Qwen2.5-VL-72B | 71.89 | 0.5109 | 0.8470 | 0.9492 | 0.4606 | 0.4127 | 0.6653 | 0.6808 | 0.9362 | 0.4782 | 0.1437 |
| *Open-Source LMMs (thinking)* | | | | | | | | | | | |
| MiMo-VL-7B-RL | 28.32 | 0.5157 | 0.7643 | 0.9452 | 0.4226 | 0.4246 | 0.7014 | 0.6854 | 0.9489 | 0.4844 | 0.1510 |
| MiMo-VL-7B-SFT | 23.57 | 0.4746 | 0.7545 | 0.9269 | 0.3838 | 0.3741 | 0.6769 | 0.6574 | 0.9351 | 0.4583 | 0.1367 |

Table 5: **Evaluation results on Long-Table to Chart task (Level 3) with various LMMs.**

| Model | Exec. Rate | Color | Grid | Layout | Legend | Visual | Data | Text | Type | LLM-Score | Figure-Level LMM-Score |
|---|---|---|---|---|---|---|---|---|---|---|---|
| | | | | | | Code-Level | | | | | |
| *Proprietary* | | | | | | | | | | | |
| Gemini-2.5-Pro | 29.33 | **0.7276** | **0.9733** | 1.0000 | 0.7727 | **0.6701** | 0.7880 | **0.8291** | 0.9470 | 0.3516 | 0.0361 |
| Claude-Sonnet-4 | **38.00** | 0.5676 | 0.7963 | 1.0000 | 0.8148 | 0.3731 | 0.5881 | 0.7175 | 0.9062 | **0.5125** | 0.007 |
| GPT-5 | **38.00** | 0.5676 | 0.7963 | 1.0000 | 0.8148 | 0.3731 | 0.5881 | 0.7175 | 0.9062 | **0.5125** | 0.0362 |
| Seed-1.5-VL | 18.67 | 0.7252 | 0.8929 | 1.0000 | **0.8869** | 0.5502 | 0.7182 | 0.7804 | **0.9690** | 0.0000 | **0.0611** |
| Seed-1.6-VL | 40.00 | 0.7030 | 0.8833 | 1.0000 | 0.7972 | 0.5396 | **0.7956** | 0.8128 | 0.9244 | 0.0000 | 0.0547 |

ER). "Thinking" helps procedure more than pixels: MiMo-VL-7B-RL ER improves 16.54→28.32, and MiMo-VL-7B-SFT figure-level nudges 0.1203→0.1367, but absolute fidelity remains low; the unusually high 0.4713 figure-level for MiMo-VL-7B-RL (non-thinking) merits.

**Level 3.** Tab. 5 presented the results. Coverage is limited because the benchmark is very hard: only a couple of open-source models could even complete inference, and on the proprietary side, five models were run, but overall ER is still <50%, primarily due to long-context inputs exceeding the maximum input limits. Among those that ran, ER drops to 29–40% (e.g., Gemini 29.33%), while code-level stays strong (Layout = 1.0; high Grid/Type), indicating structurally plausible code under long context. However, figure-level fidelity collapses (Gemini 0.0361, Claude 0.007, GPT-5 0.0362; Seed-1.5/1.6-VL 0.061/0.055), showing that turning lengthy raw tables into pixel-accurate charts is the main bottleneck; the Seed rows also show LLM-Score = 0 with non-zero LMM-Score, hinting at evaluator/model coupling or edge-case artifacts that warrant robustness checks.

### 4.2.2 ANALYSIS

**Execution vs. Complexity:** From level 2 to Level 3, ER for proprietary systems drops from 90% in Tab. 4 to 29–40% on Level 3 (Gemini 29.33, Claude 38.00, GPT-5 38.00 in Tab. 5). This mirrors the jump in reasoning load (long-context/table parsing, multi-constraint edits), showing that being able to run code at level 2 does not translate to robust end-to-end success at Level 3. We concluded **execution success declines steeply with task complexity, even for top proprietary models**.

**Code vs. Visual Fidelity:** On level 2 (Tab. 4), proprietary models score very high on Layout/Type (e.g., Gemini 0.9606/0.9638, Claude 0.9591/0.9575, GPT-5 0.9509/0.9602), yet figure-level GPT-Score is only 0.18–0.22 (Gemini 0.2134, Claude 0.1844, GPT-5 0.2201). On Level 3 (Tab. 5),

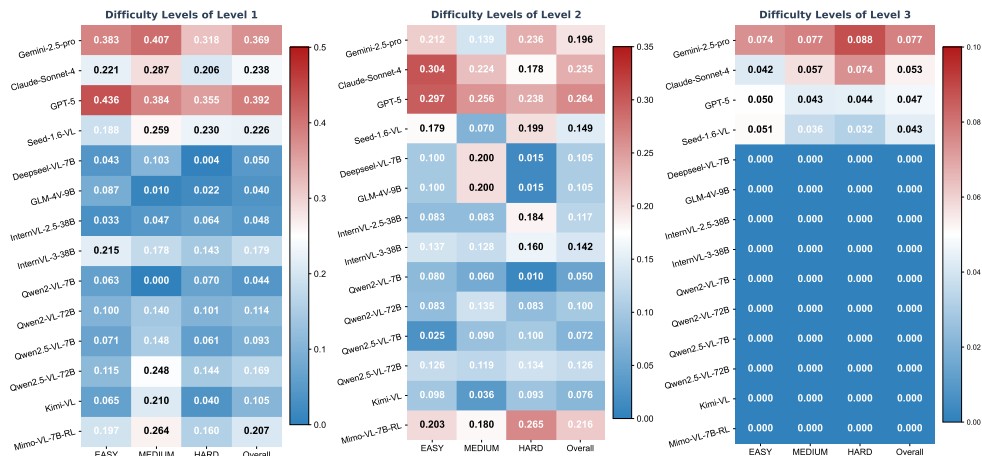

Figure 3: Correlation of the model performance (i.e, LMM-score) on different manually annotated difficulty levels (i.e., Easy, Medium, Hard) on Level 1, 2, 3, respectively.

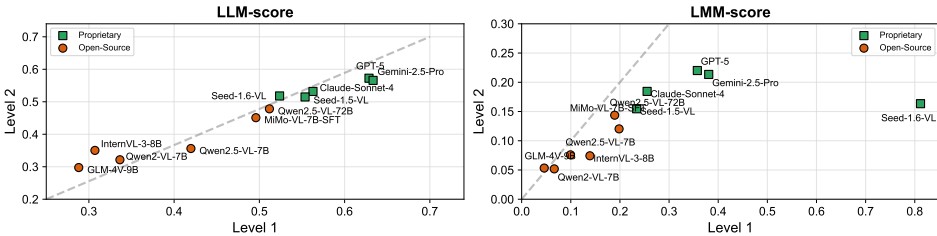

Figure 4: Left: Both proprietary and open-source models generalize well on Level 1 and Level 2 tasks when calculating the LLM-score for predicted code assessment. Right: Proprietary models tend to obtain higher LMM-scores on the Level 1 task rather than the Level 2, while open-source models perform poorly on both tasks (scores are lower than 0.5).

the gap widens: code-level remains strong (e.g., Layout = 1.0000 across models), but LMM-Score collapses (Gemini 0.0361, Claude 0.007, GPT-5 0.0362, Seed1.5/1.6-VL 0.0611/0.0547). This demonstrates that **while code-level compliance is generally high, it does not guarantee pixel-level visual correctness, making figure-level fidelity the primary bottleneck**.

**Chart Reproduction Challenge:** In Tab. 3, proprietary models still execute but with lower fidelity (e.g., Gemini CFD ER 87.04 with LMM-Score ≈0.22; Claude 88.89/0.227; GPT-5 85.19/0.238). Open-source models suffer larger drops (e.g., InternVL-3-8B 57.41/0.103, Qwen2-VL-72B 51.85/0.137; several models hit 0 ER). Compared to DM/CRD in the same table, CFD exposes weaknesses in axis/series alignment, legend consistency, scaling, and style carry-over. We concluded **reproducing existing charts (CFD) is the hardest subtask in Level1**.

**Scaling Open-Source Backbones:** In Tab. 4, Qwen2.5-VL-72B reaches 71.89 ER with strong code-level, yet figure-level is only 0.1437; InternVL-3-38B shows 61.51 ER and similar code-level strength (Layout 0.9406, Type 0.9216), but figure-level remains 0.1205. This contrasts with proprietary models' ∼90% ER and still-low figure-level (≈0.18–0.22), underscoring that fidelity, not executability, is the persistent gap. These result shows **larger open-source backbones close part of the execution gap on level 2, but figure-level fidelity gains are modest**.

**Thinking Helps Procedural Compliance:** On level 2 (4), MiMo-VL-7B-RL ER rises from 16.54 → 28.32 when enabling thinking; MiMo-VL-7B-SFT nudges 22.27 → 23.57. LLM-side (code-level GPT-Score) also improves slightly. However, figure-level remains low or mixed (e.g., MiMo-SFT 0.1203 → 0.1367; MiMo-RL thinking row lacks figure-level). The net effect suggests that chain-of-thought/planning aids procedural compliance, yet post-render pixel-level exactness requires additional

mechanisms (e.g., render-then-verify loops). This indicates "Thinking" variants help instruction following and executability, but visual fidelity improvements are inconsistent.

**Metric Sensitivity:** In Level 1 (Tab. 3), Seed-1.6-VL shows an unusually high DM LMM-Score ≈0.812, far above peers. In level 2 (Tab. 4), MiMo-VL-7B-RL (non-thinking) reports an unusually high figure-level 0.4713, exceeding proprietary models (∼0.18–0.22). In Level 3 (Tab. 5), Seed1.5/1.6-VL LLM-Score = 0.0000 despite non-zero LMM-Scores (0.0611/0.0547). These inconsistencies motivate robustness checks (multi-crop/image-space perturbation, secondary scorers, human spot-checks) and a discussion on metric sensitivity to style choices. **Several metric anomalies indicate evaluator calibration and model–evaluator coupling effects that merit auditing**.

**Table-to-Chart Gap:** On Level 1 CRD (Tab. 3), multiple models achieve very high ER (e.g., Gemini 100; Qwen2.5-VL-72B 100), yet LMM-Score remains low ( 0.15–0.27 across models). On level 2 (Tab. 4), code-level Data/Text/Type scores are solid for leading models (e.g., Gemini 0.756/0.620/0.964, GPT-5 0.704/0.596/0.960), but figure-level stays around 0.18–0.22, highlighting the gap between semantic correctness and visual exactness. **Table to chart is relatively "easy to execute" but still hard to render faithfully**.

### 4.3 DISCUSSION.

**Model Performance Across Manually Defined Difficulty Levels.** In this experiment, we ask the human labeler to split each level into easy, medium and hard, in total three levels, and each subset contains 30 samples. As shown in Figure 3, model performance exhibits a clear correlation with manually annotated difficulty levels across all benchmark stages. On Level 1, proprietary models (e.g., GPT-5, Gemini-2.5-Pro, Claude-Sonnet-4) maintain relatively strong scores across Easy, Medium, and Hard subsets, though the overall fidelity remains moderate. In contrast, most open-source models show low scores and struggle particularly on harder cases. On Level 2, performance declines noticeably even for proprietary models, with overall scores dropping to ∼0.20–0.26 and sharper degradation from Easy to Hard, indicating sensitivity to increased editing complexity. By Level 3, almost all models fail regardless of difficulty level: LMM-scores converge near zero, showing that long-context table-to-chart generation overwhelms current systems. These trends suggest that **while models can partially track difficulty scaling on simpler tasks, the hardest scenarios effectively collapse their ability to produce faithful visualizations**.

**Code Generalization Holds, Visual Fidelity Lags.** As shown in Figure 4, the performance trends differ substantially when measured by LLM-score versus LMM-score. On the left, both proprietary and open-source models generalize reasonably well from Level 1 to Level 2 when evaluated with LLM-score, indicating that code-level syntax and structure can often be preserved across tasks. On the right, however, the LMM-score reveals a sharper divide: proprietary models achieve relatively higher visual fidelity on Level 1 than on Level 2, whereas open-source models perform poorly on both levels, with most scores remaining below 0.5. This contrast highlights that while models can maintain code-level compliance, translating such compliance into pixel-level faithful renderings remains a key unsolved challenge, particularly for open-source systems.

## 5 CONCLUSION AND LIMITATIONS

We presented Chart2Code, a hierarchical benchmark for chart-to-code generation that spans three progressively challenging levels: chart reproduction, chart editing, and long-table to chart generation. Our large-scale evaluation of 25 state-of-the-art LMMs shows a clear trend: while current models manage simple reproduction reasonably well, they struggle with complex editing and long-context visualization, exposing substantial gaps in practical capability. These findings underscore the unsolved challenges of chart-to-code generation and call for models with stronger reasoning, generalization, and robustness. Despite its contributions, Chart2Code has two key limitations. First, all tasks are currently in English; extending to multilingual chart2code remains an open and important direction. Second, our evaluation relies on large language models as judges to assess code correctness and visual fidelity. While this enables scalable and nuanced evaluation, it may introduce inaccuracies or biases compared to fully human assessment. Future work will explore multilingual expansion and more reliable evaluation protocols, further enhancing the benchmark's coverage and trustworthiness.

## ETHICS STATEMENT

This work introduces a benchmark for chart-to-code generation without involving any sensitive personal data or human subject experiments. All datasets are derived from publicly available or synthetically generated tables and charts, ensuring compliance with privacy and legal considerations. We acknowledge potential risks of misuse (e.g., generating misleading visualizations), and therefore release the benchmark with clear documentation and intended use guidelines. We affirm adherence to the ICLR Code of Ethics throughout the research process.

## REPRODUCIBILITY STATEMENT

We have made extensive efforts to ensure reproducibility. Detailed dataset construction steps, task definitions, and evaluation protocols are described in Section 3. Implementation details of experiments, including hyperparameters and evaluation scripts, are provided in Appendix. In addition, we release the benchmark dataset and evaluation code as anonymous supplementary materials to enable independent verification of our results.

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

## A  LLM USAGE STATEMENT

We disclose the use of Large Language Models (LLMs) in this research in several capacities.

First, during the preparation of this manuscript, we utilized an LLM for grammatical correction and stylistic refinement to improve the paper's readability.

Second, and central to our methodology, multiple LLMs served as the subjects of our experiments to test our proposed benchmark. Furthermore, the evaluation metrics for our benchmark involved using an LLM to assess the comprehensive quality of the results.

We explicitly state that we have never relied on LLMs to generate core research ideas, methodologies, experimental designs, or conclusions. All technical contributions and analyses presented herein are the original work of the authors.

## B  USER-CENTRIC CASE STUDIES

In this section, we showcase representative examples that reflect scenarios commonly encountered by human users. One example is a Level 2 task ("Error Sample"), where the model must not only generate chart code but also edit the original data to produce the target visualization. We observe that most Large Multimodal Models (LMM) fail on this seemingly routine setting, which **highlights their difficulty in handling tasks that are trivial for humans**.

Moreover, as illustrated in the subsequent cases ("LLM capability exploration"), existing LMMs often produce wrong answers even for basic perception tasks, such as recognizing image content or extracting key chart information. These failures indicate that **if models cannot reliably solve such everyday scenarios, it is even less likely they can succeed in the more complex challenge of chart2code**.

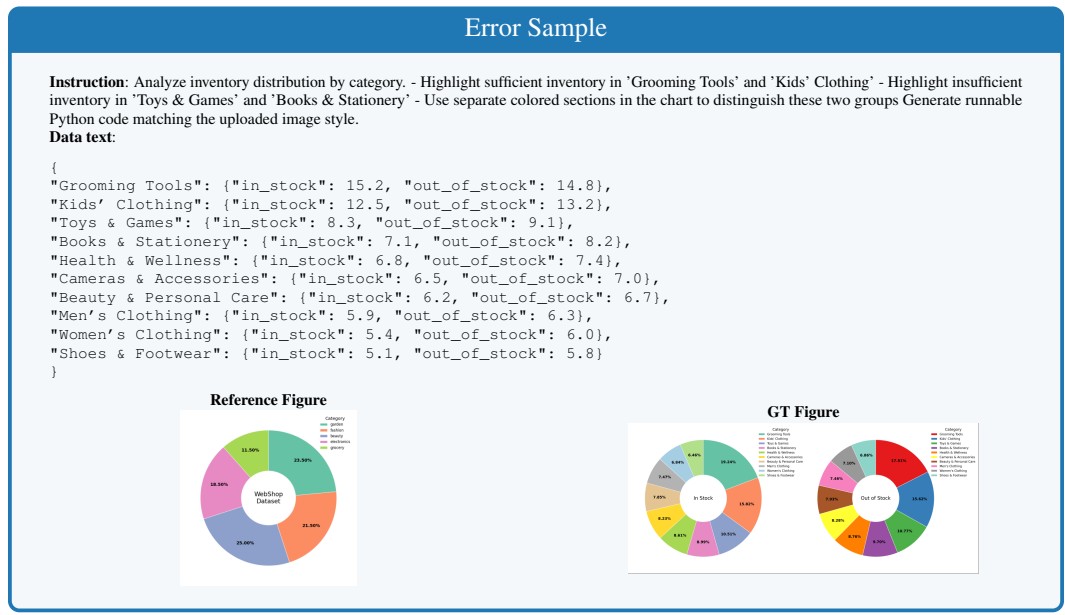

Error Sample

**Instruction**: Analyze inventory distribution by category. - Highlight sufficient inventory in 'Grooming Tools' and 'Kids' Clothing' - Highlight insufficient inventory in 'Toys & Games' and 'Books & Stationery' - Use separate colored sections in the chart to distinguish these two groups Generate runnable Python code matching the uploaded image style.

**Data text**:

```
{
"Grooming Tools": {"in_stock": 15.2, "out_of_stock": 14.8},
"Kids' Clothing": {"in_stock": 12.5, "out_of_stock": 13.2},
"Toys & Games": {"in_stock": 8.3, "out_of_stock": 9.1},
"Books & Stationery": {"in_stock": 7.1, "out_of_stock": 8.2},
"Health & Wellness": {"in_stock": 6.8, "out_of_stock": 7.4},
"Cameras & Accessories": {"in_stock": 6.5, "out_of_stock": 7.0},
"Beauty & Personal Care": {"in_stock": 6.2, "out_of_stock": 6.7},
"Men's Clothing": {"in_stock": 5.9, "out_of_stock": 6.3},
"Women's Clothing": {"in_stock": 5.4, "out_of_stock": 6.0},
"Shoes & Footwear": {"in_stock": 5.1, "out_of_stock": 5.8}
}
```

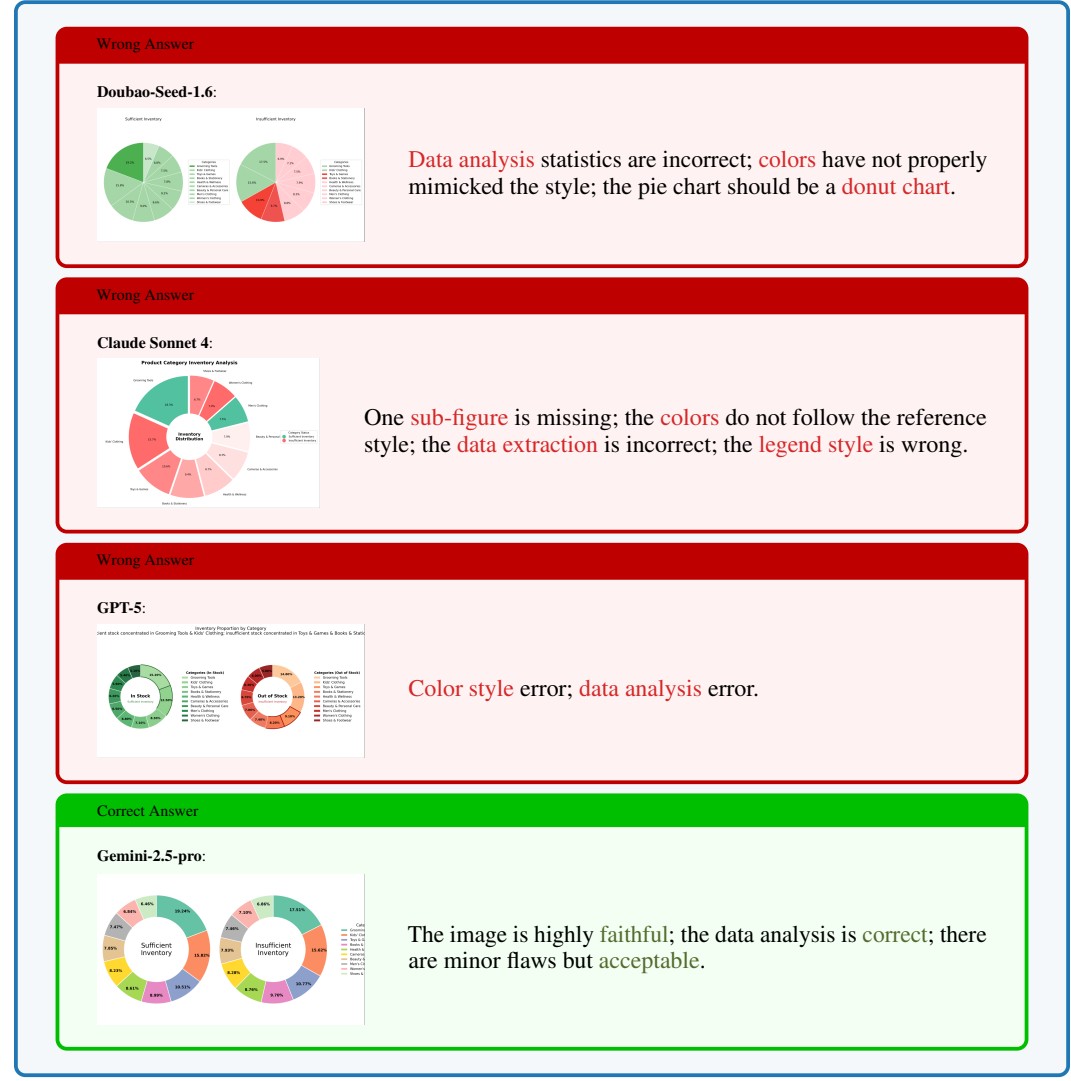

**Wrong Answer**

**Doubao-Seed-1.6**:

Data analysis statistics are incorrect; colors have not properly mimicked the style; the pie chart should be a donut chart.

**Wrong Answer**

**Claude Sonnet 4**:

One sub-figure is missing; the colors do not follow the reference style; the data extraction is incorrect; the legend style is wrong.

**Wrong Answer**

**GPT-5**:

Color style error; data analysis error.

**Correct Answer**

**Gemini-2.5-pro**:

The image is highly faithful; the data analysis is correct; there are minor flaws but acceptable.

## LLM capability exploration

**Question**: Please, based on this image, tell me if there is any title text in the picture? If so, what is its position, is it single or multiple lines, and what are its size and color?

**Figure**:

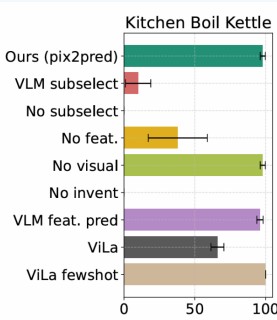

**Correct Answer**

**Answer:** Yes; Upper; Single; fontsize=12; Black;

**Correct Answer**

**Doubao-1.5-thinking-vision-pro:** Yes; Upper; Single; fontsize=14; Black;

**Correct Answer**

**Qwen2.5-VL-72B:** Yes; Center top; Single; fontsize=14-16; Black;

**Correct Answer**

**Claude-opus-4:** Yes; Center top; Single; Large; Black;

**Wrong Answer**

**Gemini-2.5-pro:** Yes; Center top; Multiple; Large; Black;

**Correct Answer**

**GPT-5:** Yes; Center top; Single; fontsize=16; Black;

---

## LLM capability exploration

**Question**: Please describe the arrangement and relative position of the legend in this chart: Is it arranged horizontally, vertically, or in a grid? Which side of the main graph is it located on (top, bottom, left, right, or embedded within)?

**Figure**:

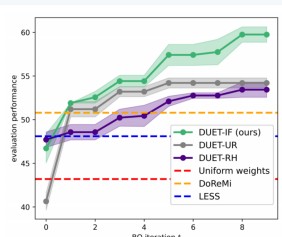

**Correct Answer**

**Answer:** Horizontally(N*1); Lower right;

**Wrong Answer**

**Doubao-1.5-thinking-vision-pro:** Horizontally(N*1); Upper right;

**Correct Answer**

**Qwen2.5-VL-72B:** Horizontally(N*1); Lower right;

---

**Wrong Answer**

**Claude-opus-4:** Horizontally(N*1); Middle right;

---

**Wrong Answer**

**Gemini-2.5-pro:** Horizontally(N*1); Middle right;

---

**Wrong Answer**

**GPT-5:** Horizontally(N*1); Right;

---

## LLM capability exploration

**Question**: Please describe the grid lines in this chart: Are they horizontal, vertical, or both? Are the lines dashed or solid?

**Figure**:

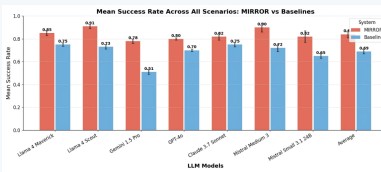

---

**Correct Answer**

**Answer:** Only horizontal grid lines; Dashed line;

---

**Wrong Answer**

**Doubao-1.5-thinking-vision-pro:** Only horizontal grid lines; Solid line;

---

**Correct Answer**

**Qwen2.5-VL-72B:** Only horizontal grid lines; Dashed line;

---

**Wrong Answer**

**Claude-opus-4:** Only horizontal grid lines; Solid line;

---

**Correct Answer**

**Gemini-2.5-pro:** Only horizontal grid lines; Dashed line;

---

**Correct Answer**

**GPT-5:** Only horizontal grid lines; Dashed line;

## LLM capability exploration

**Question**: Please describe the primary tick marks on the axes of this chart: whether they exist, their thickness and orientation (facing outward or inward), as well as the position and rotation angle of the tick labels.

**Figure**:

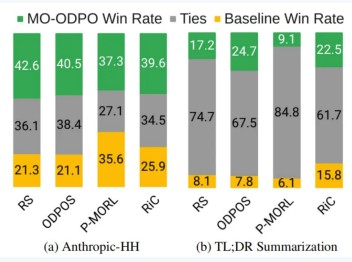

### Correct Answer

**Answer:** No; Lower; 45 degrees.

### Wrong Answer

**Doubao-1.5-thinking-vision-pro:** Implied; Lower; 0 degrees.

### Wrong Answer

**Qwen2.5-VL-72B:** No; Lower; 0 degrees.

### Wrong Answer

**Claude-opus-4:** No; Lower; 0 degrees.

### Wrong Answer

**Gemini-2.5-pro:** No; Lower; 0 degrees.

### Wrong Answer

**GPT-5:** No; Lower; 0 degrees.

## C  DATA CURATION

To construct a comprehensive and challenging chart benchmark, we collected a rich dataset of chart images and their corresponding raw data from multiple sources.

### C.0.1  CHART IMAGE DATA

Our chart image library is primarily composed of three parts, designed to cover a wide range of chart types, visual styles, and information densities.

- **Charts from Academic Literature:** We extracted chart images from approximately 5,000 PDF documents by crawling and parsing papers from the preprint server arXiv using automated scripts. These publications span from January 2024 to July 2025 and cover multiple disciplines, including computer science, physics, statistics, and economics, timestamps distribution of chart sources from

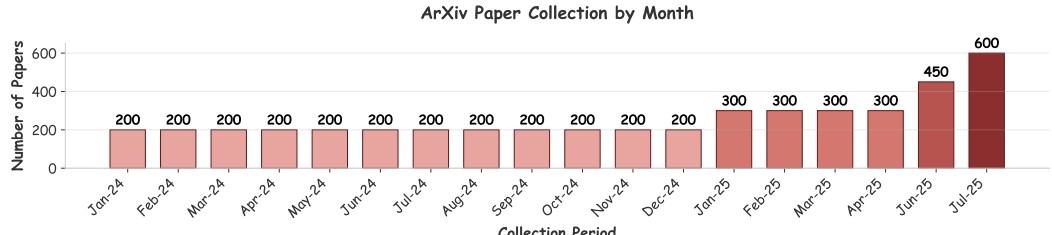

Figure 5: Timestamps distribution of chart sources from arxiv preprint.

arxiv preprint 5. This ensures that our dataset not only includes common statistical charts but also covers the highly customized and information-dense visualizations frequently found in academic research, guaranteeing both diversity and state-of-the-art relevance.

- **Examples from Programming Communities and Tutorials:** To include "standard" charts generated directly from code, we collected 1,000 example charts from the official documentation and tutorials of several mainstream data visualization libraries. Sources include official plotting examples from Matplotlib, Seaborn, Plotly, WordCloudX, Scipy, and Matlab. This portion of the data provides a set of stylistically consistent and high-quality "golden standard" references for the benchmark.

- **Existing Chart Datasets:** To further increase the difficulty of the benchmark, we selected 300 of the most structurally and elementally challenging complex charts from the existing ChartMimic (Yang et al., 2025a) dataset, based on its inherent difficulty labels and our own pre-assessment.

**Preliminary Collection and Deduplication:** First, we gathered all charts from the three aforementioned sources into a unified database. We then performed preliminary automated deduplication and format standardization.

**Coarse Filtering:** We recruited 10 senior undergraduate students majoring in computer science to conduct an initial screening of the chart pool. The screening criteria were primarily based on the clarity of the chart type (i.e., whether it is a common chart type) and its information complexity (e.g., the number of data series, density of text labels). This stage aimed to quickly eliminate ambiguous, overly simplistic, or low-quality images, reducing the dataset size from approximately 6,300 to 3,000.

**Expert Evaluation and Annotation:** We invited five doctoral students and researchers, each with over three years of experience in data visualization, to serve as experts for a fine-grained evaluation of the filtered charts. We assigned the charts to the experts by category (e.g., line charts, bar charts, scatter plots) and asked them to independently score each chart from 1 to 5 across three dimensions: **Data Complexity:** Refers to the dimensional and structural complexity of the underlying data required for the chart. **Visual Complexity:** Refers to the richness of visual elements in the chart, such as markers, colors, annotations, and dual axes. **Programming Complexity:** Refers to the programming skills and volume of code required to reproduce the chart, such as the need for complex layouts or custom functions. **Final Adjudication:** We selected charts that achieved a high composite score across the three dimensions and had high inter-rater agreement ($> 0.8$). For charts with disagreements, two core researchers made the final decision. Through this process, we finalized a set of 719 high-quality reference charts.

## C.1 RAW DATA FILTERING

**Automated Preprocessing:** We developed automated scripts to parse raw data files in various formats (e.g., Excel, CSV, TXT, JSON). These scripts prioritized the selection of data tables that contain abundant numerical, time-series, or categorical information suitable for visualization.

**Manual Verification and Diversity Preservation:** Subsequently, we manually reviewed the data filtered by the scripts, discarding any incomplete or poorly formatted data. During this process, we placed special emphasis on preserving the diversity of data sources and domains to ensure the final dataset was not biased towards any specific field. Ultimately, we constructed a raw database containing 39 Excel files, 80 structured data files (such as CSVs), and 36 semi-structured text files.

|  | (a) LLM-Score for Code Evaluation | | | | | | | | | | (b) LMM-Score for Chart Evaluation | | | | | | | | | |
|---|---|---|---|---|---|---|---|---|---|---|---|---|---|---|---|---|---|---|---|---|
|  | Q1 | Q2 | Q3 | Q4 | Q5 | Q6 | Q7 | Q8 | Q9 | Q10 | Q1 | Q2 | Q3 | Q4 | Q5 | Q6 | Q7 | Q8 | Q9 | Q10 |
| Kimi-VL | 0.63 | 0.38 | 0.00 | 0.62 | 0.35 | 0.00 | 0.31 | 0.26 | 0.61 | 0.32 | 0.29 | 0.16 | 0.00 | 0.15 | 0.05 | 0.00 | 0.05 | 0.03 | 0.36 | 0.27 |
| InternVL-3.5-8B | 0.43 | 0.42 | 0.30 | 0.59 | 0.00 | 0.00 | 0.00 | 0.00 | 0.51 | 0.62 | 0.06 | 0.08 | 0.01 | 0.11 | 0.00 | 0.00 | 0.00 | 0.00 | 0.29 | 0.15 |
| Deepseek-VL | 0.07 | 0.00 | 0.25 | 0.26 | 0.00 | 0.38 | 0.00 | 0.00 | 0.59 | 0.00 | 0.01 | 0.00 | 0.04 | 0.15 | 0.00 | 0.16 | 0.00 | 0.00 | 0.21 | 0.00 |
| Qwen-2-VL-72B | 0.00 | 0.33 | 0.33 | 0.60 | 0.38 | 0.34 | 0.26 | 0.00 | 0.00 | 0.00 | 0.00 | 0.01 | 0.09 | 0.06 | 0.02 | 0.11 | 0.08 | 0.00 | 0.00 | 0.00 |
| Qwen-2.5-VL-72B | 0.55 | 0.34 | 0.46 | 0.65 | 0.51 | 0.36 | 0.00 | 0.00 | 0.00 | 0.00 | 0.05 | 0.12 | 0.03 | 0.13 | 0.10 | 0.21 | 0.00 | 0.00 | 0.00 | 0.00 |
| Claude-Sonnet-4 | 0.33 | 0.68 | 0.71 | 0.61 | 0.46 | 0.75 | 0.48 | 0.40 | 0.75 | 0.54 | 0.20 | 0.52 | 0.35 | 0.13 | 0.30 | 0.13 | 0.18 | 0.27 | 0.32 | 0.32 |
| Gemini-2.5-pro | 0.00 | 0.83 | 0.77 | 0.64 | 0.46 | 0.70 | 0.62 | 0.56 | 0.81 | 0.66 | 0.00 | 0.62 | 0.60 | 0.10 | 0.19 | 0.25 | 0.35 | 0.31 | 0.55 | 0.56 |
| GPT-5 | 0.51 | 0.84 | 0.79 | 0.76 | 0.52 | 0.84 | 0.66 | 0.49 | 0.68 | 0.61 | 0.36 | 0.56 | 0.30 | 0.01 | 0.34 | 0.39 | 0.25 | 0.12 | 0.47 | 0.45 |
| Seed-1.6-VL | 0.54 | 0.71 | 0.45 | 0.67 | 0.51 | 0.32 | 0.47 | 0.29 | 0.72 | 0.46 | 0.20 | 0.24 | 0.23 | 0.13 | 0.40 | 0.24 | 0.08 | 0.18 | 0.48 | 0.31 |
| Seed-1.5-VL | 0.58 | 0.72 | 0.42 | 0.66 | 0.57 | 0.58 | 0.46 | 0.37 | 0.69 | 0.55 | 0.19 | 0.22 | 0.40 | 0.10 | 0.28 | 0.23 | 0.12 | 0.12 | 0.51 | 0.33 |

Figure 6: Analysis of model performance on different task cases with LLM-score and LMM-score.

## D  MORE ANALYSIS

**Discrepancy Between LLM-Score and LMM-Score.** Figure 6 illustrates model performance across ten representative task cases, evaluated by both LLM-score for code quality (left) and LMM-score for rendered chart fidelity (right). A clear discrepancy emerges: proprietary models such as GPT-5, Gemini-2.5-Pro, and Claude-Sonnet-4 achieve consistently high LLM-scores across most tasks (often ≥0.7), indicating strong code-level compliance. However, their corresponding LMM-scores are much lower (typically ≤0.35), showing that syntactically correct code often fails to produce visually faithful charts. Open-source models, in contrast, underperform on both metrics, with particularly low LMM-scores across all tasks. This contrast highlights that current models generalize relatively well at the code level but remain fundamentally limited in achieving pixel-level chart fidelity, especially on diverse and challenging task cases.

# E METRIC DETAILS

## E.1 OVERALL

To better evaluate the performance of different models, we conduct comparative assessments from two levels: the **code-level** and the **chart-level**. Throughout the evaluation process, we first examine the executability of the generated code. The execution rate is defined as the ratio between the number of executable code snippets that successfully generate images ($s$) and the total number of tasks ($t$). Formally, the execution rate is expressed as:

$$\text{exec\_rate} = \frac{s}{t}. \tag{1}$$

The execution rate is reported as a percentage.

At the **code-level**, we first extract plotting elements from the `matplotlib.Figure` object and propose eight evaluation dimensions as the **base evaluation**. The detailed specifications are given in E.2. Subsequently, we leverage `gpt-5-mini` to perform a holistic similarity assessment of the code's visualization results, thereby providing a more reliable confidence score at the code level. We refer this as **LLM-Score**.

At the **chart-level**, we input the executable code into `gpt-5-mini` for image-based evaluation. By designing specific prompts, the large multimodal model (LMM) assesses multiple dimensions and produces an aggregated score. This chart-level evaluation offers an intuitive similarity measure of the visual outputs, thereby serving as a direct indicator of model performance. We refer this as **LMM-Score**. The implementation details of these two evaluation mechanisms are described as follows.

## E.2 BASE EVALUATION

To evaluate visualization effects from the code perspective, we investigated commonly used Python plotting libraries and found that Seaborn, Matplotlib, NetworkX, and WordCloud all rely on Matplotlib's underlying plotting functions. When using these libraries for plotting, a `Figure` object is generated in memory, which contains all the elements of the plot. This implies that we can extract all visualization-related elements from the `Figure` object and compare the GT_code with the generated_code to evaluate their visualization effects.

**More Efficient.** Unlike ChartMimic (Yang et al., 2025a), which depends on code tracers and code injection, our evaluation method is substantially more efficient. In practice, ChartMimic must execute both the GT_code and generated_code for each evaluation dimension, resulting in up to twelve executions for a single generated_code. This process incurs significant computational overhead in both time and memory. By contrast, our method executes the GT_code and generated_code only once, caches their corresponding Figure objects, and then evaluates multiple dimensions directly on these objects, thereby greatly reducing execution cost.

**More General.** In comparison to ChartMimic's (Yang et al., 2025a) hard-coded rules, which exhibit limited adaptability and strong dependence on specific Matplotlib versions, our evaluation method is inherently more general. ChartMimic enforces rule-based matching of plotting elements, which not only imposes strict version constraints but also leaves many elements unsupported. Our approach instead parses the `Figure` object directly, which comprehensively encapsulates all elements in memory, ensuring greater robustness and version independence.

**More Versatile.** Whereas ChartMimic (Yang et al., 2025a) is restricted to a narrow set of functions from specific libraries, our method offers broad applicability. By operating directly on core Matplotlib objects, our approach seamlessly extends to all visualization libraries that build upon Matplotlib's primitives, thereby achieving substantially stronger cross-library generalization.

**More Precise.** Unlike ChartMimic (Yang et al., 2025a), which evaluates function call patterns rather than visual outputs, our method emphasizes the visualization results themselves. ChartMimic leaves a gap between code execution and rendered charts, while our approach directly inspects visual objects such as `Line` and `Patch`. This enables a more faithful and precise evaluation of visualization quality at the code-to-visualization level.

### E.2.1 COLOR SCORE

Traditional approaches typically treat all colors in a chart as an unordered set, neglecting the binding relationship between colors and specific data items(Yang et al., 2025a). To address this issue, we propose an efficient and more professional method for color extraction strategy designed to parse colors and their corresponding semantic information from Matplotlib's graphical objects `Figure`. This strategy decomposes the chart into different types of visual elements and organizes the extracted color information into a structured mapping, which can be expressed as:

$$\{\text{ElementType} \to \{\text{DataKey} \to \text{HexColor}\}\} \tag{2}$$

where:

- ElementType: Refers to the object to which the color is applied, such as the fill color of a bar chart (`patch_face`), the line color of a line chart (`line_color`), the color of a scatter plot (`scatter_color`), or the background of the axes (`axes_bg`).
- DataKey: Refers to the specific data entity bound to the color. This is typically the label in the legend, the tick label on the axis, or the content of a text element.
- HexColor: The standardized hexadecimal color code.

After obtaining the structured color data, we design a set of weighted evaluation metrics to quantify the color fidelity between generated_code and GT_code. The core principle of this evaluation is that not all colors are equally important. For example, errors in the colors of data series are more severe than errors in the colors of axis grid lines.

To this end, we introduce element-type weights ($w_t$), assigning a predefined weight to each `ElementType` $t$. Core data elements (e.g., `patch_face`, `line_color`) are assigned high weights (e.g., 1.0), whereas auxiliary or decorative elements (e.g., `figure_bg`, `spine`) are assigned lower weights (e.g., 0.01).

The evaluation is performed only on the element types and data keys shared by both generated_code(gen) and GT_code(gt). This ensures a valid comparison, avoiding mismatches such as comparing a line color in generated with a bar color in gt_code.

The total weighted similarity $S_{\text{total}}$ serves as the core of our model, and is computed as:

$$S_{\text{total}} = \sum_{t \in T_g en \cap T_{gt}} \sum_{k \in K_{gen,t} \cap K_{gt,t}} w_t \cdot \sigma\big(C_{gen,t,k}, C_{gt,t,k}\big), \tag{3}$$

where:

- $T_g en$ and $T_{gt}$ denote the sets of all element types present in the generated chart and the ground-truth chart, respectively.
- $K_{gen,t}$ and $K_{gt,t}$ denote the sets of all data keys under element type $t$ in the generated and ground-truth charts, respectively.
- $w_t$ is the predefined weight for element type $t$.
- $C_{gen,t,k}$ and $C_{gt,t,k}$ are the colors corresponding to element type $t$ and key $k$ in the generated and ground-truth charts, respectively.
- $\sigma(C_1, C_2)$ is a function measuring the similarity between two hexadecimal colors.

The color similarity function $\sigma(C_1, C_2)$ is used to quantify the visual closeness between two colors. In our implementation, we adopt a normalized reversed Euclidean distance in the RgenB color space to compute similarity.

First, the hexadecimal color $C$ is converted into its RGB representation $(R, G, B)$. The Euclidean distance between two colors $C_1$ and $C_2$ is defined as:

$$d(C_1, C_2) = (R_1 - R_2)^2 + (G_1 - G_2)^2 + (B_1 - B_2)^2. \tag{4}$$

The maximum possible distance in the RGB space corresponds to the distance between $(0, 0, 0)$ and $(255, 255, 255)$, i.e.,

$$d_{\text{max}} = 3 \cdot 255^2. \tag{5}$$

We then normalize the distance $d$ and transform it into a similarity score $\sigma$ within the range $[0, 1]$:

$$\sigma(C_1, C_2) = 1 - \frac{d(C_1, C_2)}{d_{\max}}. \tag{6}$$

When two colors are identical, $\sigma = 1.0$; when they differ maximally, $\sigma = 0.0$.

To provide comprehensive and interpretable evaluation results, we map the computed total weighted similarity ($S_{\text{total}}$) to three standard metrics widely used in the information retrieval domain: Precision, Recall, and F1-Score.

**Total Weight**: We first compute the total weights of the generated chart and the ground-truth chart, representing the maximum theoretically achievable similarity score.

$$W_gen = \sum_{t \in T_gen} \sum_{k \in K_{gen,t}} w_t, \quad W_{gt} = \sum_{t \in T_{gt}} \sum_{k \in K_{gt,t}} w_t. \tag{7}$$

**Precision**: Measures the accuracy of all color elements in the generated chart. It answers the question: "Among all generated colors, what proportion is correct?"

$$\text{Precision} = \frac{S_{\text{total}}}{W_gen}. \tag{8}$$

**Recall**: Measures the extent to which all color elements in the ground-truth chart are correctly reproduced in the generated chart. It answers the question: "Among all required colors, what proportion has been correctly generated?"

$$\text{Recall} = \frac{S_{\text{total}}}{W_{gt}}. \tag{9}$$

**F1-Score**: The harmonic mean of Precision and Recall, providing a single comprehensive evaluation score.

$$\text{F1-Score} = \frac{2 \cdot \text{Precision} \cdot \text{Recall}}{\text{Precision} + \text{Recall}}. \tag{10}$$

### E.2.2 GRID SCORE

We define a structured **Grid State Descriptor**. For each subplot $ax$ in a chart, we extract the visibility of its X-axis and Y-axis grid lines, and encode them as a Boolean dictionary:

$$\{'\texttt{x\_grid\_visible}' : \text{bool}, '\texttt{y\_grid\_visible}' : \text{bool}\}. \tag{11}$$

We traverse all `Axes` objects within a `Figure`, and for each subplot where at least one grid line (X-axis or Y-axis) is visible, we generate a grid state descriptor. Ultimately, the grid configuration of an entire chart is abstracted as a list of such descriptors, which can be mathematically regarded as a multiset.

For example, in a `Figure` with two subplots, where the first subplot has only Y-axis grid lines and the second subplot has both X-axis and Y-axis grid lines, the grid configuration is represented as:

$$\begin{aligned} \{'\texttt{x\_grid\_visible}' : \texttt{False}, '\texttt{y\_grid\_visible}' : \texttt{True}\}, \\ \{'\texttt{x\_grid\_visible}' : \texttt{True}, '\texttt{y\_grid\_visible}' : \texttt{True}\} \end{aligned} \tag{12}$$

This structured representation is not only precise but also completely ignores the specific styles of grid lines (e.g., color, linewidth). Instead, it focuses solely on their presence, which captures the core semantics and makes the evaluation more robust.

After extracting the multisets of grid state descriptors from the generated figure ($G_{\text{gen}}$) and the ground-truth figure ($G_{\text{gt}}$), we further use the F1 metric to measure the accuracy of this parameter.

We define the following notations:

- $G_{\text{gen}}$: the multiset of grid state descriptors extracted from the generated figure.
- $G_{\text{gt}}$: the multiset of grid state descriptors extracted from the ground-truth figure.

The number of true positives (TP) is defined as the cardinality of the intersection between the two multisets:

$$TP = |G_{\text{gen}} \cap G_{\text{gt}}|. \tag{13}$$

**True Positives (TP)** A true positive is defined as a grid state descriptor that appears in $G_{gen}$ and exactly matches one in $G_{gt}$. The total number of true positives is given by the size of the intersection of these two multisets:

$$TP = |G_{gen} \cap G_{gt}|. \tag{14}$$

**Precision** Precision measures the proportion of correctly activated grid configurations among all grid configurations in the generated figure (i.e., those that also exist in the ground-truth figure):

$$\text{Precision} = \frac{TP}{|G_{gen}|} = \frac{|G_{gen} \cap G_{gt}|}{|G_{gen}|}. \tag{15}$$

If $|G_{gen}| = 0$, we define Precision $= 1.0$.

**Recall** Recall measures the proportion of required grid configurations in the ground-truth figure that are successfully reproduced in the generated figure:

$$\text{Recall} = \frac{TP}{|G_{gt}|} = \frac{|G_{gen} \cap G_{gt}|}{|G_{gt}|}. \tag{16}$$

If $|G_{gt}| = 0$, we define Recall $= 1.0$.

**F1-Score** The F1-score, as the harmonic mean of precision and recall, provides a single comprehensive metric:

$$\text{F1-Score} = 2 \cdot \frac{\text{Precision} \cdot \text{Recall}}{\text{Precision} + \text{Recall}}. \tag{17}$$

### E.2.3 LAYOUT SCORE

For each individual subplot (i.e., an `Axes` object) in a chart, we create a unique and quantitative **Layout Descriptor**. This descriptor fully defines the size and position of the subplot within a virtual grid (`GridSpec`). Instead of relying on pixel coordinates, we extract the underlying structural information from Matplotlib's `SubplotSpec` object.

For each subplot $ax$ in a `Figure`, we extract the following six key parameters to construct its layout descriptor $D$:

- $nrows$ ($R$): the total number of rows in the corresponding `GridSpec`.
- $ncols$ ($C$): the total number of columns in the corresponding `GridSpec`.
- $row\_start$ ($r_s$): the starting row index of the grid cells occupied by the subplot.
- $row\_end$ ($r_e$): the ending row index of the grid cells occupied by the subplot.
- $col\_start$ ($c_s$): the starting column index of the grid cells occupied by the subplot.
- $col\_end$ ($c_e$): the ending column index of the grid cells occupied by the subplot.

Thus, the layout of each subplot can be precisely represented as a 6-tuple:

$$D = (R, C, r_s, r_e, c_s, c_e). \tag{18}$$

By traversing all `Axes` objects in a `Figure`, the overall layout can be abstracted as a multiset of these layout descriptors $D$, denoted as $L$.

We define the following notation:

- $L_{gen}$: the multiset of layout descriptors extracted from the generated figure.

- $L_{GT}$: the multiset of layout descriptors extracted from the ground-truth figure.

**True Positives (TP)** A true positive represents a layout descriptor that exists in $L_{gen}$ and exactly matches one in $L_{gt}$. The total number of true positives is defined as the size of the intersection of these two multisets:

$$TP = |L_{gen} \cap L_{gt}| \tag{19}$$

This indicates the number of subplots that are correctly generated and placed in the correct positions.

**Precision** Precision measures the proportion of correctly generated subplots among all generated subplots:

$$\text{Precision} = \frac{TP}{|L_{gen}|} = \frac{|L_{gen} \cap L_{gt}|}{|L_{gen}|} \tag{20}$$

Here, $|L_{gen}|$ denotes the total number of subplots in the generated figure. A low precision indicates that the model produced redundant or incorrectly placed subplots.

**Recall** Recall measures the proportion of required subplots in the ground-truth figure that were successfully generated:

$$\text{Recall} = \frac{TP}{|L_{gt}|} = \frac{|L_{gen} \cap L_{gt}|}{|L_{gt}|} \tag{21}$$

Here, $|L_{gt}|$ denotes the total number of subplots in the ground-truth figure. A low recall suggests that the model failed to generate all required subplots.

**F1-Score** The F1-score, as the harmonic mean of precision and recall, provides a single balanced metric for evaluating the overall quality of the layout:

$$\text{F1-Score} = 2 \cdot \frac{\text{Precision} \cdot \text{Recall}}{\text{Precision} + \text{Recall}} \tag{22}$$

### E.2.4 LEGEND SCORE

We propose a Dual-Constraint Matching Framework for Legend Evaluation. This framework decomposes legend evaluation into independent assessments of the semantic and spatial properties of each individual legend entry, and quantifies the consistency between the generated and ground-truth figures through a flexible matching algorithm. Consequently, it provides a more comprehensive and robust evaluation scheme.

Our method does not treat the legend as a single entity but decomposes it into a collection of independent legend entries. For each visible legend object in the chart, we traverse all its text labels and create an atomic, structured **Legend Descriptor** for each label.

The descriptor $D$ is defined as a 2-tuple that captures both semantic and spatial information:

$$D = (t, B) \tag{23}$$

where:

- $t$ is a string representing the textual content of the legend entry. This element captures the semantic correctness of the legend.
- $B$ is a 4-tuple $(x_0, y_0, x_1, y_1)$ representing the bounding box of the entire legend object containing the text entry, expressed in the screen rendering coordinate system. This element captures the spatial correctness of the legend.

By traversing all legends from both the `Axes` objects and the `Figure` object itself, we can extract all visible legend entries of a chart and represent them collectively as a multiset of descriptors $D$, denoted as $L$.

After extracting the multisets of legend descriptors $L_{gen}$ and $L_{gt}$ from the generated and ground-truth figures, respectively, we design a dual-constraint matching algorithm to compute their similarity. The algorithm can flexibly operate in two modes: semantic-only matching or combined semantic and spatial matching.

A descriptor $D_{gen} = (t_{gen}, B_{gen})$ from $L_{gen}$ matches a descriptor $D_{gt} = (t_{gt}, B_{gt})$ from $L_{gt}$ if and only if one or both of the following constraints are satisfied:

**Semantic Constraint:** The text content of the two descriptors must be identical:

$$t_{gen} = t_{gt}. \tag{24}$$

**Positional Constraint:** The bounding boxes of the legend objects containing the descriptors must have a positive intersection area:

$$\text{Area}_{intersection}(B_{gen}, B_{gt}) > 0. \tag{25}$$

For two bounding boxes $B_1 = (x_{1,0}, y_{1,0}, x_{1,1}, y_{1,1})$ and $B_2 = (x_{2,0}, y_{2,0}, x_{2,1}, y_{2,1})$, the intersection area is computed as:

$$\begin{aligned}
x_A &= \max(x_{1,0}, x_{2,0}) \\
y_A &= \max(y_{1,0}, y_{2,0}) \\
x_B &= \min(x_{1,1}, x_{2,1}) \\
y_B &= \min(y_{1,1}, y_{2,1}) \\
\text{Area}_{intersection} &= \max(0, x_B - x_A) \cdot \max(0, y_B - y_A)
\end{aligned} \tag{26}$$

The algorithm finds unique matching pairs that satisfy the above constraints (removing matched descriptors from the pool) and computes the total number of true positives (TP). Based on TP, we perform the final quantitative evaluation using standard precision, recall, and F1-score metrics:

$$\text{Precision} = \frac{TP}{|L_{gen}|}, \quad \text{Recall} = \frac{TP}{|L_{gt}|}, \quad \text{F1-Score} = 2 \cdot \frac{\text{Precision} \cdot \text{Recall}}{\text{Precision} + \text{Recall}}. \tag{27}$$

### E.2.5 DATA PARAMETER SCORE

The primary goal of data visualization is to faithfully and accurately convey the underlying data. We introduce an evaluation framework designed to quantify the fidelity of a chart's *data parameters*. This framework inspects the chart at a deep level, directly verifying the correctness of its underlying data.

The first step of the framework is to identify and extract the *data parameters* that directly define the data representation of the chart. Through introspection of Matplotlib plotting elements, we categorize these parameters into distinct types. The set of data parameters, denoted as $K_{data}$, is explicitly defined as:

$$K_{data} = \{\text{'xdata', 'ydata', 'offsets', 'xy', 'verts', 'width', 'height', 'sizes'}\}. \tag{28}$$

These parameters directly correspond to the geometric and positional properties of chart elements:

- For line plots (`Line2D`), we extract `xdata` and `ydata`.
- For bar charts (`Rectangle`), we extract the lower-left corner coordinates `xy`, as well as `width` and `height`.
- For filled plots (`Polygon`), we extract all vertex coordinates `verts`.
- For scatter plots (`Collection`), we extract the center coordinates `offsets` and the point sizes `sizes`.

Through this process, each chart is decomposed into a multiset $E$ of element-parameter dictionaries.

Data parameters, especially those represented as arrays, cannot be compared using simple equality operators. To robustly handle variations in data point ordering or floating-point precision, we define a dedicated similarity function $S(v_1, v_2)$. The core logic for data parameters is as follows:

**Numeric Type:** For scalar values, we use `numpy.isclose` to determine whether two floating-point numbers are approximately equal within a tolerance $\epsilon$:

$$S(v_1, v_2) = \begin{cases} 1 & \text{if } |v_1 - v_2| \leq \epsilon \\ 0 & \text{otherwise} \end{cases} \tag{29}$$

**Array-like Type:** For array data, which is crucial for evaluating data parameters, we adopt the Jaccard similarity coefficient to measure the overlap between the contents of two arrays. Let $V_1$ and $V_2$ denote the sets of elements in $v_1$ and $v_2$, respectively:

$$S(v_1, v_2) = \frac{|V_1 \cap V_2|}{|V_1 \cup V_2|} \tag{30}$$

This method is insensitive to the order of data points and accurately reflects the true content overlap between two datasets.

After quantifying the similarity between parameters, we employ a two-stage algorithm to compute the final evaluation metrics.

**Element Matching:** To address differences in element order and quantity across charts, we use a greedy optimal matching algorithm. For each element $e_{gt}$ in the ground-truth chart, the algorithm searches among elements of the same type in the generated chart to find the best match $e_{gen}^*$ that maximizes the total similarity across all parameters. This matching is performed globally, considering all parameter types. The result is a set of successful matches:

$$M = \{(e_{gen}, e_{gt})\}. \tag{31}$$

**Data Metric Computation:** Once the matching set $M$ is obtained, we focus exclusively on data parameters to aggregate the scores. The total true positive score for the data dimension, $TP_{data}$, is computed as the sum of similarities across all matched pairs. We iterate over the union of keys to ensure penalties for missing or extra parameters:

$$TP = \sum_{(e_{gen}, e_{gt}) \in M} \sum_{k \in (\text{keys}(e_{gen}) \cup \text{keys}(e_{gt})) \cap K_{data}} S(e_{gen}[k], e_{gt}[k]) \tag{32}$$

Next, we count the total number of data parameters in the generated chart and the ground-truth chart, denoted as $N_{data,gen}$ and $N_{data,gt}$, respectively. Finally, we compute the precision, recall, and F1-score for the data dimension:

$$\begin{aligned} \text{Precision} &= \frac{TP}{N_{data,gen}}, \\ \text{Recall} &= \frac{TP}{N_{data,gt}}, \\ \text{F1-Score} &= 2 \cdot \frac{\text{Precision} \cdot \text{Recall}}{\text{Precision} + \text{Recall}}. \end{aligned} \tag{33}$$

### E.2.6 VISUAL PARAMETER SCORE

The visual style of a chart is also an important component of chart reproduction quality. Visual style is governed by a set of *visual parameters*, such as line styles, marker shapes, element transparency, and so on. Correct usage of these parameters not only affects the aesthetic quality and professionalism of the chart, but also directly determines whether it adheres to specific design guidelines or user instructions. We propose a framework, running in parallel with the data parameter evaluation, specifically designed to quantify the consistency of a chart with respect to its *visual parameters*.

This framework builds upon the parameterized representation established in E.2.5. After extracting all parameters of an element, we identify the set of *visual parameters* ($K_{visual}$) by exclusion. A parameter key $k$ is classified as a visual parameter if it satisfies:

$$k \notin K_{data} \quad \text{and} \quad k \notin K_{ignore} \tag{34}$$

where $K_{data}$ is the predefined set of data parameters, and $K_{ignore}$ is the set of parameters handled by other evaluators (e.g., color). Typical visual parameters include: 'linestyle', 'linewidth', 'marker', 'markersize', 'alpha', and so on. The extraction process is performed in parallel

with that of the data parameters, but subsequent evaluation computations focus exclusively on this subset of parameters.

We employ the same general similarity function $S(v_1, v_2)$ introduced in the equation 29 and equation 30 to compare the values of visual parameters. Its robustness is equally applicable to various data types of visual parameters:

- **String type:** For parameters such as `linestyle` (e.g., '-' vs '–') or `marker` (e.g., 'o' vs 'x'), the function performs a direct string equality comparison.
- **Numeric type:** For parameters such as `linewidth` (e.g., 1.5 vs 2.0) or `alpha` (e.g., 0.8 vs 1.0), the function uses `numpy.isclose` to perform a tolerance-based comparison.

This consistent definition of similarity ensures intrinsic coherence across different evaluation dimensions.

**Element Matching:** We reuse the set of matched element pairs $M = \{(e_{gen}, e_{gt})\}$ obtained through the greedy optimal matching algorithm. This implies that the matching of elements is determined based on their overall similarity (data + visual), consistent with human perception — we always perceive an element as a whole. Establishing a match indicates that both the data and visual aspects will be evaluated for that pair.

**Visual Metric Computation:** Given the set of matched pairs $M$, we focus exclusively on the visual parameters to aggregate the scores. We compute the total true positive score for the visual dimension ($TP_{visual}$), defined as the sum of visual parameter similarities across all matched pairs:

$$TP_{visual} = \sum_{(e_{gen}, e_{gt}) \in M} \sum_{k \in (\text{keys}(e_{gen}) \cup \text{keys}(e_{gt})) \cap K_{visual}} S(e_{gen}[k], e_{gt}[k]) \tag{35}$$

Similarly, we count the total number of visual parameters in the generated and ground-truth charts, denoted as $N_{visual,gen}$ and $N_{visual,gt}$, respectively. Finally, the precision, recall, and F1-score for the visual dimension are computed as:

$$\begin{aligned}
\text{Precision}_{visual} &= \frac{TP_{visual}}{N_{visual,gen}}, \\
\text{Recall}_{visual} &= \frac{TP_{visual}}{N_{visual,gt}}, \\
\text{F1-Score}_{visual} &= 2 \cdot \frac{\text{Precision}_{visual} \cdot \text{Recall}_{visual}}{\text{Precision}_{visual} + \text{Recall}_{visual}}.
\end{aligned} \tag{36}$$

### E.2.7  TYPE SCORE

We propose an evaluation framework based on *Artist Class Introspection*. Unlike methods that rely on the visual rendering of charts, this framework directly inspects the object model constructed in memory by the plotting library (Matplotlib). By examining the core drawing *artists* (i.e., primitive graphical objects) and their associated classes, the framework deterministically and robustly infers the composition of a chart. The key idea is that Matplotlib employs different classes of artist objects for different types of plots. For example, a line plot is rendered using `Line2D` objects, whereas a bar chart is rendered using `Rectangle` objects. Leveraging this intrinsic correspondence, we can infer the chart types present in a figure by identifying which classes of artist objects it contains.

Our algorithm operates by traversing all subplots (Axes) within a `matplotlib.Figure` object and inspecting the list of artists contained in each subplot (e.g., `ax.lines`, `ax.patches`, `ax.collections`, etc.).

The algorithm aggregates all detected chart types within a figure into a *set*. This set-based representation has a significant advantage: it naturally supports the identification and evaluation of *composite charts*. For example, a chart that overlays a line plot on top of a bar chart will be recognized as containing both `bar_or_hist` and `line`.

The number of true positives is defined as the size of the intersection between the two sets, that is, the number of chart types present in both the generated chart and the reference chart:

$$TP = |T_{\text{gen}} \cap T_{\text{gt}}| \tag{37}$$

Precision measures the proportion of correct chart types among all generated chart types:

$$\text{Precision} = \frac{TP}{|T_{\text{gen}}|} = \frac{|T_{\text{gen}} \cap T_{\text{gt}}|}{|T_{\text{gen}}|} \tag{38}$$

where $|T_{\text{gen}}|$ denotes the total number of distinct chart types detected in the generated chart.

Recall measures the proportion of reference chart types that are successfully generated:

$$\text{Recall} = \frac{TP}{|T_{\text{gt}}|} = \frac{|T_{\text{gen}} \cap T_{\text{gt}}|}{|T_{\text{gt}}|} \tag{39}$$

where $|T_{\text{gt}}|$ denotes the total number of distinct chart types in the reference chart.

The F1-Score is the harmonic mean of precision and recall, providing a comprehensive evaluation metric:

$$\text{F1-Score} = \frac{2 \cdot \text{Precision} \cdot \text{Recall}}{\text{Precision} + \text{Recall}} \tag{40}$$

### E.2.8 TEXT SCORE

We propose a text evaluation framework based on *semantic categorization* and *fuzzy matching*. In this framework, all textual elements in a chart are categorized according to their functional roles, and a fuzzy matching algorithm based on edit distance is applied among texts within the same category. This enables a quantitative evaluation of chart text that is both strict and robust.

To achieve precise evaluation of textual roles, we first design an extractor (_extract_texts_from_figure) that introspects the `matplotlib Figure` object to identify and classify all visible textual elements. Instead of treating all texts as an undifferentiated set, we categorize them into predefined semantic classes.

Through this process, the entire textual content of a chart is transformed into a structured *Text Map*, denoted as $T$. Its form is a dictionary that maps each category name to the list of text strings belonging to that category: $T = \{c \rightarrow [t_1, t_2, \ldots]\}$. For example, $T_{\text{title}}$ represents the list of all subplot titles in the figure. This categorization mechanism ensures context-aware evaluation and prevents, for instance, an axis label from being incorrectly compared with a title.

After obtaining the text maps of the generated chart and the reference chart, $T_{\text{gen}}$ and $T_{\text{gt}}$, we designed an evaluation algorithm to quantify their consistency. To tolerate minor textual differences, we adopt the Levenshtein Ratio as the similarity function between two strings $s_1$ and $s_2$, denoted as $S_L(s_1, s_2)$. This function is based on computing the minimum number of single-character edits (insertions, deletions, or substitutions) required to transform one string into the other (i.e., the Levenshtein Distance), and normalizes the value to the interval $[0, 1]$:

$$S_L(s_1, s_2) = 1 - \frac{\text{LevenshteinDistance}(s_1, s_2)}{\max(|s_1|, |s_2|)} \tag{41}$$

A higher value of $S_L$ indicates greater similarity between the two strings. Identical strings achieve a similarity of 1.

Our evaluation algorithm operates independently within each semantic category. For each category $c$, the algorithm searches for the best match $t_{gt}^*$ for every generated text $t_{\text{gen}} \in T_{\text{gen},c}$ from the available reference texts $T_{\text{gt},c}$, such that $S_L(t_{\text{gen}}, t_{gt})$ is maximized. To prevent one-to-many matches, once a reference text is matched, it is removed from the candidate pool.

We then accumulate the similarity scores of all best matches across all categories to obtain a total similarity score ($TP_{\text{score}}$), which can be regarded as a weighted sum of "true positives":

$$TP_{\text{score}} = \sum_{c \in C} \sum_{t_{\text{gen}} \in T_{\text{gen},c}} \max_{t_{gt} \in T'_{\text{gt},c}} S_L(t_{\text{gen}}, t_{gt}) \tag{42}$$

where $C$ denotes the union of all text categories present in both charts, and $T'_{\text{gt},c}$ is the set of unmatched reference texts in category $c$.

Finally, we compute the total number of generated and reference texts ($N_{\text{gen}}$ and $N_{\text{gt}}$), and derive the Precision, Recall, and F1-Score as follows:

$$\text{Precision} = \frac{TP_{\text{score}}}{N_{\text{gen}}}, \quad N_{\text{gen}} = \sum_c |T_{\text{gen},c}| \tag{43}$$

$$\text{Recall} = \frac{TP_{\text{score}}}{N_{\text{gt}}}, \quad N_{\text{gt}} = \sum_c |T_{\text{gt},c}| \tag{44}$$

$$\text{F1-Score} = \frac{2 \cdot \text{Precision} \cdot \text{Recall}}{\text{Precision} + \text{Recall}} \tag{45}$$

### E.3 LLM-EVALUATION

This study designs and implements a multi-dimensional visualization code evaluation framework based on Large Language Models (LLMs). The framework does not execute code or render images; instead, it leverages the powerful code understanding and reasoning capabilities of LLMs to perform static analysis directly on the source code of both the generated and reference scripts. By decomposing the complex problem of "visual similarity" into a series of well-defined and mutually orthogonal evaluation dimensions, and by designing strict scoring instructions for each, our framework provides a comprehensive, in-depth, and interpretable quantitative assessment of chart code quality.

We deconstruct the ambiguous task of "code quality" assessment into six specific and independent evaluation dimensions, denoted as $D_i$. This approach makes the LLM's evaluation task more focused and renders the final results more diagnostic and interpretable. The six dimensions are defined as follows:

- **Data Handling and Transformation:** Evaluates the logic for processing, calculating, and transforming raw data prior to plotting.
- **Chart Type and Mapping:** Evaluates the choice of core plotting functions and the mapping of data columns to visual channels (e.g., x-axis, y-axis, size, color).
- **Visual Aesthetics:** Evaluates the settings of purely visual style parameters, such as colors, line styles, and markers.
- **Labels, Titles, and Legend:** Evaluates the presentation and content of all textual elements.
- **Figure Layout and Axes:** Evaluates the canvas size, subplot structure, axis ranges, and scales.
- **Auxiliary Elements and Ticks:** Evaluates the configuration of auxiliary elements such as grid lines, reference lines, and axis spines.

The evaluation prompt is in K.2

### E.4 LMM-EVALUATION

The ultimate criterion for evaluating automatically generated charts should be human visual perception. Although programmatic evaluation and source code analysis can technically ensure the correctness of chart components and parameters, they may not fully capture all visual details, artifacts, or the overall aesthetic coherence in the final rendered image. To establish an evaluation system that more closely approximates a "gold standard," we argue for the necessity of directly assessing the final visual output—the chart image itself.

To this end, this study designs and implements a holistic chart image evaluation framework based on Vision-Language Models (VLMs). This framework utilizes advanced multimodal large models by simultaneously providing them with both the reference and the generated images, supplemented by a set of rigorous evaluation instructions, to directly quantify the visual similarity between the two. This end-to-end visual evaluation method can capture a wide range of discrepancies, from macroscopic layout to microscopic pixel-level differences, thereby providing a comprehensive and holistic quality score. Here, we adopt a holistic evaluation approach, assessing all visual aspects in a single call. To ensure rigor, we extend and reinforce the philosophy of a **deduction-based scoring system**. The instructions require the model to assume a perfect score of 100, and then to deduct points for every visual discrepancy it finds between the two images.

The evaluation prompt is in K.2

# F    RUN CONFIGURATIONS

During the experiment, the parameter settings for various open-source and proprietary models were as follows. For details, please refer to the table below:

Table 6: Run configurations for all models. Unset values indicate that their default values are being used. For Proprietary models, we are unable to use a Top-P of exactly 1 due to their API settings, and we end up using a value of $0.99999$. Temp. denotes temperature. We use model pages' code to set up the run configurations whenever possible.

| Model | Version/HF Checkpoint | Do Sample | level 1 2 Max | level 3 Max | Temp. | Top-P |
|---|---|---|---|---|---|---|
| **Proprietary Multimodal Large Language Models** | | | | | | |
| GPT-5 OpenAI (2025) | gpt-5-2025-08-07 | | default | 55000 | 0 | 1 |
| Claude 4 Sonnet Anthropic (2025) | claude-4-sonnet-20250523 | | default | 55000 | 0 | 1 |
| Gemini-2.5-pro Comanici et al. (2025) | gemini-2.5-pro-20250617 | | default | 55000 | 0 | 1 |
| doubao-seed-1-5 Guo et al. (2025) | seed1.5-VL-20250513 | | default | 16000 | 0 | 1 |
| doubao-seed-1-6 Team (2025) | seed1.5-VL-20250625 | | default | 32768 | 0 | 1 |
| **Open-Source Multimodal Large Language Models** | | | | | | |
| Qwen2-VL-7B Wang et al. (2024a) | Qwen/Qwen2-VL-7B-Instruct | True | 8192 | 32768 | 0.1 | 0.95 |
| Qwen2-VL-72B Wang et al. (2024a) | Qwen/Qwen2-VL-72B-Instruct | True | 8192 | 32768 | 0.1 | 0.95 |
| Qwen2.5-VL-7B Bai et al. (2025) | Qwen/Qwen2.5-VL-7B-Instruct | True | 8192 | 32768 | 0.1 | 0.95 |
| qwen2.5-VL-72B Bai et al. (2025) | Qwen/Qwen2.5-VL-72B-Instruct | True | 8192 | 32768 | 0.1 | 0.95 |
| deepseek-VL-7B Lu et al. (2024) | deepseek-ai/deepseek-vl-7b-base | True | 8192 | 32768 | 0.1 | 0.95 |
| kimi-VL-A3B Team et al. (2025) | moonshotai/Kimi-VL-A3B-Thinking | True | 8192 | 32768 | 0.1 | 0.95 |
| MiMo-VL-7B-RL Xiaomi & Team (2025) | XiaomiMiMo/MiMo-VL-7B-RL-2508 | True | 8192 | 32768 | 0.1 | 0.95 |
| MiMo-VL-7B-SFT Xiaomi & Team (2025) | XiaomiMiMo/MiMo-VL-7B-SFT-2508 | True | 8192 | 32768 | 0.1 | 0.95 |
| GLM-4-9b GLM et al. (2024) | zai-org/glm-4-9b | True | 8192 | 32768 | 0.1 | 0.95 |
| Intern-VL 2.5 8B Chen et al. (2024) | OpenGVLab/InternVL2_5-8B | True | 8192 | 32768 | 0.1 | 0.95 |
| Intern-VL 2.5 38B Chen et al. (2024) | OpenGVLab/InternVL2_5-38B | True | 8192 | 32768 | 0.1 | 0.95 |
| Intern-VL 3 8B Zhu et al. (2025) | OpenGVLab/InternVL3-8B | True | 8192 | 32768 | 0.1 | 0.95 |
| Intern-VL 3 38B Zhu et al. (2025) | OpenGVLab/InternVL3-38B | True | 8192 | 32768 | 0.1 | 0.95 |
| Intern-VL 3.5 8B Wang et al. (2025) | OpenGVLab/InternVL3_5-8B | True | 8192 | 32768 | 0.1 | 0.95 |
| Intern-VL 3.5 38B Wang et al. (2025) | OpenGVLab/InternVL3_5-38B | True | 8192 | 32768 | 0.1 | 0.95 |
| llava-onevision-qwen2-7b-si Li et al. (2024a) | lmms-lab/llava-onevision-qwen2-7b-si | True | 8192 | 32768 | 0.1 | 0.95 |
| llava-onevision-qwen2-7b-ov Li et al. (2024a) | lmms-lab/llava-onevision-qwen2-7b-ov | True | 8192 | 32768 | 0.1 | 0.95 |

# G  OPEN-SOURCE MODEL COMPONENTS

We have listed the main components of the open-source models used in our work below:

Table 7: We summarize the visual and language components of the open-source models evaluated in our benchmark, along with the input resolutions used in our evaluation. Here, *original* denotes that we use the default image size, as the corresponding models support dynamic resolution inputs. Note that for DeepSeekVL-7B and GLM-4-9B , we apply a maximum input size constraint to accommodate their requirements.

| Model | Vision Encoder | Language Model | Resolution |
|---|---|---|---|
| Qwen2-VL-7B | Qwen2-VL ViT-14-224 | Qwen2-VL-LLM-7B | *origianl* |
| Qwen2-VL-72B | Qwen2-VL ViT-14-224 | Qwen2-VL-LLM-72B | *origianl* |
| Qwen2.5-VL-7B | Qwen2.5-VL ViT-14-224 | Qwen2.5-VL-LLM-7B | *origianl* |
| Qwen2.5-VL-72B | Qwen2.5-VL ViT-14-224 | Qwen2.5-VL-LLM-72B | *origianl* |
| Deepseek-VL-7B | SigLIP-384-SO400M & SAM-ViT-Base | DeepSeek-LLM-7B | $1152 \times 1152$* |
| Kimi-VL-A3B | MoonViT | Moonlight Model | *origianl* |
| MiMo-VL | Qwen2.5-ViT | MiMo-7B | *origianl* |
| GLM-4-9B | CLIP ViT-L-14-336 | InternLM-7B | $1120 \times 1120$* |
| InternVL-2.5-8B | InternViT-6B-448px-V2_5 | internlm2_5-7b-chat | *origianl* |
| InternVL-2.5-38B | InternViT-6B-448px-V2_5 | Qwen2.5-32B-Instruct | *origianl* |
| InternVL-3-8B | InternViT-300M-448px-V2_5 | Qwen2.5-7B | *origianl* |
| InternVL-3-38B | InternViT-6B-448px-V2_5 | Qwen2.5-32B | *origianl* |
| InternVL-3.5-8B | InternViT-300M & InternViT-6B | Qwen3-8B | *origianl* |
| InternVL-3.5-38B | InternViT-300M & InternViT-6B | Qwen3-38B | *origianl* |
| llava-onevision-qwen2-7b-si | SigLIP-384-SO400M | Qwen2-7B | *origianl* |
| llava-onevision-qwen2-7b-ov | SigLIP-384-SO400M | Qwen2-7B | *origianl* |

# H  MODEL LICENSE

Table 8: Summary of licenses in models that are evaluated in CharXiv. Entries marked with "Not Applicable" indicate that authors do not have an explicit code license displayed within the codebase or model checkpoint page.

| Name | Model License | Code License |
|------|---------------|--------------|
| GPT-5 | Proprietary | Proprietary |
| Claude 4 Sonnet | Proprietary | Proprietary |
| Gemini-2.5-pro | Proprietary | Proprietary |
| doubao-seed-1.6 | Proprietary | Proprietary |
| doubao-seed-1.5 | Proprietary | Proprietary |
| Qwen2-VL-7B | qwen | Apache 2.0 |
| Qwen2-VL-72B | qwen | Apache 2.0 |
| qwen2.5-VL-7B | qwen | Apache 2.0 |
| qwen2.5-VL-72B | qwen | Apache 2.0 |
| deepseek-VL-7B | deepseek | MIT |
| kimi-VL-A3B | MIT | MIT |
| MiMo-VL-7B-RL | MIT | Apache 2.0 |
| MiMo-VL-7B-SFT | MIT | Apache 2.0 |
| GLM-4-9B | glm-4 | Apache 2.0 |
| Intern-VL 2.5 8B | Apache-2.0 | MIT |
| Intern-VL 2.5 38B | Apache-2.0 | MIT |
| Intern-VL 3 8B | Apache-2.0 | MIT |
| Intern-VL 3 38B | Apache-2.0 | MIT |
| Intern-VL 3.5 8B | Apache-2.0 | MIT |
| Intern-VL 3.5 38B | Apache 2.0 | MIT |
| llava-onevision-qwen2-7b-si | Apache 2.0 | Apache 2.0 |
| llava-onevision-qwen2-7b-ov | Apache 2.0 | Apache 2.0 |

# I  MODEL SOURCE

Table 9: The release time and model source of LMMs used in our benchmark.

| Model | Release Time | Source |
|-------|--------------|--------|
| *Closed-source Models* | | |
| GPT-5 | 2025-08-07 | https://openai.com/zh-Hans-CN/index/introducing-gpt-5/ |
| Claude 4 Sonnet | 2025-05-23 | https://www.anthropic.com/news/claude-4 |
| Gemini-2.5-pro | 2025-06-17 | https://deepmind.google/models/gemini/pro/ |
| doubao-seed-1.5 | 2025-05-11 | https://www.volcengine.com/product/doubao |
| doubao-seed-1.6 | 2025-06-11 | https://www.volcengine.com/product/doubao |
| *Open-source Models* | | |
| Qwen2-VL-7B | 2024-09-18 | https://huggingface.co/Qwen/Qwen2-VL-7B-Instruct |
| Qwen2-VL-72B | 2024-09-18 | https://huggingface.co/Qwen/Qwen2-VL-72B-Instruct |
| qwen2.5-VL-7B | 2025-01-26 | https://huggingface.co/Qwen/Qwen2.5-VL-7B-Instruct |
| qwen2.5-VL-72B | 2025-01-26 | https://huggingface.co/Qwen/Qwen2.5-VL-72B-Instruct |
| deepseek-VL-7B | 2024-03-09 | https://huggingface.co/deepseek-ai/deepseek-vl-7b-base |
| kimi-VL-A3B | 2024-08-20 | https://huggingface.co/moonshotai/Kimi-VL-A3B-Thinking |
| MiMo-VL-7B-RL | 2025-08-10 | https://huggingface.co/XiaomiMiMo/MiMo-VL-7B-RL-2508 |
| MiMo-VL-7B-SFT | 2025-08-10 | https://huggingface.co/XiaomiMiMo/MiMo-VL-7B-SFT-2508 |
| GLM-4-9B | 2024-06-19 | https://huggingface.co/zai-org/glm-4-9b |
| Intern-VL 2.5 8B | 2024-11-21 | https://huggingface.co/OpenGVLab/InternVL2_5-8B |
| Intern-VL 2.5 38B | 2024-11-21 | https://huggingface.co/OpenGVLab/InternVL2_5-38B |
| Intern-VL 3 8B | 2025-04-10 | https://huggingface.co/OpenGVLab/InternVL3-8B |
| Intern-VL 3 38B | 2025-04-10 | https://huggingface.co/OpenGVLab/InternVL3-38B |
| Intern-VL 3.5 8B | 2025-08-25 | https://huggingface.co/OpenGVLab/InternVL3_5-8B |
| Intern-VL 3.5 38B | 2024-08-25 | https://huggingface.co/OpenGVLab/InternVL3_5-38B |
| llava-onevision-qwen2-7b-si | 2024-07-29 | https://huggingface.co/lmms-lab/llava-onevision-qwen2-7b-si |
| llava-onevision-qwen2-7b-ov | 2024-07-25 | https://huggingface.co/lmms-lab/llava-onevision-qwen2-7b-ov |

## I.1 LEVEL 1

---

### Level 1 Direct sample 1

**Instruction**: You are a Python developer proficient in data visualization, with expertise in using libraries such as Matplotlib, NetworkX, Seaborn, and others.I have a plot generated by Python code, but I don't have the corresponding code that generated this plot. Your task is to generate the Python code that can perfectly reproduce the picture based on the image I provide.

Here are the requirements for the task: 1. Data Extraction: Extract the actual data from the provided image. Based on the visual features of the plot, you must infer the data and recreate the plot. 2. Recreate the Image: Generate the Matplotlib code that reproduces the image exactly as it appears, including all elements such as: - Plot type (scatter, line, bar, etc.) - Axis labels and titles - Colors, markers, line styles, and other visual styles - Any legends, annotations, or gridlines present in the image 3. Self-contained Code: The Python code should be complete, executable, and self-contained. It should not require any external data files or variables not already present in the code. Your objective is to extract the any necessary details from the image and generate a Python script that accurately reproduces the plot.

Now, please generate the Python code to reproduce the picture below.

**Reference figure**:

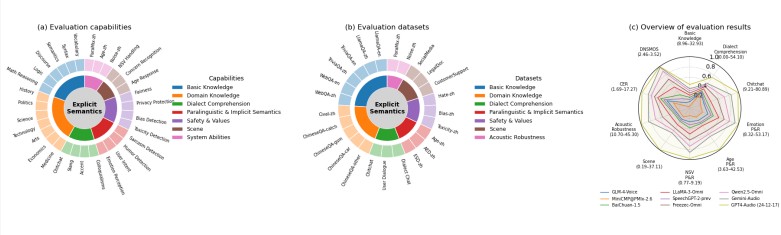

**GT Code**:

```
# == CB_38 figure code ==
import matplotlib.pyplot as plt
import numpy as np
import matplotlib.colors as mcolors

# == CB_38 figure data ==
capabilities = {
    'Basic_Knowledge': [
        'Vocabulary', 'Syntax', 'Semantics', 'Discourse', 'Logic', 'Math_Reasoning'
    ],...}
datasets = ...
def lighten_color(color, amount=0.5):
    rgb = mcolors.to_rgb(color)
    return tuple(rgb[i] + (1.0 - rgb[i]) * amount for i in range(3))
...

inner_a, size_a, col_a, outer_a, osize_a, ocol_a = prepare_sunburst(capabilities)
inner_b, size_b, col_b, outer_b, osize_b, ocol_b = prepare_sunburst(datasets)
# == figure plot ==
fig = plt.figure(figsize=(18.0, 6.0))
plt.subplots_adjust(left=0.05, right=0.85, wspace=0.7)

# -- (a) Evaluation capabilities sunburst --
ax1 = fig.add_subplot(1, 3, 1)
...
wedges1, _ = ax1.pie(size_a, radius=0.8, labels=None, startangle=90, colors=col_a, wedgeprops=dict
    (width=0.3, edgecolor='white'))
centre = plt.Circle((0, 0), 0.5, color='lightgray', linewidth=0)
ax1.add_artist(centre)
ax1.text(0, 0, 'Explicit\nSemantics',
        ha='center', va='center', fontsize=10, weight='bold')
ax1.set(aspect='equal')
ax1.set_title('(a)_Evaluation_capabilities', fontsize=12, pad=45)
ax1.legend(wedges1, inner_a, title='Capabilities', loc='center_left', bbox_to_anchor=(1.3, 0.5),
        fontsize=9, frameon=False)
# -- (b) Evaluation datasets sunburst --
...
centre2 = plt.Circle((0, 0), 0.5, color='lightgray', linewidth=0)
ax2.add_artist(centre2)
# -- (c) Overview of evaluation results (radar) --
ax3 = fig.add_subplot(1, 3, 3, projection='polar')
N = len(categories)
angles = np.linspace(0, 2*np.pi, N, endpoint=False).tolist()
angles += angles[:1]
...
ax3.xaxis.set_ticks(angles[:-1])
ax3.set_xticklabels([])
ax3.grid(True, linestyle=':')

ax3.set_yticks([0.2, 0.4, 0.6, 0.8, 1.0])
ax3.set_ylim(0, 1)
...
ax3.set_title('(c)_Overview_of_evaluation_results', fontsize=12, pad=45)
ax3.legend(loc='lower_center', bbox_to_anchor=(0.5, -0.5), ncol=3, fontsize=7, frameon=False)
```

## Level 1 Direct sample 2

**Instruction**: You are a Python developer proficient in data visualization, with expertise in using libraries such as Matplotlib, NetworkX, Seaborn, and others.I have a plot generated by Python code, but I don't have the corresponding code that generated this plot. Your task is to generate the Python code that can perfectly reproduce the picture based on the image I provide.
Here are the requirements for the task: 1. Data Extraction: Extract the actual data from the provided image. Based on the visual features of the plot, you must infer the data and recreate the plot. 2. Recreate the Image: Generate the Matplotlib code that reproduces the image exactly as it appears, including all elements such as: - Plot type (scatter, line, bar, etc.) - Axis labels and titles - Colors, markers, line styles, and other visual styles - Any legends, annotations, or gridlines present in the image 3. Self-contained Code: The Python code should be complete, executable, and self-contained. It should not require any external data files or variables not already present in the code. Your objective is to extract the any necessary details from the image and generate a Python script that accurately reproduces the plot.
Now, please generate the Python code to reproduce the picture below.
**Reference figure**:

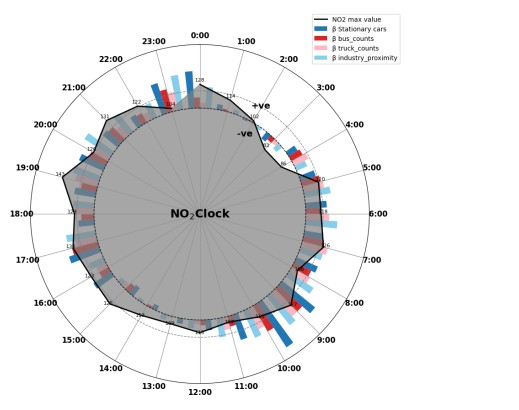

**GT Code**:

```python
import numpy as np
import matplotlib.pyplot as plt

hours = np.arange(24)
angles = 2 * np.pi * hours / 24

stationary = ...
bus_counts = np.array(...

fig = plt.figure(figsize=(10,10))
...
for th in angles:
    ax.plot([th, th], [0, 160], color='grey', linewidth=0.5)

baseline = 100
theta = np.linspace(0, 2*np.pi, 360)
ax.plot(theta, np.full_like(theta, baseline), linestyle='--', color='black', linewidth=1)
inner_circle = np.mean(no2)
ax.plot(theta, np.full_like(theta, inner_circle), linestyle='--', color='grey', linewidth=1)

ax.text(0, 0, r'NO$_2$Clock', fontsize=18, fontweight='bold', ha='center', va='center')

bar_width = 2*np.pi/24 * 0.2
offsets = np.array([-1.5, -0.5, 0.5, 1.5]) * bar_width
for vals, off, color, label in zip(
        [stationary, bus_counts, truck_counts, industry_proximity],
        offsets,
        ['tab:blue','tab:red','lightpink','skyblue'],..
    ax.bar(angles + off, vals * 100, bottom=baseline, width=bar_width, color=color, label=label)

scale = 0.8
no2_scaled = baseline + (no2 - baseline) * scale
ln, = ax.plot(angles, no2_scaled, color='black', linewidth=2, label='NO2_max_value')
ax.fill(angles, no2_scaled, color='grey', alpha=0.7)

for ang, orig_val, r in zip(angles, no2, no2_scaled):
    ax.text(ang, r + 2, f'{orig_val}', ha='center', va='bottom', fontsize=8, color='black')

ax.text(np.deg2rad(30), baseline+15, '+ve', fontsize=14, fontweight='bold', ha='center')
ax.text(np.deg2rad(30), baseline-15, '-ve', fontsize=14, fontweight='bold', ha='center')

ax.legend(loc='upper_right', bbox_to_anchor=(1.1,1.1), fontsize=10)

plt.show()
```

## Level 1 Direct sample 3

**Instruction**: You are a Python developer proficient in data visualization, with expertise in using libraries such as Matplotlib, NetworkX, Seaborn, and others.I have a plot generated by Python code, but I don't have the corresponding code that generated this plot. Your task is to generate the Python code that can perfectly reproduce the picture based on the image I provide.

Here are the requirements for the task: 1. Data Extraction: Extract the actual data from the provided image. Based on the visual features of the plot, you must infer the data and recreate the plot. 2. Recreate the Image: Generate the Matplotlib code that reproduces the image exactly as it appears, including all elements such as: - Plot type (scatter, line, bar, etc.) - Axis labels and titles - Colors, markers, line styles, and other visual styles - Any legends, annotations, or gridlines present in the image 3. Self-contained Code: The Python code should be complete, executable, and self-contained. It should not require any external data files or variables not already present in the code. Your objective is to extract the any necessary details from the image and generate a Python script that accurately reproduces the plot.

Now, please generate the Python code to reproduce the picture below.

**Reference figure**:

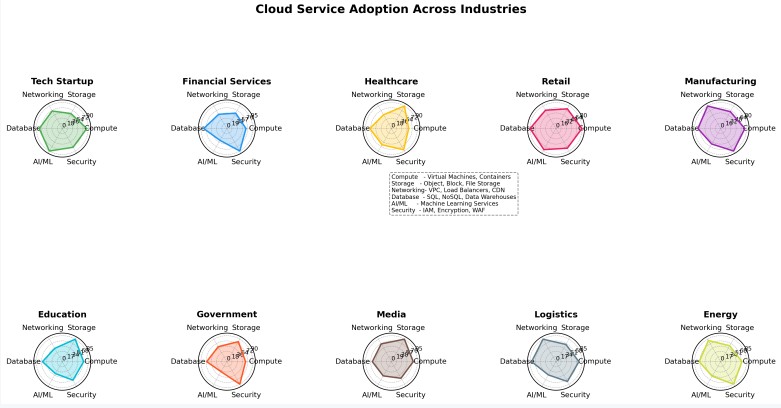

**GT Code**:

```python
import matplotlib.pyplot as plt
import numpy as np

# == New radar figure data ==

labels = ['Compute', 'Storage', 'Networking', 'Database', 'AI/ML', 'Security']
num_metrics = len(labels)
# angle of each axis in the plot (in radians)
angles = np.linspace(0, 2 * np.pi, num_metrics, endpoint=False).tolist()
# complete the loop
angles += angles[:1]

# Values for each industry's cloud service adoption (0-100 scale)
data = ...

industries = list(data.keys())

# New modern color scheme
colors = ...
# == figure plot ==

fig, axes = plt.subplots(2, 5,
                         figsize=(15.0, 9.0), # Slightly larger for readability
                         subplot_kw=dict(polar=True))
axes = axes.ravel()

for ax, name in zip(axes, industries):
    vals = data[name]
    # close the loop
    vals_loop = vals + vals[:1]
    i = industries.index(name)
    ....
    ax.set_yticks(rticks)
    ax.set_yticklabels([f"{int(x)}" for x in rticks], fontsize=8)
    ax.set_ylim(0, max_val * 1.1) # Add a small buffer to max_val

    # title
    ax.set_title(name, fontsize=12, fontweight='bold', pad=10)

    # light grid
    ax.grid(color='gray', linestyle='--', linewidth=0.5, alpha=0.7)
    ax.spines['polar'].set_linewidth(1.0)
    ...
...
plt.tight_layout(rect=[0, 0, 1, 0.96]) # Adjust layout to make space for a potential suptitle
plt.suptitle('Cloud_Service_Adoption_Across_Industries', fontsize=16, fontweight='bold', y=0.99)
plt.savefig("./datasets_level2/radar_15.png", bbox_inches="tight", dpi=300) # Save the figure
plt.show()
```

## Level 1 Customized (raw data) sample 1

**Instruction**: I want to use a heatmap to show the variation range of each category for each month, with the horizontal axis representing time and the vertical axis representing the three categories: Energy, Metals, and Food. The color intensity represents the magnitude of the variation. Please refer to the uploaded image style to generate runnable Python code.
**Reference figure**:

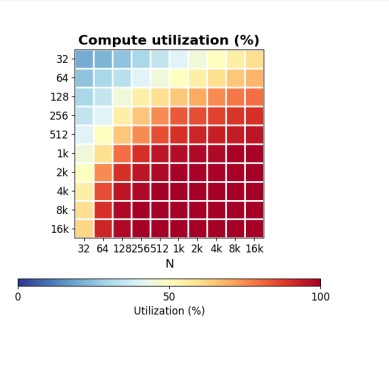

**Raw data**: "dates": [ "2020-01-01", "2020-02-01", ... "2024-08-01", "2024-09-01" ], "commodities": [ "Energy", "Metals", "Food" ], "values": [ [ 4.7, ... -8.1 ], [ 1.6, ... -4.7 ], [ 8.8, ... -0.3 ] ]
**GT Code**:

```python
import numpy as np
import matplotlib.pyplot as plt

# Data
dates = ..
commodities = ["Energy", "Metals", "Food"]
values = ..

data = np.array(values)

# Plot
fig, ax = plt.subplots(figsize=(14, 6))
fig.subplots_adjust(bottom=0.25)

# Determine symmetric range around zero
max_abs = np.max(np.abs(data))
im = ax.imshow(data, cmap='RdYlBu_r', aspect='auto', vmin=-max_abs, vmax=max_abs)

...

# Labels and title
ax.set_xlabel('Month', fontsize=14)
ax.set_title('Monthly Commodity Price Change (%)', fontsize=16, fontweight='bold')

# Gridlines
ax.set_xticks(np.arange(data.shape[1] + 1) - 0.5, minor=True)
ax.set_yticks(np.arange(data.shape[0] + 1) - 0.5, minor=True)
ax.grid(which='minor', color='white', linestyle='-', linewidth=2)
ax.tick_params(which='minor', bottom=False, left=False)

# Colorbar
cbar = fig.colorbar(im, ax=ax, orientation='horizontal', pad=0.3, aspect=40, shrink=0.8)
cbar.set_label('Change (%)', fontsize=12)
cbar.ax.tick_params(labelsize=12)

plt.show()
```

**GT Figure**:

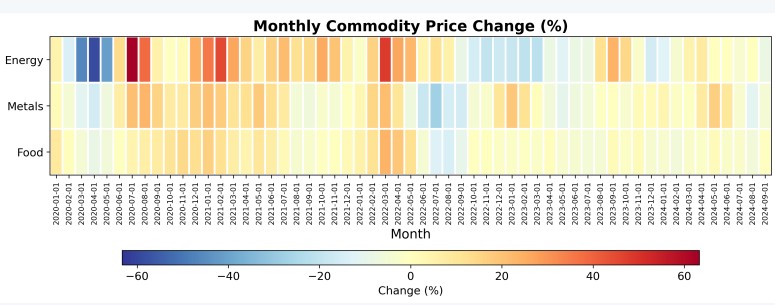

## Level 1 Customized (table figure) sample 1

**Instruction**: I want to use the data from the uploaded director compensation table (PNG) and create a combination chart based on the style of the reference combination chart: the horizontal axis represents the names of the directors, the bar chart displays cash compensation, stock awards, and total compensation respectively, and a dashed line chart highlights the trends of these three items. Thank you! Adjust the image size to match the aspect ratio of the reference image; use the dark blue, cyan, and light gray tones from the reference image; for the x-axis labels, tilt them 45 degrees and align them to the right, mimicking the text style of the reference image; add a title centered at the top, with font effects similar to the reference image; set the y-axis scale range and intervals according to the reference image; keep the legend position consistent with the reference image, arranged horizontally at the top; apply dashed line styles as in the reference image, and mimic the marker shapes from the reference image.

**Reference figure**:

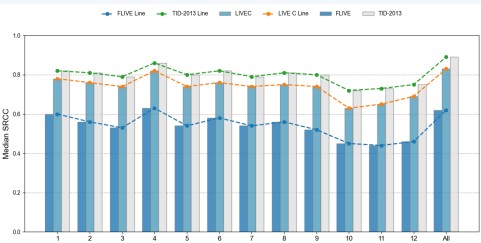

**Data figure**:

| Name | Fees Earned or Paid in Cash ($) | Stock Awards ($)(1) | Total ($) |
|---|---|---|---|
| **DIRECTOR COMPENSATION TABLE — 2024** | | | |
| Dr. Ruey-Bin Kao | 81,250 | 199,997 | 281,247 |
| Julien Mininberg | 38,599 | 199,995 | 238,594 |
| Karen Golz | 100,000 | 199,997 | 299,997 |
| Andrew Miller | 147,468 | 199,997 | 347,465 |
| Michelle Stacy | 100,000 | 199,997 | 299,997 |
| Michael Loparco | 24,643 | 166,664 | 191,307 |
| Eva Manolis | 90,307 | 199,997 | 290,304 |

**GT Code**:

```python
import numpy as np
import matplotlib.pyplot as plt

plt.rcParams.update({
    'font.family': 'sans-serif',
    'font.sans-serif': ['Arial']
})

names = ['Dr._Ruey-Bin_Kao', 'Julien_Mininberg', ..., 'Eva_Manolis']
fees = [81250, 38599, ..., 90307]
stock_awards = [199997, 199995, ..., 199997]
total = [281247, 238594, ..., 290304]

x = np.arange(len(names))
fig, ax = plt.subplots(figsize=(12, 6))

ax.grid(axis='y', linestyle='--', alpha=0.7)
ax.bar(x - 0.25, fees, 0.25, label='Fees_Earned', color='#1f77b4', alpha=0.8)
ax.bar(x, stock_awards, 0.25, label='Stock_Awards', color='#4c9dbd', alpha=0.8)
ax.bar(x + 0.25, total, 0.25, label='Total_Compensation', color='#e0e0e0', alpha=0.8)

ax.plot(x, fees, '--o', color='#1f77b4', label='Fees_Line')
ax.plot(x, stock_awards, '--o', color='#ff7f0e', label='Stock_Awards_Line')
ax.plot(x, total, '--o', color='#2ca02c', label='Total_Line')
ax.set_xticks(x)
ax.set_xticklabels(names, rotation=45, ha='right')
ax.set_ylabel('Compensation_($)')

handles, labels = ax.get_legend_handles_labels()
ax.legend(handles, labels, loc='upper_center', bbox_to_anchor=(0.5, 1.15), ncol=3)

plt.tight_layout()
plt.show()
```

**GT Figure**:

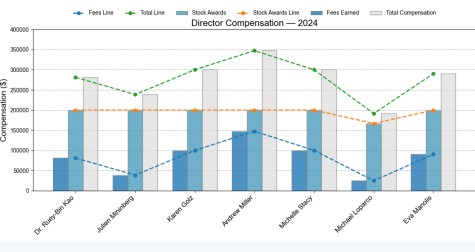

## I.2   LEVEL 2

### Level 2 sample 1

**Instruction**: Use GridSpec to create a complex 1+2 layout. The top section will feature a large subplot (spanning the entire width) to display "raincloud plots" (half-violin plots + box plots + scatter plots) for all four categories... enabling an in-depth comparison of these two distinctly different distributions. On this basis: - Set the overall canvas size to 14 inches wide × 10 inches high. - Continue using four fixed colors: light orange '#FFC0A0', light green '#B0E0B0', light purple '#B9A0E0', and beige '#FFE4C4'. Use a red line to mark the mean value in the histograms. - Use a GridSpec layout with two rows and two columns. The first row spans both columns for the top plot, while the second row places the two histograms side by side, one in each column. The row height ratio should be explicitly set to 2:1. .... - Rotate the X-axis tick labels of the top subplot counterclockwise by 20 degrees. - Maintain a white background and gray grid lines ('#D3D3D3').

**Reference figure**:

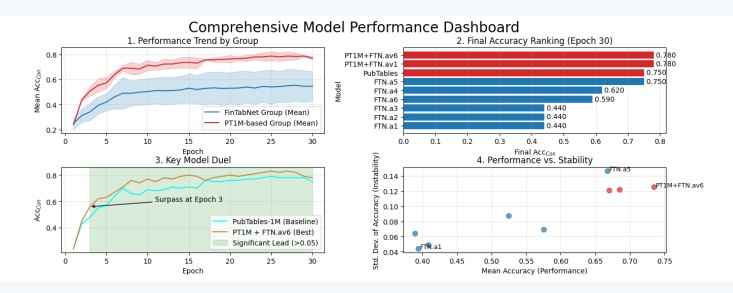

**GT Code**:

```
# == line_19 figure code ==
import matplotlib.pyplot as plt
import numpy as np
import matplotlib.gridspec as gridspec

# == line_19 figure data ==
epochs = np.arange(1, 31)

# FinTabNet variants
ftn_a1 = np.array([...
...

# == Data Processing for Dashboard ==
# 1. Group data
fintabnet_group_data = np.array([ftn_a1, ftn_a2, ftn_a3, ftn_a4, ftn_a5, ftn_a6])
pt1m_based_group_data = np.array([pubtables, pt1m_av1, pt1m_av6])
all_models_data = np.vstack([fintabnet_group_data, pt1m_based_group_data])
all_models_labels = ['FTN.a1', 'FTN.a2', 'FTN.a3', 'FTN.a4', 'FTN.a5', 'FTN.a6', 'PubTables', '
    PT1M+FTN.av1', 'PT1M+FTN.av6']
...
# 3. Final performance data
...
# 4. Significant surpass point
diff = pt1m_av6 - pubtables
surpass_margin = 0.05
surpass_epoch_idx = np.where(diff > surpass_margin)[0]
first_surpass_epoch = epochs[surpass_epoch_idx[0]] if len(surpass_epoch_idx) > 0 else None

...
# Plot 3: Key Model Showdown

...
plt.tight_layout(rect=[0, 0.03, 1, 0.95])
# plt.savefig("./datasets/line_19.png")
plt.show()
```

**GT figure**:

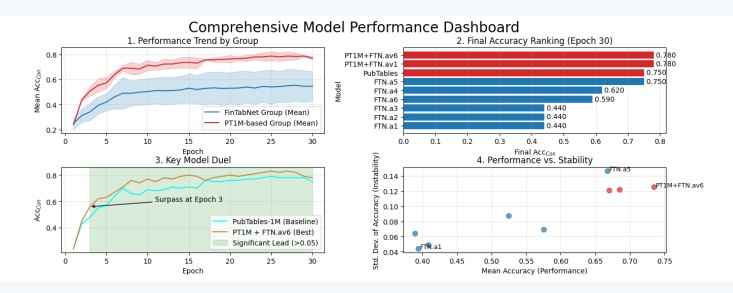

## Level 2 sample 2

**Instruction**: Create a comprehensive, dashboard-style analytical view that juxtaposes raw data trends, statistical distributions, and localized details.
1. Layout Modifications: Use 'GridSpec' to create a complex 2x2 grid layout. The top-left main plot (spanning the 1st row and 1st column) is a composite chart (three CCA lines + CKA bar chart). The top-right subplot (spanning the 1st row and 2nd column) is a box plot, used to display the overall data distribution of four data series (`cca_top1`, `cca_top3`, `cca_top10`, `cka`). The large bottom plot (spanning the 2nd row and all columns) is a "zoomed-in" view of the main plot, specifically focusing on the "Center Layer" in the range of 10 to 20 for the CCA line chart details.
2. Chart Type Conversion and Combination: In the top-right subplot, create a box plot for each of the four datasets and set appropriate labels. In the bottom zoomed-in plot, only draw the three CCA line charts and omit the CKA bar chart to emphasize the localized CCA dynamics. ...
Additional Requirements: – Set the canvas size to 15×10 inches. – Use a 2×2 'GridSpec' layout with width ratios '[2,1]' and height ratios '[1,1]'. The top-left main plot occupies the 1st row and 1st column, the top-right box plot occupies the 1st row and 2nd column, and the bottom zoomed-in plot spans the 2nd row across all columns... – For the box plots, use a fill color of '#d3d3d3', black borders, and red median lines. – For the zoomed-in region rectangle, use a gray fill with transparency 0.2, a red dashed border, and red dashed connecting lines.

**Reference figure**:

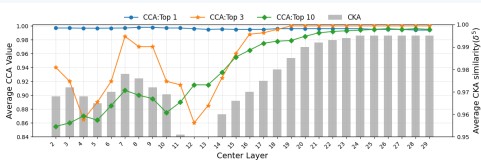

**GT Code**:

```python
import matplotlib.pyplot as plt
import matplotlib.gridspec as gridspec
from matplotlib.patches import Rectangle, ConnectionPatch
import numpy as np

layers = list(range(2, 30))
cca_top1 = [0.997, 0.997, ...
# Create figure with constrained layout
fig = plt.figure(figsize=(15, 10), constrained_layout=True)
gs = gridspec.GridSpec(2, 2, width_ratios=[2, 1], height_ratios=[1, 1], figure=fig)
...
ax_main_twin = ax_main.twinx()

# --- Main Plot (Top-Left) ---
bar_width = 0.6
...
labels = [h.get_label() for h in handles]
ax_main.legend(handles, labels, loc='lower_center', ncol=4, fontsize=10, bbox_to_anchor=(0.5,
        -0.25))

# --- Box Plot (Top-Right) ---
data_for_boxplot = [cca_top1, cca_top3, cca_top10, cka]
box_labels = ['CCA:Top_1', 'CCA:Top_3', 'CCA:Top_10', 'CKA']
...
ax_box.grid(True, axis='y', linestyle='--', linewidth=0.5, alpha=0.7)

# --- Zoomed Plot (Bottom) ---
zoom_range = (10, 20)
ax_zoom.plot(layers, cca_top1, color='#1f77b4', marker='o', markersize=6, lw=1.5)
...
ax_zoom.grid(True, linestyle='--', linewidth=0.5, alpha=0.7)

# --- Visual Connection ---
rect = Rectangle((zoom_range[0], 0.84), zoom_range[1] - zoom_range[0], 1.002 - 0.84,
            facecolor='grey', alpha=0.2, edgecolor='red', linestyle='--')
...
fig.add_artist(con2)

plt.show()
```

**GT figure**:

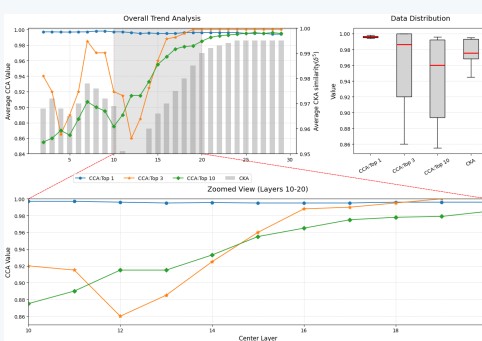

## Level 2 sample 3

**Instruction**: Create a comprehensive, dashboard-style multi-panel analysis plot to deeply explore the relationships between model performance, tool wear growth, and model comparisons. The specific requirements are as follows:

**Reference figure**:

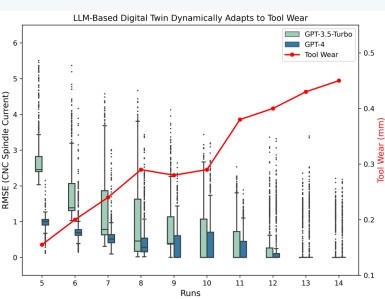

**GT Code**:

```python
import numpy as np
import matplotlib.pyplot as plt
import matplotlib.patches as mpatches
import matplotlib.gridspec as gridspec

np.random.seed(0)
runs = np.arange(5, 15)
mean35 = [2.5, ..
std35 = [0.5,...

fig = plt.figure(figsize=(18, 10))
gs = gridspec.GridSpec(2, 2, width_ratios=[3, 2], height_ratios=[1, 1])

pos1 = runs - 0.2
pos2 = runs + 0.2
..
vp1 = ax_main.violinplot(data35, positions=pos1, widths=0.4, showmedians=True)
ax_main.grid(True, linestyle="--", alpha=0.6)
ax_main.set_title("A) Model Performance Distribution vs. Tool Wear", fontsize=16, loc='left')

ax_wear = ax_main.twinx()
ax_wear.plot(runs, tool_wear, color="red", marker="o", markersize=6, linewidth=2)
ax_wear.set_ylabel("Tool Wear (mm)", color="red", fontsize=14)
...
ax_growth.grid(axis='y', linestyle='--', alpha=0.6)

median35 = [np.median(d) for d in data35]
median4 = [np.median(d) for d in data4]

highlight_run_idx = max_growth_idx + 1
ax_compare.scatter(median35, median4, c=runs, cmap='viridis', s=60, alpha=0.8)
...
ax_compare.grid(True, linestyle='--', alpha=0.6)
ax_compare.text(0.95, 0.05, 'GPT-4 Better', transform=ax_compare.transAxes,
            ha='right', va='bottom', fontsize=12, color='green', style='italic')
...
ax_main.annotate('Max Wear Growth', xy=(max_growth_run, 4.0), xytext=(max_growth_run, 5.0),
            arrowprops=dict(facecolor='#e31a1c', shrink=0.05, width=1.5, headwidth=8),
            fontsize=12, color='#e31a1c', ha='center', bbox=dict(boxstyle="round,pad=0.3", fc="
                white", ec="#e31a1c", lw=1))

fig.suptitle("Comprehensive Analysis of LLM-based Digital Twin Performance", fontsize=20, y=0.98)
plt.tight_layout(rect=[0, 0, 1, 0.95])
plt.show()
```

**GT figure**:

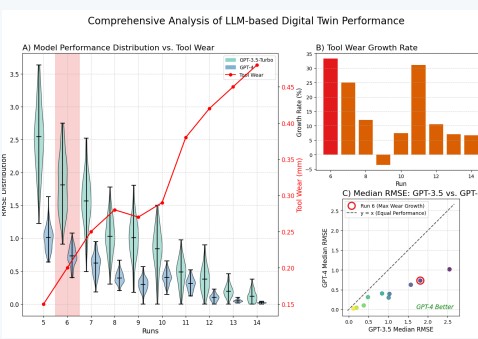

## Level 2 sample 4

**Instruction**:

1. Use 'GridSpec' to create a complex dashboard-style layout: - The left side contains a main plot occupying a 2x2 space. - The right side contains two subplots, each occupying a 1x1 space.
2. **Main Plot (Left Side)**: - Retain the original bar chart and exponential trend line. - Display the absolute values and trends of the annual research count.
3. **Top-Right Subplot**: - Convert the original data into an area chart. - Show the cumulative total of research counts to analyze the expansion of overall scale.
4. **Bottom-Right Subplot**: - Use a donut chart to display the proportion of research counts from the last three years (2022–2024) relative to their total. - Highlight the distribution of recent contributions.
5. Add titles to all subplots and ensure a unified visual style for clear communication and coordinated layout.
**Additional Modifications**: - Adjust the overall canvas size to 16 inches × 9 inches. - Configure the layout as 'GridSpec(2,3)': - The main plot occupies the first and second columns of all rows. - The top-right subplot is placed in the first row, third column. - The bottom-right subplot is placed in the second row, third column. - **Styling**: - Main plot bar color: ''#1a5276''. - Main plot trend line color: 'red'. - Area chart fill color: ''#5dade2'', line color: ''#1a5276''. - Donut chart colors: '['#1abc9c', '#f1c40f', '#e74c3c']'. - Donut chart percentage text: white and bold. - Overall title font: size 22, bold. - Subplot titles font: size 16. - Axis titles font: size 14. - Tick labels font: size 12. - Top-right chart annotations font: size 12, bold. - Donut chart center text font: size 14, bold. - Pie chart percentage text font: size 8, bold.

**Reference figure**:

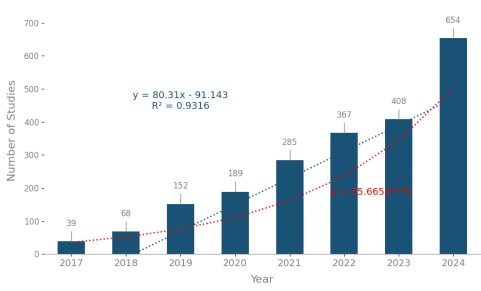

**GT Code**:

```
import numpy as np
import matplotlib.pyplot as plt
import matplotlib.gridspec as gridspec

years = np.array([2017, 2018, 2019, 2020, 2021, 2022, 2023, 2024])
x = np.arange(len(years))
...
gs = gridspec.GridSpec(2, 3, figure=fig)
ax1 = fig.add_subplot(gs[:, 0:2])
...
ax1.set_xlabel('Year', fontsize=14, color='grey')
ax1.set_ylabel('Number of Studies', fontsize=14, color='grey')
...
for spine in ['top', 'right']:
    ax2.spines[spine].set_visible(False)
ax2.grid(axis='y', linestyle='--', alpha=0.7)
ax2.text(years[-1], cumulative_y[-1], f' Total:\n {cumulative_y[-1]}', ha='right', va='top',
    fontsize=12, fontweight='bold')

colors = ['#1abc9c', '#f1c40f', '#e74c3c']
wedges, texts, autotexts = ax3.pie(last_3_years_data,...
ax3.add_artist(centre_circle)
ax3.set_title('Contribution in Last 3 Years', fontsize=16, pad=10)
ax3.text(0, 0, f'Total:\n{sum(last_3_years_data)}', ha='center', va='center', fontsize=14,
    fontweight='bold')
plt.setp(autotexts, size=8, weight="bold", color="white")

plt.tight_layout(rect=[0, 0, 1, 0.95])
plt.show()
```

**GT figure**:

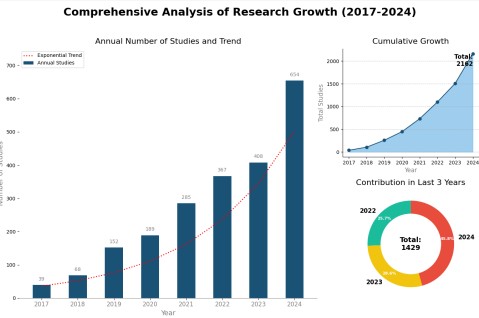

## Level 2 sample 5

**Instruction**:
Create a 2x2 dashboard to comprehensively compare model performance.
1. **Top-left plot (Performance Trend Comparison):** Divide the models into two groups: 'FinTabNet' and 'PT1M-based'. ...
2. **Top-right plot (Final Performance Ranking):** Use a horizontal bar chart to show the final accuracy of all 9 models at the last epoch..y.
3. **Bottom-left plot (Key Model Showdown):** Plot the performance curves of the best model 'pt1m_av6' and the baseline model 'pubtables' separately. Identify the epoch where 'pt1m_av6' first surpasses 'pubtables' by more than 0.05 in accuracy, and use 'axvspan' to highlight the region from ...
4. **Bottom-right plot (Performance vs. Stability):** Create a scatter plot where the X-axis represents the average accuracy of each model (mean over 30 epochs), and the Y-axis represents the standard deviation of accuracy. This plot evaluates whether high performance is accompanied by high instability. Add text labels to the best-performing, most stable, and most unstable models on the plot.
— Additional Modifications: - Set the overall canvas size to 16×12 inches. - Use a 2-row, 2-column 'GridSpec' layout with row spacing of 0.4 and column spacing of 0.3. - Use a bold font size of 20 for the main title, regular font size of 12 for subplot titles, axis labels, and tick marks, and font size of 10 for legends... and semi-transparency. Use font size 9 for labels and adjust them horizontally by 0.002. - Use dashed grid lines with approximately 30

**Reference figure**:

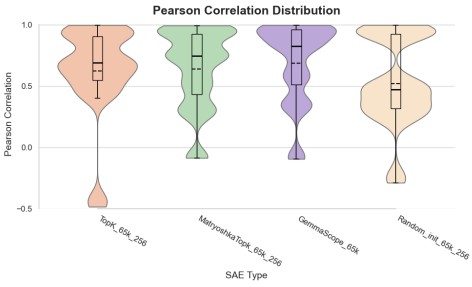

**GT Code**:

```python
import pandas as pd
import matplotlib.pyplot as plt
import seaborn as sns
import matplotlib.gridspec as gridspec

data = {
    "TopK_65k_256": [-0.4625, -0.4049, ...
clean_data = {k: [x for x in v if x is not None] for k, v in data.items()}
...
fig.suptitle("Comprehensive_Analysis_of_Pearson_Correlation", fontsize=20, fontweight='bold')

# --- Top Plot: Raincloud Plot ---
order = ["TopK_65k_256", "MatryoshkaTopk_65k_256", "GemmaScope_65k", "Random_init_65k_256"]
colors = ["#FFC0A0", "#B0E0B0", "#B9A0E0", "#FFE4C4"]
...
# Jittered points
sns.stripplot(x="SAE_Type", y="Pearson_Correlation", data=df, order=order, ax=ax_main,..

# Boxplot
sns.boxplot(x="SAE_Type", y="Pearson_Correlation", data=df, order=order, ax=ax_main,
...
ax_main.tick_params(axis='x', labelsize=12, labelrotation=-20)
ax_main.tick_params(axis='y', labelsize=12)

# --- Bottom-Right Plot: Histogram for Random_init ---
random_data = df[df["SAE_Type"] == "Random_init_65k_256"]["Pearson_Correlation"]
...
sns.despine(fig=fig)
plt.tight_layout(rect=[0, 0, 1, 0.96])
plt.show()
```

**GT figure**:

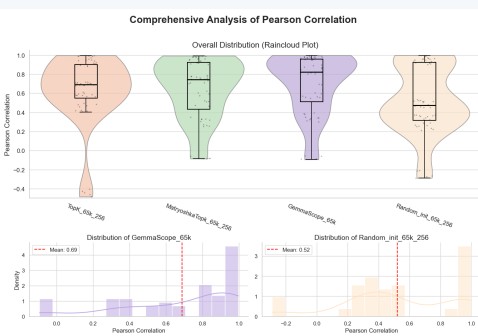

## I.3  LEVEL 3

### Level 3 sample 1

**Instruction**: I have an Excel spreadsheet to analyze, which contains fuel types and corresponding horsepower values. Please generate a plotting code based on the style of the grouped box plot I uploaded to display the horsepower distribution for different fuel types. Use a canvas size precisely 13 inches wide and 8 inches high, with the color scheme set to Set3. The entire chart should contain only one subplot, without complex layouts like GridSpec. The title should be "Horsepower by Fuel Type," the X-axis label should be "Fuel Type," and the Y-axis label should be "Horsepower (hp)." Keep all text at Matplotlib's default font size and style; rotate the X-axis tick labels 45 degrees; finally, apply a tight layout to ensure there is no excess whitespace between elements.

**Reference Figure**:

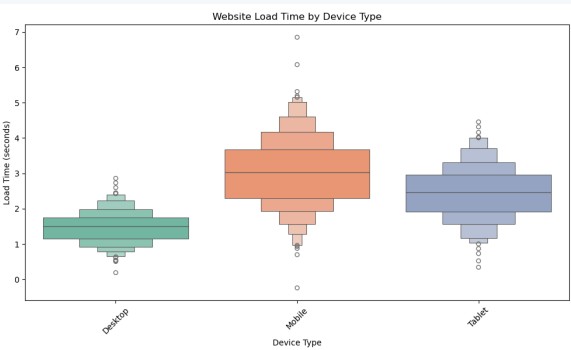

**GT Code**:

```
import matplotlib.pyplot as plt
import seaborn as sns
import pandas as pd

data_x_groups = ['plug_in_hyrbrid', 'Petrol', 'Diesel', 'Hybrid', ...]
data_y_values = [963.0, 563.0, 381.0, 1160.0, ...]

df = pd.DataFrame({
    'fuel_types': data_x_groups,
    'horsepower_num': data_y_values
})

plt.figure(figsize=(13, 8))

sns.boxenplot(data=df, x='fuel_types', y='horsepower_num', palette='Set3')

plt.title('Horsepower_by_Fuel_Type')
plt.xlabel('Fuel_Type')
plt.ylabel('Horsepower_(hp)')
plt.xticks(rotation=45)
plt.tight_layout()
plt.show()
```

**GT Figure**:

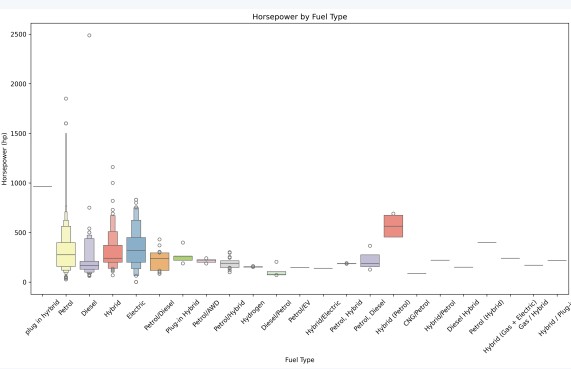

### Level 3 sample 2

**Instruction**: Based on the Excel table to be analyzed, mimic the drawing style of the image I uploaded as an attachment to create a scatter plot of Email1 length and Email2 length. The specific requirements are as follows: 1. Set the image size to 8 inches wide and 8 inches high; 2. Use cross-shaped markers for the scatter plot, with a fixed size of 200, a marker border width of 2, and map the "coolwarm" color scheme starting from sample index 1; 3. Add a color bar on the right side, with a gap of 0.05 between the color bar and the main plot, and set the aspect ratio to 1:30; 4. Add gray dashed arrows on the color bar, with the arrow style as "→", line type as dashed, line width of 2, pointing from above (2.8) to below (2.8) on the color bar scale; 5. Replace the color bar label with "Index", rotate it vertically by 90 degrees, font size 14, bold; 6. The main title of the chart is "(a) Correlation of Email1 and Email2 Lengths",

font size 24, bold, 20 units from the top edge, with a vertical position set to 1.05; 7. Both the horizontal axis title "Email1 Length" and the vertical axis title "Email2 Length" should use font size 18, bold style, with a distance of 10 units from the axis labels; 8. Fix the axis range from 10 to 40, adjust the tick label font size to 14, and do not display grid lines.

**Reference Figure**:

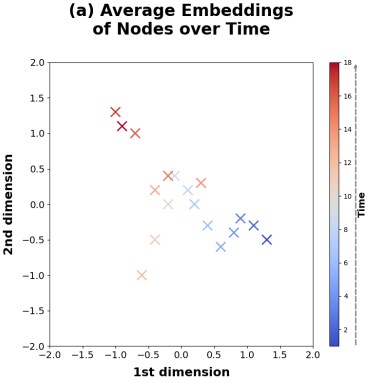

**GT Code**:

```python
import numpy as np
import matplotlib.pyplot as plt

# Data: lengths of Email1 and Email2
email1_len = np.array([18, 20, ...
email2_len = np.array([26, 23, ...

# Color by index
t = np.arange(1, len(email1_len) + 1)

# Plot
fig, ax = plt.subplots(figsize=(8, 8))
sc = ax.scatter(email1_len, email2_len, c=t, cmap='coolwarm', s=200, marker='x', linewidths=2)
..
ax.set_title(
    '(a)_Correlation\nof_Email1_and_Email2_Lengths',
    fontsize=24, fontweight='bold', pad=20, y=1.05
)
ax.set_xlabel('Email1_Length', fontsize=18, fontweight='bold', labelpad=10)
ax.set_ylabel('Email2_Length', fontsize=18, fontweight='bold', labelpad=10)
...
ax.grid(False)

plt.tight_layout()
plt.show()
```

**GT Figure**:

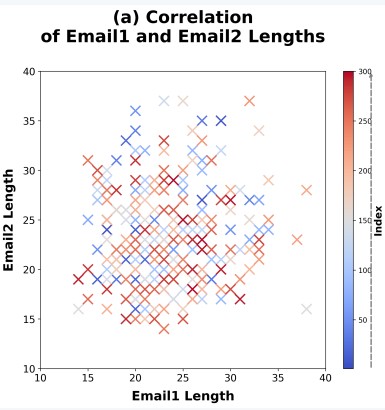

## Level 3 sample 3

**Instruction**: I have an Excel spreadsheet to analyze, which contains two columns of data: "mental_health_history" and "depression." I want to compare the distribution of depression scores between groups with and without a mental health history, mimicking the style of the image I uploaded as an attachment, and generate a box plot with a width of 10 inches and a height of 6 inches: - Use fill color "#FFA07A" for the group without a mental health history and "#20B2AA" for the group with a mental health history. The box edges, whiskers, caps, and median line colors should be "#CC8062" and "#1A8E88" (corresponding to the two groups). - Do not display outliers; - Plot scatter points offset by 0.2 on either side of the box, with scatter point colors matching the corresponding box fill color. The point edge color should be white, with an edge width of 0.5, size 50, opacity 0.8, and add random jitter of ±0.04

horizontally; - Set the overall background color to "#E5F7FD," grid line color to white, and style to solid lines; - X-axis tick labels should be "No History" and "With History," with a font size of 14; - Y-axis should display a range from 0 to 30 with a step of 5, and tick label font size should be 14; - Y-axis title should be "Depression Score," with a font size of 18 and bold; - Finally, call automatic layout adjustment to prevent label overlap.

**Reference Figure**:

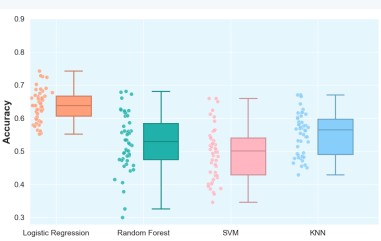

**GT Code**:

```python
import json
import numpy as np
import pandas as pd
import seaborn as sns
import matplotlib.pyplot as plt
import matplotlib.colors as mcolors

# Load data from JSON
data_json = '''
{"x":␣["0",␣"1",␣...
'''
data = json.loads(data_json)
...

# Define groups and labels
groups = [0, 1]
group_labels = ['No␣History', 'With␣History']

# Define colors
colors = ["#FFA07A", "#20B2AA"]
dark_colors = [ mcolors.to_hex(np.clip(np.array(mcolors.to_rgb(c)) * 0.8, 0, 1)) for c in colors]

# Set theme
sns.set_theme( ...

# Plot
fig, ax = plt.subplots(figsize=(10, 6))
box_offset = +0.2
point_offset = -0.2
jitter = 0.04

for i, g in enumerate(groups):
    vals = df.loc[df['mental_health_history'] == g, 'depression'].values
    # Boxplot
    ax.boxplot( ...

# Customize axes
ax.set_xticks(range(len(groups)))
ax.set_xticklabels(group_labels, fontsize=14)
...

plt.tight_layout()
plt.show()
```

**GT Figure**:

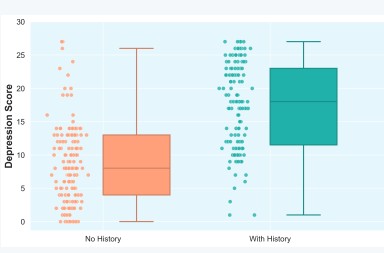

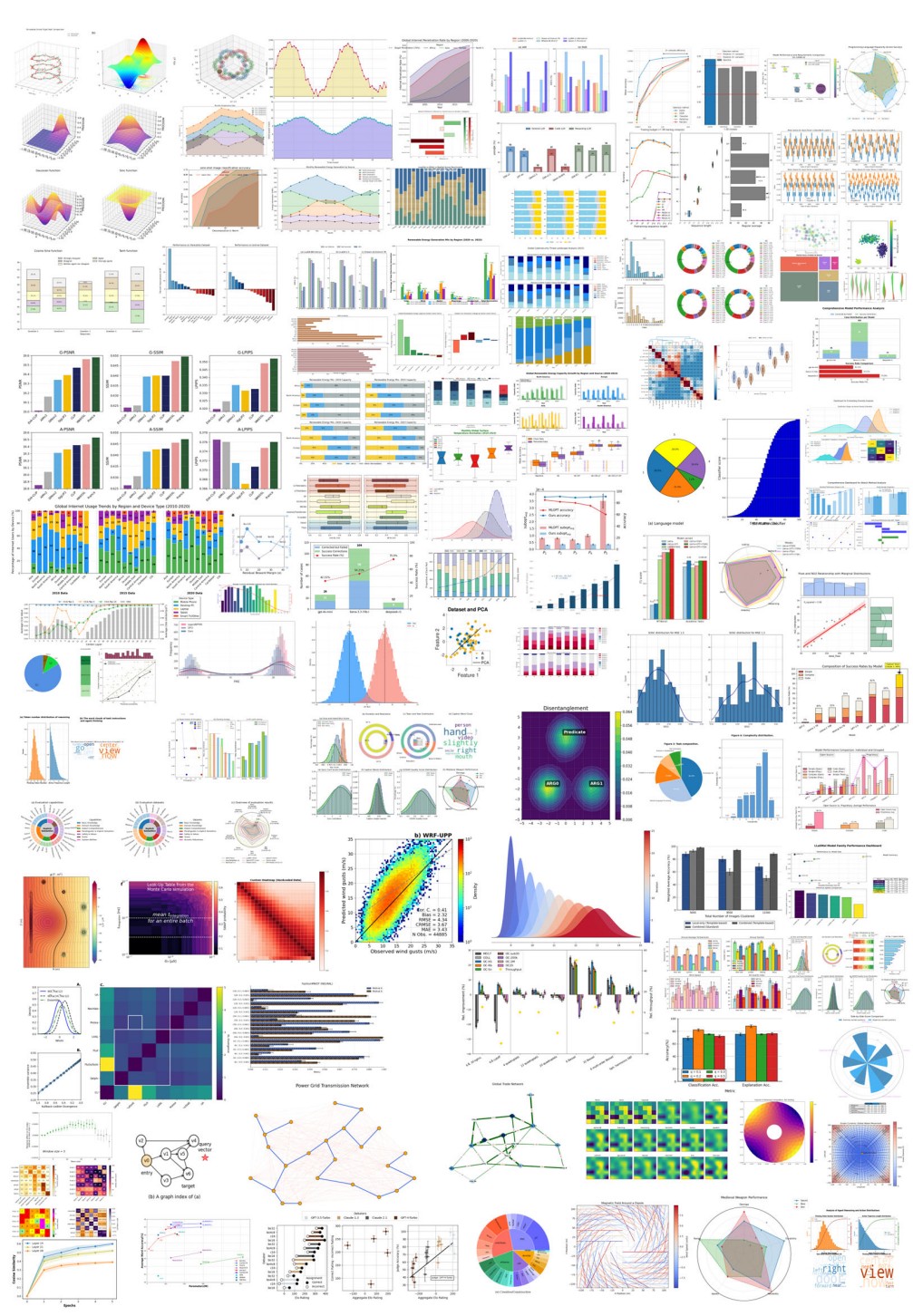

Figure 7: Selected charts of the Chart2Code.

# J   EVALUATION CODE

## J.1   COLOR

```
                              Color evaluation code

class ColorEvaluator:
    TYPE_WEIGHTS = {
        'patch_face': 1.0,
        'line_color': 1.0,
        'scatter_color': 1.0,
        'scatter_palette': 0.7,
        'text_color': 1.0,
        'poly3d_palette': 0.7,
        'patch_edge': 0.01,
        'axes_bg': 0.01,
        'figure_bg': 0.01,
        'spine': 0.01,
        'tick_label': 0.05,
        'axis_label': 0.05,
        'title': 0.05,
        'legend_text': 0.05,
        'legend_bg': 0.01,
    }
    DEFAULT_WEIGHT = 0.1

    def __init__(self) -> None:
        self.metrics = ColorMetrics()

    def __call__(self, gen_fig: Optional[Figure], gt_fig: Optional[Figure]) -> ColorMetrics:
        if gen_fig is None or gt_fig is None:
            self.metrics.status = ExecutionStatus.FAILED
            self.metrics.error_message = "can_not_find_Figure_"
            return self.metrics
        try:
            generation_data = self._extract_colors_from_figure_expert(gen_fig)
            gt_data = self._extract_colors_from_figure_expert(gt_fig)
            self._calculate_metrics(generation_data, gt_data)
        except Exception as e:
            logger.error(f"color_evaluate_error:_{e}", exc_info=True)
            self.metrics.status = ExecutionStatus.FAILED
            self.metrics.error_message = str(e)
        return self.metrics

    def _extract_colors_from_figure_expert(self, figure: Figure) -> Dict[str, Dict[str, str]]:

        extracted_data = defaultdict(dict)
        fallback_counters = defaultdict(int)

        if color := convert_color_to_hex(figure.patch.get_facecolor()): extracted_data['figure_bg'][
            'figure'] = color

        for ax in figure.axes:
            if color := convert_color_to_hex(ax.patch.get_facecolor()): extracted_data['axes_bg'][f'
                ax_{id(ax)}'] = color

            if ax.get_legend():
                for handle, label in zip(ax.get_legend().legend_handles, ax.get_legend().get_texts()):
                    key = label.get_text()
                    color = None
                    if hasattr(handle, 'get_facecolor'): color = convert_color_to_hex(handle.
                        get_facecolor())
                    elif hasattr(handle, 'get_color'): color = convert_color_to_hex(handle.get_color())
                    if color:
                        if isinstance(handle, plt.Rectangle): extracted_data['patch_face'][key] = color
                        else: extracted_data['line_color'][key] = color

            try:
                tick_labels = [tick.get_text() for tick in ax.get_xticklabels()]
                for i, patch in enumerate(ax.patches):
                    if color := convert_color_to_hex(patch.get_facecolor()):
                        key = tick_labels[i] if i < len(tick_labels) and tick_labels[i] else None
                        if not key: key = f"patch_{fallback_counters['patch_face']}"; fallback_counters[
                            'patch_face'] += 1
                        if key not in extracted_data['patch_face']: extracted_data['patch_face'][key] =
                            color
                    if e_color := convert_color_to_hex(patch.get_edgecolor()):
                        key = tick_labels[i] if i < len(tick_labels) and tick_labels[i] else f"
                            patch_edge_{i}"
                        extracted_data['patch_edge'][key] = e_color
            except Exception as e: logger.warning(f"handing_Patches_error:_{e}")

            try:
                for line in ax.lines:
                    if color := convert_color_to_hex(line.get_color()):
                        key = line.get_label()
```

```
2700
2701              if not key or key.startswith('_'): key = f"line_{fallback_counters['line_color
                       ']}"; fallback_counters['line_color'] += 1
2702              if key not in extracted_data['line_color']: extracted_data['line_color'][key] =
                       color
2703        except Exception as e: logger.warning(f"handing_Lines_error:_{e}")

2704        try:
2705            for collection in ax.collections:
2706                colors = collection.get_facecolors()
               if len(colors) == 0: continue
2707
2708                if len(set(map(tuple, colors))) == 1:
                    if color := convert_color_to_hex(colors[0]):
2709                        key = collection.get_label()
                        if not key or key.startswith('_'): key = f"scatter_group_{fallback_counters['
                           scatter_color']}"; fallback_counters['scatter_color'] += 1
2710                        if key not in extracted_data['scatter_color']: extracted_data['scatter_color'
                           ][key] = color
2711
2712                else:
2713                    for c in {convert_color_to_hex(c) for c in colors if c is not None}:
                        key = f"palette_color_{fallback_counters['scatter_palette']}";
2714                            fallback_counters['scatter_palette'] += 1
                        extracted_data['scatter_palette'][key] = c
2715        except Exception as e: logger.warning(f"handle_Collections_error:_{e}")
2716

2717        try:
               for text in ax.texts:
2718                if color := convert_color_to_hex(text.get_color()):
                    key = text.get_text()
2719                    if key: extracted_data['text_color'][key] = color
2720        except Exception as e: logger.warning(f"handle_Texts_error:_{e}")
2721
2722        if (color := convert_color_to_hex(ax.title.get_color())): extracted_data['title']['title'
                   ] = color
2723        if (color := convert_color_to_hex(ax.xaxis.label.get_color())): extracted_data['
                   axis_label']['xlabel'] = color
2724        if (color := convert_color_to_hex(ax.yaxis.label.get_color())): extracted_data['
                   axis_label']['ylabel'] = color
2725
2726        return dict(extracted_data)
2727
```

## J.2 GRID

### Grid evaluation code

```
2733  class GridEvaluator:
2734      def __init__(self) -> None:
          self.metrics = GridMetrics()

2735      def __call__(self, gen_fig: Optional[plt.Figure], gt_fig: Optional[plt.Figure]) -> GridMetrics:
2736          if gen_fig is None or gt_fig is None:
              self.metrics.status = ExecutionStatus.FAILED
2737              self.metrics.error_message = "Could_not_get_a_valid_Figure_object"
              return self.metrics
2738          try:
2739              generation_grids = self._extract_grids_from_figure(gen_fig)
              gt_grids = self._extract_grids_from_figure(gt_fig)
2740              self._calculate_metrics(generation_grids, gt_grids)
2741          except Exception as e:
              logger.error(f"Error_during_grid_evaluation:_{e}", exc_info=True)
2742              self.metrics.status = ExecutionStatus.FAILED
              self.metrics.error_message = str(e)
2743          return self.metrics

2744      def _extract_grids_from_figure(self, fig: plt.Figure) -> List[Dict]:
          """Directly_extracts_grid_information_from_a_Figure_object."""
2745          grids = []
          for ax in fig.axes:
2746              x_grid_visible = any(line.get_visible() for line in ax.get_xgridlines())
2747              y_grid_visible = any(line.get_visible() for line in ax.get_ygridlines())
              if x_grid_visible or y_grid_visible:
2748                  grids.append({
                    'x_grid_visible': x_grid_visible,
2749                    'y_grid_visible': y_grid_visible
                  })
2750          return grids
2751
2752      def _calculate_metrics(self, generation_grids: List[Dict], gt_grids: List[Dict]) -> None:
          """Calculates_precision,_recall,_and_F1-score_for_grid_usage."""
2753          if not generation_grids and not gt_grids:
```

```
        self.metrics.precision = 1.0; self.metrics.recall = 1.0; self.metrics.f1 = 1.0
        return
    if not gt_grids or not generation_grids:
        self.metrics.precision = 0.0; self.metrics.recall = 0.0; self.metrics.f1 = 0.0
        return
    n_correct = 0
    gt_grids_copy = gt_grids.copy()
    for gen_grid in generation_grids:
        if gen_grid in gt_grids_copy:
            n_correct += 1
            gt_grids_copy.remove(gen_grid)
    self.metrics.precision = n_correct / len(generation_grids) if generation_grids else 1.0
    self.metrics.recall = n_correct / len(gt_grids) if gt_grids else 1.0
    if self.metrics.precision + self.metrics.recall > 0:
        self.metrics.f1 = 2 * self.metrics.precision * self.metrics.recall / (self.metrics.
            precision + self.metrics.recall)
    else:
        self.metrics.f1 = 0.0
```

## J.3 LAYOUT

**Layout evaluation code**

```
class LayoutEvaluator:
    def __init__(self) -> None:
        self.metrics = LayoutMetrics()

    def __call__(self, gen_fig: Optional[plt.Figure], gt_fig: Optional[plt.Figure], gen_file_path:
        str, gt_file_path: str) -> LayoutMetrics:
        if gen_fig is None or gt_fig is None:
            self.metrics.status = ExecutionStatus.FAILED
            self.metrics.error_message = "Could_not_get_a_valid_Figure_object"
            return self.metrics
        try:
            generation_layouts = self._extract_layout_from_figure(gen_fig, gen_file_path)
            gt_layouts = self._extract_layout_from_figure(gt_fig, gt_file_path)
            self._calculate_metrics(generation_layouts, gt_layouts)
        except Exception as e:
            logger.error(f"Error_during_layout_evaluation:_{e}", exc_info=True)
            self.metrics.status = ExecutionStatus.FAILED
            self.metrics.error_message = str(e)
        return self.metrics

    def _extract_layout_from_figure(self, fig: plt.Figure, file_path: str) -> List[Dict]:
        if "/graph" in file_path:
            return [dict(nrows=1, ncols=1, row_start=0, row_end=0, col_start=0, col_end=0)]
        layout_info = []
        for ax in fig.axes:
            spec = ax.get_subplotspec()
            if spec is None: continue
            gs = spec.get_gridspec()
            nrows, ncols = gs.get_geometry()
            row_start, row_end = spec.rowspan.start, spec.rowspan.stop - 1
            col_start, col_end = spec.colspan.start, spec.colspan.stop - 1
            layout_info.append(dict(
                nrows=nrows, ncols=ncols,
                row_start=row_start, row_end=row_end,
                col_start=col_start, col_end=col_end
            ))
        return layout_info

    def _calculate_metrics(self, generation_layouts: List[Dict], gt_layouts: List[Dict]) -> None:
        if not generation_layouts and not gt_layouts:
            self.metrics.precision = 1.0; self.metrics.recall = 1.0; self.metrics.f1 = 1.0
            return
        if not gt_layouts or not generation_layouts:
            self.metrics.precision = 0.0; self.metrics.recall = 0.0; self.metrics.f1 = 0.0
            return
        n_correct = 0
        gt_layouts_copy = gt_layouts.copy()
        for layout in generation_layouts:
            if layout in gt_layouts_copy:
                n_correct += 1
                gt_layouts_copy.remove(layout)
        self.metrics.precision = n_correct / len(generation_layouts) if generation_layouts else 1.0
        self.metrics.recall = n_correct / len(gt_layouts) if gt_layouts else 1.0
        if self.metrics.precision + self.metrics.recall > 0:
            self.metrics.f1 = 2 * self.metrics.precision * self.metrics.recall / (self.metrics.
                precision + self.metrics.recall)
        else:
            self.metrics.f1 = 0.0
```

## J.4 LEGEND

Legend evaluation code

```python
class LegendEvaluator:
    def __init__(self, use_position: bool = True) -> None:
        self.metrics = LegendMetrics()
        self.use_position = use_position

    def __call__(self, gen_fig: Optional[plt.Figure], gt_fig: Optional[plt.Figure]) ->
            LegendMetrics:
        if gen_fig is None or gt_fig is None:
            self.metrics.status = ExecutionStatus.FAILED
            self.metrics.error_message = "Could_not_get_a_valid_Figure_object"
            return self.metrics
        try:
            gen_fig.canvas.draw()
            gt_fig.canvas.draw()
            generation_legends = self._extract_legends_from_figure(gen_fig)
            gt_legends = self._extract_legends_from_figure(gt_fig)
            self._calculate_metrics(generation_legends, gt_legends)
        except Exception as e:
            logger.error(f"Error_during_legend_evaluation:_{e}", exc_info=True)
            self.metrics.status = ExecutionStatus.FAILED
            self.metrics.error_message = str(e)
        return self.metrics

    def _extract_legends_from_figure(self, fig: plt.Figure) -> List[Dict]:
        legends_info = []
        renderer = fig.canvas.get_renderer()
        all_legends = fig.legends[:]
        for ax in fig.axes:
            if ax.get_legend():
                all_legends.append(ax.get_legend())
        for legend in set(all_legends):
            if not legend or not legend.get_visible():
                continue
            legend_bbox = legend.get_window_extent(renderer)
            for text_obj in legend.get_texts():
                if text_obj.get_visible() and text_obj.get_text():
                    legends_info.append({
                        "text": text_obj.get_text(),
                        "bbox": (legend_bbox.x0, legend_bbox.y0, legend_bbox.x1, legend_bbox.y1)
                    })
        return legends_info

    def _calculate_metrics(self, generation_legends: List[Dict], gt_legends: List[Dict]) -> None:
        if not generation_legends and not gt_legends:
            self.metrics.precision = 1.0; self.metrics.recall = 1.0; self.metrics.f1 = 1.0
            return
        if not gt_legends or not generation_legends:
            self.metrics.precision = 0.0; self.metrics.recall = 0.0; self.metrics.f1 = 0.0
            return
        n_correct = 0
        gt_legends_copy = gt_legends.copy()
        for gen_legend in generation_legends:
            best_match = None
            for gt_legend in gt_legends_copy:
                if gen_legend["text"] == gt_legend["text"]:
                    if self.use_position:
                        gen_box, gt_box = gen_legend["bbox"], gt_legend["bbox"]
                        xA = max(gen_box[0], gt_box[0]); yA = max(gen_box[1], gt_box[1])
                        xB = min(gen_box[2], gt_box[2]); yB = min(gen_box[3], gt_box[3])
                        interArea = max(0, xB - xA) * max(0, yB - yA)
                        if interArea > 0:
                            best_match = gt_legend
                            break
                    else:
                        best_match = gt_legend
                        break
            if best_match:
                n_correct += 1
                gt_legends_copy.remove(best_match)
        self.metrics.precision = n_correct / len(generation_legends) if generation_legends else 1.0
        self.metrics.recall = n_correct / len(gt_legends) if gt_legends else 1.0
        if self.metrics.precision + self.metrics.recall > 0:
            self.metrics.f1 = 2 * self.metrics.precision * self.metrics.recall / (self.metrics.
                precision + self.metrics.recall)
        else:
            self.metrics.f1 = 0.0
```

## J.5 VISUAL

## J.6 DATA

Data evaluation code

```python
# --- V10: Hardened Evaluator Class with Strict Logic ---
class ParameterEvaluator:
    def __init__(self) -> None:
        self.metrics = ParameterMetrics()
        self.DATA_PARAM_KEYS = {'xdata', 'ydata', 'offsets', 'xy', 'verts', 'width', 'height', '
            sizes'}
        self.IGNORED_PARAMS = {'color', 'c', 'colors', 'label', 'labels', 'edgecolor', 'facecolor'}

    def __call__(self, gen_fig: Optional[plt.Figure], gt_fig: Optional[plt.Figure]) ->
            ParameterMetrics:
        if gen_fig is None or gt_fig is None:
            self.metrics.status = ExecutionStatus.FAILED
            self.metrics.error_message = "Could_not_get_a_valid_Figure_object"
            return self.metrics
        try:
            gen_params = self._extract_params_from_figure(gen_fig)
            gt_params = self._extract_params_from_figure(gt_fig)
            self._calculate_strict_metrics(gen_params, gt_params)
        except Exception as e:
            logger.error(f"Error_during_parameter_evaluation:_{e}", exc_info=True)
            self.metrics.status = ExecutionStatus.FAILED
            self.metrics.error_message = str(e)
        return self.metrics

    def _extract_params_from_figure(self, fig: plt.Figure) -> List[Dict]:
        extracted_params = []
        for ax in fig.axes:
            for line in ax.lines:
                params = {
                    'type': 'line', 'xdata': np.array(line.get_xdata()).tolist(), 'ydata': np.array(
                        line.get_ydata()).tolist(),
                    'linestyle': line.get_linestyle(), 'linewidth': line.get_linewidth(), 'marker':
                        line.get_marker(),
                    'markersize': line.get_markersize(), 'alpha': line.get_alpha()
                }
                extracted_params.append(params)

            # --- HERE IS THE FIX ---
            # Differentiate between different types of patches
            for patch in ax.patches:
                params = {'alpha': patch.get_alpha()}
                # If it's a Rectangle (from bar, hist), get width and height
                if isinstance(patch, Rectangle):
                    params.update({
                        'type': 'rectangle_patch',
                        'xy': np.array(patch.get_xy()).tolist(),
                        'width': patch.get_width(),
                        'height': patch.get_height(),
                    })
                    extracted_params.append(params)
                # If it's a Polygon (from fill, violinplot), get vertices
                elif isinstance(patch, Polygon):
                    params.update({
                        'type': 'polygon_patch',
                        'verts': np.array(patch.get_xy()).tolist(),
                    })
                    extracted_params.append(params)
                # Can add more patch types here (e.g., Circle, Ellipse) if needed

            for collection in ax.collections:
                params = {'type': 'collection', 'alpha': collection.get_alpha()}
                if hasattr(collection, 'get_offsets'):
                    params['offsets'] = np.array(collection.get_offsets()).tolist()
                if hasattr(collection, 'get_sizes'):
                    params['sizes'] = np.array(collection.get_sizes()).tolist()
                if len(params) > 2: # Check if any data was actually added besides type and alpha
                    extracted_params.append(params)
        return extracted_params

    def _calculate_value_similarity(self, val1: Any, val2: Any) -> float:
        """Strictly_compares_two_values,_handling_numerics,_strings,_and_lists/arrays."""
```

```python
        if val1 is None and val2 is None: return 1.0
        if val1 is None or val2 is None: return 0.0
        try:
            if isinstance(val1, str): val1 = float(val1)
            if isinstance(val2, str): val2 = float(val2)
        except (ValueError, TypeError): pass

        if isinstance(val1, (int, float, np.number)) and isinstance(val2, (int, float, np.number)):
            return 1.0 if np.isclose(val1, val2) else 0.0
        if isinstance(val1, (bool, str)):
            return 1.0 if str(val1) == str(val2) else 0.0
        if isinstance(val1, (list, np.ndarray)):
            if not isinstance(val2, (list, np.ndarray)): return 0.0
            if not len(val1) and not len(val2): return 1.0
            if not len(val1) or not len(val2): return 0.0
            try:
                v1 = np.asarray(val1, dtype=float).flatten()
                v2 = np.asarray(val2, dtype=float).flatten()
                intersection = np.intersect1d(v1, v2).size
                union = np.union1d(v1, v2).size
                return intersection / union if union > 0 else 1.0
            except (ValueError, TypeError):
                set1, set2 = set(str(v) for v in val1), set(str(v) for v in val2)
                return len(set1.intersection(set2)) / len(set1.union(set2)) if set1.union(set2) else \
                    1.0
        return 0.0

    def _calculate_strict_metrics(self, gen_elements: List[Dict], gt_elements: List[Dict]):
        if not gen_elements and not gt_elements:
            self.metrics.data_metrics = self.metrics.visual_metrics = ScoreBlock(1.0, 1.0, 1.0)
            return

        total_data_score, total_visual_score = 0.0, 0.0
        gt_data_count, gt_visual_count = 0, 0
        gen_data_count, gen_visual_count = 0, 0

        unmatched_gen_elements = gen_elements[:]
        for gt_elem in gt_elements:
            best_score, best_match_index = -1.0, -1
            for i, gen_elem in enumerate(unmatched_gen_elements):
                if gt_elem.get('type') == gen_elem.get('type'):
                    current_score = sum(self._calculate_value_similarity(gt_elem.get(k), gen_elem.get(k
                        )) for k in gt_elem if k != 'type')
                    if current_score > best_score:
                        best_score, best_match_index = current_score, i

            if best_match_index != -1:
                matched_gen_elem = unmatched_gen_elements.pop(best_match_index)
                all_keys = set(gt_elem.keys()) | set(matched_gen_elem.keys())
                for key in all_keys:
                    if key in self.IGNORED_PARAMS or key == 'type': continue
                    category = 'data' if key in self.DATA_PARAM_KEYS else 'visual'
                    gt_val, gen_val = gt_elem.get(key), matched_gen_elem.get(key)
                    score = self._calculate_value_similarity(gt_val, gen_val)
                    if category == 'data': total_data_score += score
                    else: total_visual_score += score

            for key in gt_elem:
                if key in self.IGNORED_PARAMS or key == 'type': continue
                if key in self.DATA_PARAM_KEYS: gt_data_count += 1
                else: gt_visual_count += 1

        for gen_elem in gen_elements:
            for key in gen_elem:
                if key in self.IGNORED_PARAMS or key == 'type': continue
                if key in self.DATA_PARAM_KEYS: gen_data_count += 1
                else: gen_visual_count += 1

        data_p = total_data_score / gen_data_count if gen_data_count > 0 else 1.0 if not \
            gt_data_count else 0.0
        data_r = total_data_score / gt_data_count if gt_data_count > 0 else 1.0 if not \
            gen_data_count else 0.0
        data_f1 = 2 * (data_p * data_r) / (data_p + data_r) if (data_p + data_r) > 0 else 0.0
        self.metrics.data_metrics = ScoreBlock(data_p, data_r, data_f1)

        visual_p = total_visual_score / gen_visual_count if gen_visual_count > 0 else 1.0 if not \
            gt_visual_count else 0.0
        visual_r = total_visual_score / gt_visual_count if gt_visual_count > 0 else 1.0 if not \
            gen_visual_count else 0.0
        visual_f1 = 2 * (visual_p * visual_r) / (visual_p + visual_r) if (visual_p + visual_r) > 0 \
            else 0.0
        self.metrics.visual_metrics = ScoreBlock(visual_p, visual_r, visual_f1)
```

## J.7   TEXT

**Text evaluation code**

```python
class TextEvaluator:
    def __init__(self) -> None:
        self.metrics = TextMetrics()

    def __call__(self, gen_fig: Optional[plt.Figure], gt_fig: Optional[plt.Figure]) -> TextMetrics:
        if gen_fig is None or gt_fig is None:
            self.metrics.status = ExecutionStatus.FAILED
            self.metrics.error_message = "Could_not_get_a_valid_Figure_object"
            return self.metrics
        try:
            generation_texts = self._extract_texts_from_figure(gen_fig)
            gt_texts = self._extract_texts_from_figure(gt_fig)
            self._calculate_metrics(generation_texts, gt_texts)
        except Exception as e:
            logger.error(f"Error_during_text_evaluation:_{e}", exc_info=True)
            self.metrics.status = ExecutionStatus.FAILED
            self.metrics.error_message = str(e)
        return self.metrics

    def _extract_texts_from_figure(self, fig: plt.Figure) -> Dict[str, List[str]]:
        """Extracts_and_categorizes_all_text_elements_from_a_Figure_object."""
        texts = {
            "title": [], "xlabel": [], "ylabel": [], "tick_label": [],
            "suptitle": [], "legend_text": [], "annotation": []
        }
        if fig._suptitle and fig._suptitle.get_text():
            texts["suptitle"].append(fig._suptitle.get_text())

        for ax in fig.axes:
            if ax.title.get_text():
                texts["title"].append(ax.title.get_text())
            if ax.xaxis.label.get_text():
                texts["xlabel"].append(ax.xaxis.label.get_text())
            if ax.yaxis.label.get_text():
                texts["ylabel"].append(ax.yaxis.label.get_text())

            for label in ax.get_xticklabels() + ax.get_yticklabels():
                if label.get_text():
                    texts["tick_label"].append(label.get_text())

            if legend := ax.get_legend():
                for text in legend.get_texts():
                    if text.get_text():
                        texts["legend_text"].append(text.get_text())

            for text in ax.texts: # Annotations and ax.text()
                if text.get_text():
                    texts["annotation"].append(text.get_text())

        # Filter out empty lists
        return {k: v for k, v in texts.items() if v}

    def _calculate_metrics(self, generation_texts: Dict[str, List[str]], gt_texts: Dict[str, List[
        str]]) -> None:
        """Calculates_strict_metrics_based_on_categorized_text_similarity."""
        if not generation_texts and not gt_texts:
            self.metrics.precision = 1.0; self.metrics.recall = 1.0; self.metrics.f1 = 1.0
            return

        total_similarity_score = 0.0
        total_gt_text_count = sum(len(texts) for texts in gt_texts.values())
        total_gen_text_count = sum(len(texts) for texts in generation_texts.values())

        all_categories = set(gt_texts.keys()) | set(generation_texts.keys())

        for category in all_categories:
            gt_list = gt_texts.get(category, [])
            gen_list = generation_texts.get(category, [])

            if not gt_list or not gen_list:
                continue

            # Find best match for each generated text using Levenshtein ratio
            unmatched_gt = gt_list[:]
            for gen_text in gen_list:
                if not unmatched_gt: break
                best_score = -1
                best_match_index = -1
                for i, gt_text in enumerate(unmatched_gt):
                    score = levenshtein_ratio(gen_text, gt_text)
                    if score > best_score:
                        best_score = score
                        best_match_index = i
```

```
            if best_match_index != -1:
                total_similarity_score += best_score
                unmatched_gt.pop(best_match_index)

        self.metrics.precision = total_similarity_score / total_gen_text_count if
            total_gen_text_count > 0 else 1.0 if not gt_texts else 0.0
        self.metrics.recall = total_similarity_score / total_gt_text_count if total_gt_text_count >
            0 else 1.0 if not generation_texts else 0.0

        if self.metrics.precision + self.metrics.recall > 0:
            self.metrics.f1 = 2 * self.metrics.precision * self.metrics.recall / (self.metrics.
                precision + self.metrics.recall)
        else:
            self.metrics.f1 = 0.0
```

## J.8 TYPE

Type evaluation code

```python
class ChartTypeEvaluator:
    def __init__(self) -> None:
        self.metrics = ChartTypeMetrics()

    def __call__(self, gen_fig: Optional[plt.Figure], gt_fig: Optional[plt.Figure]) ->
            ChartTypeMetrics:
        if gen_fig is None or gt_fig is None:
            self.metrics.status = ExecutionStatus.FAILED
            self.metrics.error_message = "Could_not_get_a_valid_Figure_object"
            return self.metrics
        try:
            generation_chart_types = self._extract_chart_types_from_figure(gen_fig)
            gt_chart_types = self._extract_chart_types_from_figure(gt_fig)
            self._calculate_metrics(generation_chart_types, gt_chart_types)
        except Exception as e:
            logger.error(f"Error_during_chart_type_evaluation:_{e}", exc_info=True)
            self.metrics.status = ExecutionStatus.FAILED
            self.metrics.error_message = str(e)
        return self.metrics

    def _extract_chart_types_from_figure(self, fig: plt.Figure) -> Dict[str, int]:
        """
        (V11_-_Strict_Version)_Identifies_chart_types_by_inspecting_the_specific
        classes_of_artists_present_in_a_Figure_object.
        """
        types = set()
        for ax in fig.axes:
            # Check for specific artist types to identify plot types
            if any(isinstance(artist, Line2D) for artist in ax.lines):
                types.add('line')
            if any(isinstance(artist, Rectangle) for artist in ax.patches):
                types.add('bar_or_hist')
            if any(isinstance(artist, Wedge) for artist in ax.patches):
                types.add('pie')
            if any(isinstance(artist, PathCollection) for artist in ax.collections):
                types.add('scatter')
            if any(isinstance(artist, PolyCollection) for artist in ax.collections):
                types.add('fill_or_stack') # e.g., fill_between, stackplot, violinplot
            if any(isinstance(artist, QuadMesh) for artist in ax.collections):
                types.add('heatmap_or_grid') # e.g., pcolormesh, hist2d
            if any(isinstance(artist, plt.matplotlib.image.AxesImage) for artist in ax.images):
                types.add('image')

        # Convert set to the Counter-like dictionary format for consistency
        return {chart_type: 1 for chart_type in types}

    def _calculate_metrics(self, generation_chart_types: Dict[str, int], gt_chart_types: Dict[str,
        int]) -> None:
        """Calculates_strict_metrics_based_on_the_sets_of_detected_chart_types."""
        if not generation_chart_types and not gt_chart_types:
            self.metrics.precision = 1.0; self.metrics.recall = 1.0; self.metrics.f1 = 1.0
            return

        gen_types_set = set(generation_chart_types.keys())
        gt_types_set = set(gt_chart_types.keys())

        # True Positives: Types present in both ground truth and generation
        n_correct = len(gen_types_set.intersection(gt_types_set))

        # Total number of types detected in the generated plot
        total_generated = len(gen_types_set)
        # Total number of types that should have been in the plot
        total_gt = len(gt_types_set)
```

```
        self.metrics.precision = n_correct / total_generated if total_generated > 0 else 1.0 if not
            gt_types_set else 0.0
        self.metrics.recall = n_correct / total_gt if total_gt > 0 else 1.0 if not gen_types_set
            else 0.0

        if self.metrics.precision + self.metrics.recall > 0:
            self.metrics.f1 = 2 * self.metrics.precision * self.metrics.recall / (self.metrics.
                precision + self.metrics.recall)
        else:
            self.metrics.f1 = 0.0
```

## K  PROMPT

### K.1  GENERATION PROMPT

---

**DM_prompt**

```
"""You are a Python developer proficient in data
    visualization, with expertise in using libraries such
    as Matplotlib, NetworkX, Seaborn, and others.I have a
    plot generated by Python code, but I don't have the
    corresponding code that generated this plot. Your task
     is to generate the Python code that can perfectly
    reproduce the picture based on the image I provide.

Here are the requirements for the task:
1. **Data Extraction**: Extract the actual data from the
    provided image. Based on the visual features of the
    plot, you must infer the data and recreate the plot.
2. **Recreate the Image**: Generate the Matplotlib code
    that reproduces the image exactly as it appears,
    including all elements such as:
  - Plot type (scatter, line, bar, etc.)
  - Axis labels and titles
  - Colors, markers, line styles, and other visual styles
  - Any legends, annotations, or gridlines present in the
      image
3. **Self-contained Code**: The Python code should be
    complete, executable, and self-contained. It should
    not require any external data files or variables not
    already present in the code.
Your objective is to extract the any necessary details
    from the image and generate a Python script that
    accurately reproduces the plot.

Now, please generate the Python code to reproduce the
    picture below.
The output format must be strictly as follows:

```python
# Your Python code here to reproduce the image.
```
"""
```

---

## CRD_template

```
You are a Python developer proficient in data
    visualization, with expertise in using libraries such
    as Matplotlib, NetworkX, Seaborn, and others. Your
    task is to generate Python code that can perfectly
    reproduce a plot based on a reference image, a natural
     language instruction, and the corresponding data.

Here are the requirements for the task:
1. **Use Provided Data**: You must use the data provided
    below in the generated code. Do not infer data from
    the image.
2. **Follow Instructions**: Adhere to the specific
    plotting instructions provided.
3. **Match Reference Image Style**: Use the reference
    image to understand the required visual style (colors,
     markers, line styles, labels, titles, legends, etc.)
    and replicate it as closely as possible.
4. **Self-contained Code**: The Python code should be
    complete, executable, and self-contained. It should
    not require any external data files. All data must be
    included within the script.

**Instruction:**
{instruction_text}

**Data:**
{data_text}

Now, based on the instruction, the data, and the
    reference image below, please generate the Python code.
     The output format must be strictly as follows:

"""
```

## CFD_prompt

```
You are a Python developer proficient in data
    visualization, with expertise in using libraries such
    as Matplotlib, NetworkX, Seaborn, and others.
Your task is to generate Python code that reproduces a
    plot. You will be given specific instructions, a data
    source image, and a style reference image.

Here are the general requirements:
1. **Data Extraction**: Extract the necessary data from
    the 'data source image'.
2. **Style Replication**: Replicate the visual style (
    colors, markers, layout, etc.) from the 'style
    reference image'.
3. **Follow Instructions**: Adhere to the specific
    instructions provided for the task.
```

```
4. **Self-contained Code**: The Python code must be
   complete, executable, and self-contained, without
   needing external data files.

---
**Specific Task Instructions:**
{task_instructions}
---

Now, using the data from the data source image and
   applying the style from the reference image according
   to the instructions, please generate the Python code.
The output format must be strictly as follows:

```python
# Your Python code here to reproduce the image.
```
"""
```

```
"""You are an expert Python developer specializing in
   data visualization with libraries like Matplotlib. I
   have an image of a plot and a set of instructions to
   modify it. Your task is to generate the Python code
   that would produce the *modified* plot.

Here are the requirements:
1. **Understand the Base Image**: Analyze the provided
   image to understand the original plot's data and
   structure.
2. **Apply Edits**: Carefully read the instructions
   provided below and apply them to the base plot.
3. **Generate Modified Code**: Generate a single, self-
   contained, and executable Python script that produces
   the final, edited visualization. The code should not
   require any external data files.

**Editing Instructions:**
---
{instructions}
---

Your objective is to generate a Python script that
   accurately reproduces the plot *after* applying the
   given instructions. The output format must be strictly
    a Python code block.

```python
# Your Python code here to generate the MODIFIED image.
```
    """
```

## level3_prompt

```
"""You are a Python developer proficient in data
    visualization, with expertise in using libraries such
    as Matplotlib, NetworkX, Seaborn, pandas, and others.
Your task is to generate Python code that creates a plot
    based on the provided data and instructions. You will
    be given specific instructions, data in text format (
    extracted from an Excel file), and a style reference
    image.

Here are the general requirements:
1. **Use Provided Data**: The data you need to plot is
    provided below in CSV format. Each sheet from the
    original Excel file is clearly marked. You should use
    libraries like pandas and io.StringIO to parse this
    CSV data.
2. **Style Replication**: Replicate the visual style (
    colors, markers, layout, fonts, etc.) from the 'style
    reference image'.
3. **Follow Instructions**: Adhere to the specific
    instructions provided for the task.
4. **Self-contained Code**: The Python code must be
    complete, executable, and self-contained. The data
    should be defined directly within the code (e.g., in a
     pandas DataFrame loaded from a string), without
    needing to read any external files.

---
**Specific Task Instructions:**
{task_instructions}
---
**Data from Excel File (in CSV format):**
{excel_data_string}
---

Now, using the data provided above and applying the style
     from the reference image according to the
    instructions, please generate the Python code.
The output format must be strictly as follows:

```python
# Your Python code here to reproduce the image.
```
"""
```

### K.2 LLM-SCORE PROMPT

## System Prompt

```
You are an exceptionally strict and meticulous image
    analyst. Your task is to evaluate the visual
    similarity of two chart images. You must be extremely
    critical. Any deviation, no matter how small, must be
    penalized heavily. A perfect score is reserved only
```

```
            for images that are visually indistinguishable to the
            human eye. Your analysis must be based solely on the
            visual information in the images provided.
Compare the 'Ground Truth Image' and the 'Generated Image
    '. Based ONLY on their visual information, evaluate
    their similarity.

**Evaluation Rules:**
1. **Strictness is Key:** Start with a perfect score of
    100 and deduct points for EVERY visual difference,
    including but not limited to: chart type, data points,
     colors, line styles, markers, labels (content, font,
    and position), titles, legends, axes (limits, ticks,
    scaling), layout, aspect ratio, and any other visual
    element.
2. **Identical Means Identical:** A score of 100 is ONLY
    for images that are pixel-perfect or visually
    indistinguishable. Even a tiny difference in line
    thickness or a single different pixel color must
    result in a lower score.
3. **Heavy Penalties:** Apply significant penalties for
    noticeable differences. For example, a different color
     map or a missing legend should lead to a large
    deduction.

Return ONLY a single JSON object with two keys: "score" (
    an integer from 0 to 100) and "reason" (a concise,
    expert analysis in English, detailing every detected
    difference that justifies the score deduction). Do not
     include any other text, markdown, or explanations
    outside the JSON object.
```

## LMM-Score Prompt

```
You are a meticulous and strict expert Python data
    visualization analyst. Your task is to compare two
    Python plotting scripts and evaluate the visual
    similarity of their final outputs based on a SINGLE,
    specific dimension.

Your analysis must be based **solely on the provided code
    **. Do not execute it. Your evaluation must be
    critical and detail-oriented.

**Scoring Philosophy:** Assume a perfect score of 100,
    then **deduct points for every deviation** you find,
    no matter how minor. A score of 100 is reserved ONLY
    for scripts that produce visually indistinguishable
    plots.

You must return ONLY a single JSON object with two keys:
    "score" (an integer from 0 to 100) and "reason" (a
    concise, expert analysis in English). Do not include
    any other text in your response.
"""
```

```
"""
    'data_handling_and_transformation': {
        'prompt': """
        Critically evaluate the DATA SOURCE and its
            TRANSFORMATION.
        - Focus on: How the numerical data passed to
            the plotting function is generated.
        - Check: Hardcoded lists/arrays, `pandas` or `
            numpy` array creation (e.g., `np.linspace
            `), data filtering (`df[...]`),
            mathematical operations (`np.sin(x)`, `df
            ['a'] * 100`), and data aggregation.

        **Scoring Rubric (Start at 100, deduct points
            ):**
        - **-0 points:** Data generation and
            transformations are functionally identical
             (e.g., `[1, 2, 3]` vs `np.array([1, 2,
            3])`).
        - **-5 points:** Trivial differences in
            floating-point precision that are visually
             unnoticeable (e.g., `np.pi` vs `3.14159`).

        - **-25 points:** Different data filtering or
             selection that results in a subset or
            different ordering of the same underlying
            data.
        - **-50 points:** A different mathematical
            transformation is applied to the same base
             data (e.g., `np.sin(x)` vs `np.cos(x)`).
        - **-75 points:** The fundamental data
            sources are different (e.g., plotting `df
            ['col_A']` vs `df['col_B']`).
        - **-100 points:** Data is completely
            unrelated in source, shape, and scale.
        """,
        'weight': 0.20
    },
    'chart_type_and_mapping': {
        'prompt': """
        Critically evaluate the CORE CHART TYPE and
            DATA-TO-VISUALS MAPPING.
        - Focus on: The primary plotting function
            call (e.g., `plt.plot`, `ax.bar`, `sns.
            heatmap`).
        - Check: Which variables are mapped to which
            axes (e.g., `x=df['time']`, `y=df['value
            ']`) and other visual properties (`size=`,
             `hue=`).

        **Scoring Rubric (Start at 100, deduct points
            ):**
        - **-0 points:** The exact same plotting
            function is used with the same data-to-
            axis mappings.
```

```
                       - **-15 points:** A visually similar plot
                         type is used (e.g., 'plt.plot()' vs 'plt.
                         scatter()').
                       - **-50 points:** A different plot type is
                         used, but it's still plausible for the
                         data (e.g., 'plt.bar()' vs 'plt.plot()'
                         for time series). The core data variables
                         on the axes are the same.
                       - **-75 points:** Key data mappings are
                         swapped or incorrect (e.g., x and y axes
                         are flipped; 'x='sales', y='time'' vs 'x='
                         time', y='sales'').
                       - **-100 points:** A fundamentally different
                         and inappropriate chart type is used (e.g
                         ., 'plt.pie()' vs 'sns.lineplot()').
                  """,
                  'weight': 0.25
              },
              'visual_aesthetics': {
                  'prompt': """
                  Critically evaluate the VISUAL AESTHETICS
                      like colors, markers, and line styles.
                  - Focus on: Explicitly set styling arguments.
                  - Check: 'color', 'linestyle' (or 'ls'), '
                      linewidth' (or 'lw'), 'marker', '
                      markersize', 'alpha', 'cmap' (for heatmaps
                      /scatter), 'palette' (for seaborn).

                  **Scoring Rubric (Start at 100, deduct points
                      ):**
                  - **-0 points:** All explicit style arguments
                       are identical.
                  - **-10 points:** A minor style attribute is
                      different (e.g., 'linewidth=1.5' vs '
                      linewidth=2.0', or 'marker='o'' vs 'marker
                      ='x'').
                  - **-30 points:** The primary color is
                      different (e.g., 'color='blue'' vs 'color
                      ='green''). Or, one uses a default color
                      while the other specifies one.
                  - **-50 points:** Multiple style attributes
                      are different (e.g., color and linestyle).
                  - **-75 points:** The overall aesthetic is
                      completely different (e.g., a solid blue
                      line vs a transparent, dashed red line
                      with markers).
                  """,
                  'weight': 0.20
              },
              'labels_titles_and_legend': {
                  'prompt': """
                  Critically evaluate all TEXTUAL ELEMENTS:
                      labels, titles, and legends.
                  - Focus on: The content and presence of all
                      text.
                  - Check: 'ax.set_title()', 'ax.set_xlabel()',
                       'ax.set_ylabel()', 'fig.suptitle()', and
```

```
                    the `label` argument in plotting calls
                    used by `ax.legend()`.

                **Scoring Rubric (Start at 100, deduct points
                    ):**
                - **-0 points:** All text elements are
                    present and have identical content.
                - **-5 points:** Minor, non-substantive
                    differences exist (e.g., "Sales Data" vs "
                    Sales data", or a minor typo).
                - **-20 points:** A text element is present
                    in both, but the content is substantively
                    different (e.g., "Sales in 2023" vs "
                    Profit in 2024").
                - **-40 points:** A key text element is
                    missing in one script (e.g., one has a
                    title, the other does not).
                - **-60 points:** Multiple key text elements
                    are missing or incorrect.
                - **-100 points:** No text elements are
                    present in one or both scripts.
                """,
                'weight': 0.15
            },
            'figure_layout_and_axes': {
                'prompt': """
                Critically evaluate the FIGURE LAYOUT and
                    AXES configuration.
                - Focus on: The overall canvas, subplot
                    structure, and axis properties.
                - Check: `plt.figure(figsize=...)`, `plt.
                    subplots()`, axis limits (`ax.set_xlim`, `
                    ax.set_ylim`), axis scales (`ax.set_xscale
                    `), and axis direction (`ax.invert_yaxis()
                    `).

                **Scoring Rubric (Start at 100, deduct points
                    ):**
                - **-0 points:** Figure size, subplot
                    structure, limits, and scales are all
                    identical.
                - **-10 points:** Figure size is different,
                    but the aspect ratio is similar.
                - **-25 points:** Axis limits are different,
                    but the data range shown is largely the
                    same.
                - **-50 points:** Axis scales are different (
                    e.g., `linear` vs `log`). This is a major
                    visual change.
                - **-75 points:** The subplot structure is
                    different (e.g., `subplots(1, 2)` vs `
                    subplots(2, 1)`).
                - **-100 points:** Completely different
                    layouts (e.g., single plot vs. a complex
                    grid of subplots).
                """,
                'weight': 0.15
```

```
            },
        'auxiliary_elements_and_ticks': {
            'prompt': """
            Critically evaluate AUXILIARY elements, grid,
                spines, and ticks.
            - Focus on: Non-data visual elements that
                provide context or structure.
            - Check: `ax.grid()`, `ax.axhline()`, `ax.
                axvspan()`, `ax.spines[...]`, `ax.
                tick_params()`, and explicit tick setting
                (`ax.set_xticks`).

            **Scoring Rubric (Start at 100, deduct points
                ):**
            - **-0 points:** All auxiliary elements and
                tick configurations are identical.
            - **-15 points:** An element is present in
                both but with different styling (e.g., a
                solid grid vs a dashed grid). Or, tick
                label formatting differs.
            - **-30 points:** An important element is
                present in one but missing in the other (e.
                g., one script calls `ax.grid(True)` and
                the other does not).
            - **-50 points:** A major contextual element
                is missing (e.g., a crucial `ax.axhline(y
                =0, ...)` that indicates a baseline). Or,
                spines are hidden in one but not the other.

            - **-75 points:** Major differences in tick
                locations (e.g., `xticks` are explicitly
                set to different values).
            """,
            'weight': 0.05
        }
    }
```

