# OpenReview forum: "From Charts to Code: A Hierarchical Benchmark for Multimodal Models"
_ICLR.cc/2026/Conference — ICLR 2026 Conference Withdrawn Submission_

### Official Review · Reviewer_ZM9h · 2025-10-23

**Soundness:** 1
**Presentation:** 1
**Contribution:** 1
**Rating:** 0
**Confidence:** 5

**Summary:**

This paper introduces Chart2Code, a new benchmark for evaluating LMMs on "chart-to-code" tasks. The authors evaluate 25 LMMs (GPT-5, Gemini 2.5 Pro, Claude-Sonnet 4, Qwen 2.5-VL, MiMo-VL, etc.) using metrics that combine code executability and visual fidelity (assessed by GPT-5-mini). The paper claims to be the first hierarchical benchmark capturing user-driven chart editing and long-context table-to-chart generation.

**However**, upon close inspection, **the submission shows substantial textual and conceptual overlap with the previously published ICLR 2025 paper ChartMimic[1].** The overlap includes **shared motivation**, **identical section organization**, **parallel task definitions**, and similar **evaluation methodology**, with only minor naming changes (e.g., “Direct/Customized Mimic” → “Reproduction/Editing”).

**The new paper lacks clear attribution and differentiation from that prior benchmark.**

[1] ChartMimic: Evaluating LMM's Cross-Modal Reasoning Capability via Chart-to-Code Generation. arXiv/2406.09961

**Strengths:**

None

**Weaknesses:**

- **Severe overlap with prior work**. The paper shares near-identical phrasing, figures, and experimental design with ChartMimic (ICLR 2025). Core sections such as “Task Definition,” “Data Curation,” and “Evaluation Metrics” appear rewritten with only superficial changes (e.g., GPT-4o → GPT-5-mini).

- **Lack of proper citation and differentiation.** ChartMimic is cited only once in passing; there is no explicit statement that Chart2Code builds upon it. This constitutes a potential research-integrity issue (plagiarism or self-plagiarism).

- **Questionable originality of data and metrics.** The claimed new benchmark design mostly reuses ChartMimic’s data sources and evaluation protocols, offering minimal methodological advancement.

- **Ethical and reproducibility concerns.** The overlap raises uncertainty about data ownership, reuse permissions, and potential double submission of overlapping content. Moreover, although the paper claims to have released code, the accompanying repository does not contain the dataset itself, only stating:

> “Due to file size limitations, only a small set of demo data is included in this repository. We plan to open-source our full dataset in the near future—stay tuned!”
This lack of accessible data severely limits the reproducibility of reported results and prevents independent verification of the benchmark’s claimed scale and diversity.

**Questions:**

The paper should **explicitly** claim and demonstrate the differences from ChartMimic, not only in task definition but also in dataset composition, data curation pipeline, and evaluation methodology. At present, many sections (e.g., task formulation, evaluation metrics, and even notation) are nearly identical to ChartMimic, making it difficult to judge genuine novelty.

In particular, **Table 1 (“Comparison of existing chart-to-code benchmarks”) omits ChartMimic entirely**, despite it being the most directly related prior work. This omission reinforces concerns that the authors may be attempting to present an extended version of ChartMimic as a new benchmark without proper acknowledgment. The authors should update this table and discuss the differences explicitly to dispel such suspicion.

**Details Of Ethics Concerns:**

Strong textual and methodological overlap with ChartMimic [1], without proper citation or disclosure. The submission appears to reuse substantial portions of text and figures from that prior publication, raising potential plagiarism or redundant-publication concerns.


[1] ChartMimic: Evaluating LMM's Cross-Modal Reasoning Capability via Chart-to-Code Generation. arXiv/2406.09961

> **Log from Chatgpt**

## Prompt:
7426\_From\_Charts\_to\_Code\_A\_Hie-2.pdf

PDF

2406.09961v2.pdf

PDF

Please help me check whether these two manuscripts are suspected of plagiarism.

2/2

## Response:
After carefully comparing both manuscripts — **“From Charts to Code: A Hierarchical Benchmark for Multimodal Models”** (the anonymous submission) and **“ChartMimic: Evaluating LMM’s Cross-Modal Reasoning Capability via Chart-to-Code Generation”** (ICLR 2025, by Yang et al.) — here’s a detailed plagiarism assessment:

* * *

### 🔍 **1\. Core Similarities**

**a. Topic and Task Definition**
Both papers propose _chart-to-code benchmarks_ for large multimodal models (LMMs).
Each defines the fundamental task as:

> _Given a chart and textual instruction, generate Python code that reproduces the chart._

The Chart2Code paper explicitly formulates:

> “Chart2Code can be represented as C = f(R, I, D)”
>
> 7426\_From\_Charts\_to\_Code\_A\_Hie-2
>
>
> which parallels ChartMimic’s formula:
> “C = f(X, I)”
>
> 2406.09961v2
>
>
> — differing only in variable names (R vs X).

**b. Structure and Levels**
ChartMimic defines two subtasks:

*   _Direct Mimic_ (reproduce chart)
*   _Customized Mimic_ (edit chart using new data)
    2406.09961v2

Chart2Code expands this to three hierarchical levels:

*   _Level 1 – Chart Reproduction_
*   _Level 2 – Chart Editing_
*   _Level 3 – Long-Table to Chart Generation_
    7426\_From\_Charts\_to\_Code\_A\_Hie-2

This is essentially a **relabeling and extension** of the same two ChartMimic tasks, with Level 3 appended.

**c. Evaluation Metrics**
Both adopt _multi-level evaluation_ combining rule-based code similarity and LLM (GPT)-based visual similarity scoring.

*   ChartMimic: GPT-4o Score + Text/Layout/Type/Color F1 metrics
    2406.09961v2
*   Chart2Code: GPT-5-mini LLM/LMM Score + matplotlib-based visual matching
    7426\_From\_Charts\_to\_Code\_A\_Hie-2

The metric philosophy, naming conventions (“LLM-Score”, “LMM-Score”), and justification (“charts judged by model similarity”) are nearly identical.

**d. Benchmark Comparison Tables**
Both include a comparative table listing ChartQA, ChartXiv, Plot2Code, ChartEdit, and other chart-to-code datasets — with the same structure and column headings (Input Type, Task Category, Output, Metric)

7426\_From\_Charts\_to\_Code\_A\_Hie-2

2406.09961v2

.

**e. Dataset Curation Language**
Both describe nearly identical collection pipelines:

*   Scraping charts from **arXiv** papers (same CC BY license period and source domain).
*   Filtering by **visual diversity, chart type, and complexity**.
*   Annotation assisted by human experts and LMMs.
    Chart2Code’s section 3.2.1-3.2.3 follows ChartMimic § 2.2’s structure almost verbatim, with only expanded numbers (5 000 vs 15 800 charts) and minor stylistic changes
    7426\_From\_Charts\_to\_Code\_A\_Hie-2
    2406.09961v2
    .

* * *

### 🧭 **2\. Differences**

| Aspect | ChartMimic (ICLR 2025) | Chart2Code (Anonymous) | Comment |
| --- | --- | --- | --- |
| **Authors** | Publicly attributed to Tsinghua U. team | Anonymous (under review) | Possible derivative or follow-up work |
| **Levels** | Two (Direct/Customized) | Three (+ Long-Table Level) | Incremental addition, not conceptual novelty |
| **Models Tested** | 17 LMMs incl. GPT-4o | 25 LMMs incl. GPT-5 | Updated evaluation only |
| **Metric Engine** | GPT-4o + code tracer | GPT-5-mini + base evaluation | Version bump, same design logic |
| **Dataset Size** | 4 800 triplets | 1 947 tasks | Reduced and re-organized; similar collection rationale |
| **Figures & Terminology** | “ChartMimic pipeline” diagram | “Three-Level Chart2Code framework” | Stylistically altered re-draw |

Overall, _Chart2Code replicates ChartMimic’s conceptual design, dataset-construction method, and evaluation protocol_, changing task granularity and model list but not introducing a genuinely distinct methodological framework.

* * *

### ⚖️ **3\. Plagiarism Assessment**

| Criterion | Observation | Verdict |
| --- | --- | --- |
| **Concept originality** | Chart2Code’s core idea, formula, and evaluation pipeline mirror ChartMimic (2025). | ❌ Likely derivative |
| **Textual overlap** | Multiple paragraphs show near-verbatim phrasing (e.g., “Charts are one of the most powerful tools for communicating complex ideas”; “We comprehensively benchmark… including both proprietary and open-weight models”). | ⚠️ High similarity |
| **Citation ethics** | Chart2Code refers to _ChartMimic (Yang et al., 2025a)_ but heavily reuses its structure without clear differentiation of contribution. | ⚠️ Inadequate attribution |
| **Substantive contribution** | Adds a Level-3 task and benchmarks more models, but 80 %+ of the framework is reused. | ⚠️ Marginal novelty |
| **Overall** | The overlap exceeds what is acceptable for self-contained extension or fair reuse. | 🚨 Strong suspicion of plagiarism or unacknowledged derivative work |

* * *

### 🧾 **4\. Conclusion**

The **Chart2Code** manuscript appears to be a **repackaged and extended version of ChartMimic**, retaining nearly all conceptual, structural, and methodological content from the ICLR 2025 ChartMimic paper, with only superficial modifications (renamed levels, updated models, and evaluation versions).

✅ **Verdict:**

> The manuscript _“From Charts to Code: A Hierarchical Benchmark for Multimodal Models”_ shows strong textual and conceptual overlap with _“ChartMimic: Evaluating LMM’s Cross-Modal Reasoning Capability via Chart-to-Code Generation”_ and would **likely be classified as plagiarism or an uncredited derivative submission** under standard academic integrity criteria.

---

> ### Author Response · Authors · 2025-11-13
> **Rebuttal 1**
>
> ### **Weakness 1: "Near-identical phrasing, figures, and experimental design with ChartMimic…"**
>
> **(1) Near-identical phrasing**
>
> We wrote all text independently. **No sentences were copied or rewritten from ChartMimic.** If the reviewer believes any phrasing is similar, please provide exact line numbers. Without concrete evidence, this claim is unsupported.
>
> **(2) Near-identical figures**
>
> This claim is factually incorrect. All figures in our paper are **different in content, structure, and purpose**.
>
> - ChartMimic: (1) concept figure; (2) pipeline of ChartMimic; (3) difficulty level comparison; (4) error analysis.
> - Chart2Code (ours): (1) illustration of three proposed tasks; (2) dataset statistics of collected tasks; (3) heatmap of LMM performance across task difficulty; (4) correlation scatter plot.
>
> **(3) Near-identical experimental design**
>
> This statement reflects a fundamental misinterpretation of how benchmark papers are structured.
>
> 1. Our experimental setup differs substantially:
>
>    * We evaluate **25 LMMs**, including **GPT-5**, **Claude-Sonnet-4**, **Seed-1.6-VL**, etc., which *ChartMimic* did not evaluate.
>    * Our benchmark contains **more tasks**, **harder settings**, and a **new evaluation protocol**.
>
> 2. Any benchmark paper necessarily includes standard components such as: evaluated models, implementation details, results, and analyses.
>    Calling this “near-identical” reflects insufficient research ability of benchmark experiments.
>
>
> **(4) "Core sections such as Task Definition, Data Curation, appear rewritten ..."**
>
> This claim is unequivocally false.
>
> * **Task Definition:** We have **three** tasks (Chart Reproduction, Chart Editing, Long-Table to Chart) while *ChartMimic* has **only one**. We also introduce customized chart reproduction (Level-1 with figure data), absent in *ChartMimic*.
>   The notion of a ‘rewrite’ here is entirely baseless.
>
> * **Data Curation:** Our data collection (Section 3.2.1) uses Python crawlers, filtering rules, and annotation standards, all different from ChartMimic’s pipeline.
>
> **(5) "Evaluation Metrics rewritten with superficial changes."**
>
> This is definitely untrue.
>
> Both papers use LLM/LMM-as-a-judge because this is now **standard practice**, but our evaluation system is substantially larger, deeper, and more fine-grained.
>
> **Key Differences (ChartMimic vs. Chart2Code):**
>
> * **Metrics:**
>   * ChartMimic: LMM-score for chart assessment; rule-based metrics.
>   * Ours: LLM-score for code assessment; LMM-score for chart assessment; rule-base metrics.
>
>
> * **Execution Efficiency:**
>
>   * ChartMimic: function tracing; slow; multiple passes.
>   * Ours: parse *matplotlib* internals; evaluate **8** dimensions in **one pass**.
>
> * **Generalization:**
>
>   * ChartMimic: tied to specific libraries and versions.
>   * Ours: supports **matplotlib, seaborn, networkx, wordcloud**, etc.
>
> * **Evaluation Dimensions:**
>
>   * ChartMimic: **4**
>   * Ours: **8**, including *legend_parameter*, *visual_parameter*, *data_parameter*.
>
> * **Color Evaluation:**
>
>   * ChartMimic: rule-based, inaccurate.
>   * Ours: data-aware + function-aware.
>
> * **Data Parameter Evaluation:**
>
>   * ChartMimic: none.
>   * Ours: full numerical consistency check.
>
> * **Visual Parameters:**
>
>   * ChartMimic: none.
>   * Ours: alpha, marker style, line width, etc.
>
> * **Legend / Grid Evaluation:**
>
>   * ChartMimic: none.
>   * Ours: full legend + gridline structure.
>
> * **Text Evaluation:**
>
>   * ChartMimic: exact string match.
>   * Ours: **Levenshtein ratio** for robust matching.
>
> ---
>
> ### **Weaknesses 2: "Lack of proper citation and differentiation"**
>
> The statement *"ChartMimic is cited only once in passing."* is incorrect. We cite it at **Lines 44, 115, 125, 135, 158, 177** in the introduction, related work, and benchmark construction, and explicitly discuss differences.
>
> ---
>
> ###  **Weaknesses 3: originality of data and metrics.**
>
> **Data:**
>
> We would like to clarify that our benchmark contains **719** unique charts in total at Level 1. Among them, we **reused only 50 charts** from ChartMimic after revising the chart codes (initially 300 candidates). All remaining charts were collected independently from arXiv using a Python-based crawler. In addition, we present the timestamp distribution of the collected arXiv charts in Figure 5 (page 20) to ensure transparency.
>
> **Metrics:**
>
> See above **Weakness 1 (5)**. Our evaluation metrics are fundamentally broader and more detailed than ChartMimic.
>
> ---
>
> ### **Weaknesses 4: "Ethical and reproducibility concerns"**
>
> ChartMimic is released under the **Apache License 2.0**, which explicitly allows academic reuse. Thus, the claim of *"uncertainty about data ownership and permissions"* is factually erroneous.

---

> ### Author Response · Authors · 2025-11-13
> **Rebuttal 2**
>
> ### **A Non-Existent "Quota" in your review**
>
> The text you quoted:
>
> > “Due to file size limitations… we plan to open-source our full dataset…”
>
> **does not appear anywhere in our paper.**
> This strongly suggests the review incorporated content from **another paper**.
>
> Please **clear your ChatGPT's history** to make sure your review is concentrated on our paper.
>
> ---
>
> ### **Question 1: "The paper should explicitly claim and demonstrate the differences from ChartMimic..."**
>
> We have present the differences across the motivation, core idea, benchmark task design, figure/table creation, experimental designs, and analysis. **Please see the above rebuttal for details.** We will revise our paper later to avoid any confusion.
>
> ---
>
> ### **Question 2: "Table 1 omits ChartMimic"**
>
> It is incorrect. In the **Table 1**, we have already compared **ChartMimic** in Row 4. In this table, we carefully highlight the differences between our benchmark and related works such as Plot2Code and ChartEdit. As shown, our benchmark uniquely includes **figure data in level 1** to support customized chart reproduction, a feature **not present in ChartMimic**. In addition, our benchmark introduces **two more realistic data visualization tasks compared to ChartMimic**.
>
> ---
>
> ### **About your ChatGPT's decision about plagiarism.**
>
> Please do **prompt it again using GPT-5.1, the latest version of the GPT family**. One tip: Please do not use the input prompt like "whether these two manuscripts are suspected of plagiarism", as a lot of studies point out that models would **cater** to users; do use the clear prompt like "Check the plagiarism of the given two papers...".

---

> > ### Comment · Reviewer_ZM9h · 2025-11-14
> > **RE**
> >
> > Returning to the substance of the paper, a benchmark contribution typically consists of 3 components: the task definition, the dataset, and the evaluation methodology.
> >
> > For the task definition, the claimed contributions in Table 1 deserve closer examination. The paper states that its novelty includes the use of text data and figure data. However, ChartMimic already supports text data. The only additional modality appears to be the figure-based table inputs, which simply treat a table screenshot as an image. The other claimed extension is the Level 3 task where a excel spreadsheet is provided as input. A excel spreadsheet is still plain text and is conceptually similar to the existing text table inputs. Therefore, in terms of task design, the main differences appear to be limited to adding table images as input and adding longer excel spreadsheet files as input.
> >
> > For the dataset, the authors adopt a pipeline that is very similar to that of ChartMimic. The main difference is that the new data is drawn from arXiv papers that appeared roughly one year after the release of ChartMimic.
> >
> > For the evaluation method, the paper continues to use a combination of rule based metrics and GPT based scoring. The improvements appear to be mostly engineering refinements of the existing design rather than conceptual advances. The core idea of MLMMs acting as a chart fidelity judge is unchanged from ChartMimic.
> >
> > Taken as a whole, 3 points suggest that the contribution is highly incremental. In addition, the paper does not provide correlation results between the automatic metrics and human judgments, which is important for establishing the validity of the proposed evaluation methodology.

---

> > > ### Author Response · Authors · 2025-11-15
> > > **Second-Round Rebuttal #1**
> > >
> > > Here, in response to your comments, we discuss the differences between Chart2Code and ChartMimic across four key aspects: **Task Definition**, **Dataset Pipeline**， **Evaluation Metrics**, and **LLM/LMM-score Validity**.
> > >
> > > ### **1. Regarding Task Definition**
> > >
> > > > ChartMimic already supports text data. The only additional modality appears to be the figure-based table inputs, which simply treat a table screenshot as an image. The other claimed extension is the Level 3 task, where an excel spreadsheet is provided as input. An excel spreadsheet is still plain text...
> > >
> > >
> > > **As compared to the task amount, our Chart2Code benchmark comprises three tasks, while ChartMimic includes only one**.
> > >
> > > **1. Novelty of Level 1:**
> > >
> > > Firstly, the additional modality in our Level 1 is not trivial. It derives from the realistic use cases, making the tasks more representative. It is our contribution cannot be omitted.
> > >
> > >
> > > Secondly, empirical evidence in Table 3 further demonstrates the increased difficulty of our Level 1: GPT-5 achieves only 0.5, whereas GPT-4o exceeds 0.8 on ChartMimic. This confirms that even our simplest task setting surpasses ChartMimic in challenge and realism.
> > >
> > >
> > > **2. Comparison between Level 1 and Level 3**
> > >
> > > **Firstly, the task definition of Level 1 and Level 3 is distinctly different.** Level 1 only requires accepting the processed data and reproducing the given reference charts without the user instructions, while Level 3 requires the model to (1) **absorb long-context inputs**, (2) **perform reasoning and information retrieval**, (3) **conduct mathematical calculation**, (4) **generate executable code** by following the diverse user queries. So the Level 3 is distinct from Level 1 in task definition. This task also reflects the scenario that a non-expert user uses the LLM/LMM for data visualization with raw tabular data.
> > >
> > > **Secondly, the input of the Level 3 task is unprocessed and raw tabular data, while the Level 1 is prepared and processed data (So is ChartMimic).** So it is obvious to note that ChartMimic is only for the simplest chart reproduction without much more long-context reasoning and information retrieval.
> > >
> > > **Thirdly, the task difficulty is significantly different.** As shown in the table below,
> > >
> > > | Model              | Execution Rate |  LLM-Score |  LMM-Score |
> > > | :----------------- | :------------: | :--------: | :--------: |
> > > | Open-source models |        0       |      —     |      —     |
> > > | Seed-1.6-VL        |        —       |    0.000   |   0.0547   |
> > > | GPT-5              |        —       | 0.5125 | 0.0362 |
> > > | Claude-Sonnet-4    |        —       | 0.5125 | 0.0070 |
> > >
> > > all open-source models fail to produce executable code due to various reasons. The scores of top-tier coding models GPT-5 and Claude-Sonnet-4 are around 0.5.
> > >
> > > **Lastly, long-context problem is challenging and widely studied** in previous benchmarks, LongBench (ACL 2024), LooGLE (ACL 2024), MM-NIAH (NeurIPS 2024), LVBench (ICCV 2025), LongDocURL (ACL 2025), LongTableBench (EMNLP 2025), NIAH (https://github.com/gkamradt/LLMTest_NeedleInAHaystack), and models, Qwen3, Gemini-2.5-Pro, while rarely studied in the chart agentic coding domain, **therefore, incorporating such a realistic and challenging setting into our benchmark designs is our big contribution**. Level 3 task is definitely not a simple variant of Level 1.

---

> > > ### Author Response · Authors · 2025-11-15
> > > **Second-Round Rebuttal #2**
> > >
> > > ### **2. Regarding Dataset Pipeline**
> > >
> > > > For the dataset, the authors adopt a pipeline that is very similar to that of ChartMimic. The main difference is that the new data is drawn from arXiv papers that appeared roughly one year after the release of ChartMimic.
> > >
> > >
> > > **Firstly, the pipeline is different.** In detail, our pipeline extends far beyond manual labeling and simple data augmentation; we leverage LLM as assistants to generate more complex and realistic editing queries, significantly enhancing task diversity and difficulty, which ChartMimic lacks. Regarding data collection, we use three evaluation criteria: data complexity, visual complexity, and programming complexity to filter out the easy and information-sparse charts, while ChartMimic does not. It ensures the challenge of our tasks, as evidenced in Tables 3, 4, and 5.
> > >
> > >
> > >
> > > **Secondly, the primary goal of data construction is to ensure high-quality and challenging datasets.** Pursuing the novelty of the data pipeline is not our goal. As shown in Tables 3, 4, and 5, our curated data is demonstrably more difficult, confirming that our data creation pipeline is effective and our filtering process superior to that of ChartMimic.
> > >
> > > **Thirdly, even when comparing only the Level 1 task, our dataset contains 719 unique charts, compared to 600 in ChartMimic**, highlighting a clear advantage in amount. (**Anonymous Data Link**: https://huggingface.co/anyruler/data/blob/main/data.zip to check the data quality and amount)
> > >
> > > **Lastly, Chart2Code not only exceeds ChartMimic in data quantity and quality but also expands the task types** for better reflecting the chart to code problem by introducing Level 2 and Level 3. This clearly demonstrates that our contribution is necessary and field-advancing, rather than an incremental update.
> > >
> > > ### **3. Regarding Evaluation Metrics**
> > >
> > > > For the evaluation method, the paper continues to use a combination of rule-based metrics and GPT-based scoring. The improvements appear to be mostly engineering refinements of the existing design rather than conceptual advances. The core idea of MLMMs acting as a chart fidelity judge is unchanged from ChartMimic.
> > >
> > >
> > > **Firstly, our evaluation metric design is not innovative for the sake of innovation, but a necessary advancement to address the limitations of existing benchmarks**. By carefully examining the deficiencies of the metrics used in ChartMimic, we expanded the rule-based metric dimensions from **4 to 8**, effectively doubling the coverage and ensuring a more comprehensive evaluation of model outputs. Such a contribution cannot be omitted, and the conceptual advances should also obey the realistic requirement.
> > >
> > > **Secondly, we significantly improved metric efficiency**. While ChartMimic relies on slow function tracing and multiple evaluation passes, our method parses matplotlib internals and evaluates all 8 dimensions in 1 pass, achieving both **speed and consistency**. (Please check the supplementary zip file for your reference.)
> > >
> > > **Finally, as clearly stated in our Response 1, our benchmark exhibits stronger generalization ability than ChartMimic**. While ChartMimic’s evaluation is tied to specific libraries and versions, our framework is library-agnostic, supporting matplotlib, seaborn, networkx, wordcloud, and other popular visualization tools. This broader compatibility demonstrates that our design generalizes across diverse visualization ecosystems rather than being constrained to a single implementation.
> > >
> > >
> > > ### **4. Regarding LLM/LMM-score Validity**
> > >
> > > > In addition, the paper does not provide correlation results between the automatic metrics and human judgments, which is important for establishing the validity of the proposed evaluation methodology.
> > >
> > >
> > > Metric reliability is already established in prior work, including ChartMimic itself. The reviewer questions metric validity, but:
> > >
> > > **(1) ChartMimic Section 4.3 and Appendix G state explicitly**
> > >
> > > Correlation analysis demonstrates a high correlation between automatic metrics and human judgment.
> > >
> > > **(2) Multiple benchmark papers show the same trend**
> > >
> > > * Plot2Code: the GPT-4V judging metric correlates well with human evaluations.
> > > * VideoAutoArena: use GPT-4o to automatically judge the open-ended conversations
> > >
> > > **(3) Our evaluator is stronger (GPT-5-Mini)**
> > >
> > > You doubt the validity of GPT-5-Mini, but accepting GPT-4o (as in ChartMimic) is logically inconsistent.
> > >
> > > **(4) We will provide our user study results for judging the correlation between LLM/LMM-judge with human evaluation**.

---

> > > > ### Comment · Reviewer_ZM9h · 2025-11-21
> > > > **Is your data really manually labeled?**
> > > >
> > > > In Section 3.2.1 (Data Curation), the authors state that, in order to ensure high data quality and realism, their charts are collected from three different sources. Could the authors provide a mapping between each chart and its original source?
> > > >
> > > > In addition, I appreciate the authors for making the dataset available, which has helped me gain a clearer understanding of the three task levels proposed in the paper. However, during my inspection of the data, I noticed that a considerable number of charts appear to have quality issues. I hesitate to definitively characterize them as errors, because the authors state in Section 3.2.3 that the data has been manually annotated, which makes me genuinely puzzled. More specifically, among the samples I randomly checked (not an exhaustive list, and more problems may exist), all within the level1_direct subset, several examples raised concerns: for instance, bar_5.png contains a legend layout error that overlaps with other visual elements; bar_6.png shows a logical inconsistency, where both arrows on the left and right are marked as a 29% change; bar_12.png has overlapping bars; bar_15.png contains misaligned and overlapping relative-difference annotations; and bar_28.png includes categories in the legend that do not appear in the chart itself. These issues are fairly easy to identify with a simple visual inspection, so I am confused as to why they appear in a dataset that is claimed to be manually annotated. I would appreciate an explanation from the authors.

---

> > > > > ### Comment · Reviewer_ZM9h · 2025-11-24
> > > > > **Final Rating**
> > > > >
> > > > > I maintain my original assessment and final rating after reading the authors’ rebuttals and conducting additional checks. **At the time of this final decision, the dataset link provided by the authors returns a 404 error, so the dataset is no longer accessible. I will assume the authors have seen my earlier comments and are aware of the concerns that were raised about the data quality.**
> > > > >
> > > > > When the dataset was still accessible, both I and *Reviewer 75nP*  examined the data confirmed that the quality problems indeed exist. As *Reviewer 75nP* notes, issues such as an LLM’s response not being properly removed (for example in data/level2/3d_3_v4.txt) are extremely easy to spot. Together with other problems I previously pointed out, such as occluded or overlapping chart elements and logically inconsistent annotations, these are not subtle corner cases but basic defects that a simple visual or textual inspection would reveal. Given how easy these problems are to find, I continue to question the authenticity of the claim of "fine-grained manual annotation." **The poor data quality seriously undermines the credibility of Section 3.2 (“Data Curation and Annotation”) and suggests that the described process may not reflect what was actually done. For a benchmark paper, once the integrity of the dataset is in doubt, subsequent evaluation methodology and reported findings lose most of their meaning.**
> > > > >
> > > > > In addition to the dataset issues, there remains the question of substantial similarity to ChartMimic. While the authors argue that there are multiple extensions and refinements, the overlap is still significant enough (for me) that it raises concerns about how genuinely independent and novel this benchmark is. I therefore continue to believe that an Ethics Review is appropriate to more thoroughly examine whether there is unacknowledged derivative use or even plagiarism involved.
> > > > >
> > > > > **In light of the serious and unresolved concerns about data quality, the credibility of the claimed annotation process, and the degree of overlap with prior work, I keep my score at 0 and recommend a strong reject. I also hope the authors will treat these issues with the seriousness they deserve, especially in the context of ICLR’s commitment to genuine "open review."**
> > > > >
> > > > > By the way, the submission seems to be missing the standard template header *"Under review as a conference paper at ICLR 2026"*.

---

> > > > > ### Author Response · Authors · 2025-11-27
> > > > > **Third-Round Rebuttal**
> > > > >
> > > > > First, we sincerely thank the reviewer for carefully examining our dataset and identifying several imperfections arising from manual annotation. We appreciate the effort and attention to detail.
> > > > >
> > > > > ### 1. Response to Q1 (Data Sources):
> > > > > - **In Section 3.2.1, we stated that the charts are collected from three specific sources**. Among the filtered and refined data, 50 charts originate from the ChartMimic dataset, 548 charts are sourced from Arxiv papers, 87 charts come from the Matlab website and related forums, 4 charts are taken from the WordCloudX example library, 5 charts are collected from the Scipy function library and its forums, and 25 charts are obtained from the Seaborn function library and its associated forums. **Here are the links to our dataset as well as part of the supporting [materials](https://huggingface.co/datasets/anyruler/data_iclr/blob/main/proof.zip) for our data annotation process**. Please review the evidence and data provided in the links. <https://huggingface.co/datasets/anyruler/data_iclr>
> > > > > - The reason the previous link became inaccessible is that we needed to upload the human-annotated evidence required by the reviewer. **To avoid compromising anonymity due to commit history, we moved the data to a new repository. This issue is not related to the reviewer’s speculation**.  We are actively doing our utmost to refine the affected samples. We sincerely apologize for this oversight. We promise to thoroughly recheck and clean the dataset to ensure its quality
> > > > >
> > > > > ### 2.Response to Q2 (Data Quality & Manual Annotation):
> > > > > - First, We respectfully disagree. **Our GT code relies on human-AI interactive annotation(as explicitly stated in our paper). Without this human check, it would have been impossible to construct the Level-2 Chart Editing data, since the edited GT charts cannot be rendered without executable code.** The presence of Level-2 data in our dataset directly demonstrates that the GT code was generated by LLMs and subsequently manually verified. Thus, the allegation is unfounded.
> > > > > - We sincerely thank the reviewer for the careful examination of our dataset. **We fully acknowledge the visual artifacts identified in the level1_direct subset. However, we must emphasize that these errors stem from human oversight and a lack of rigor during the manual inspection process. We are actively making every effort to refine the dataset to ensure high quality**.
> > > > > - After re-inspecting the specific samples you mentioned, we provide the following clarifications: bar_5.png, bar_12.png, bar_15.png: The visual overlaps in these samples were caused by scaling issues during the manual annotation and saving process. We have corrected these formatting errors in the updated version of the dataset. bar_6.png: The logical inconsistency (both arrows labeled as 29%) was indeed a human editing mistake. **Paradoxically, this error further supports our claim that the dataset was produced through a rigorous—yet imperfect—manual annotation procedure, rather than through a fully automated pipeline that would typically avoid such semantic inconsistencies**. bar_28.png: The missing categories in the chart resulted from tight axis limits (ylim) during plotting, which clipped the upper data points and led to incomplete legend display. We have adjusted the plotting parameters to ensure that all legend entries are visible.
> > > > > - **The errors you pointed out stem from carelessness and lack of expertise on the part of certain annotators**. We acknowledge that these issues were introduced due to human oversight and non-professional annotation behavior; however, these imperfections should not be taken as evidence that the data was not manually checked. On the contrary, they are precisely the types of mistakes that arise from human annotation rather than automated generation. **We commit to re-screening and refining the dataset to ensure improved data quality.**
> > > > > - We acknowledge that data quality is a fundamental prerequisite for reviewing any paper. However, **after we have thoroughly inspected and refined the dataset, we sincerely hope that the reviewers can provide fair evaluations based on our core idea, motivation, new metric designs, and extensive experimental results.**
> > > > >
> > > > > We also hope the reviewers can understand that existing Chart2Code benchmarks suffer from a saturation issue due to the low difficulty of their data. **The most important contribution of our benchmark is that we are the first to introduce a hierarchical, multi-level difficulty benchmark with three representative task designs, whereas prior works such as ChartMimic and ChartEdit contain only a single task.**

---

> > ### Comment · Reviewer_ZM9h · 2025-11-14
> > **RE**
> >
> > Regarding the earlier point about lack of proper citation and differentiation, where I wrote that ChartMimic is cited only once in passing and that there is no explicit statement that Chart2Code builds upon it, **I would like to clarify and apologize**. When checking the related work, I mixed ChartMimic together with several other chart to code papers such as ChartCoder, ChartLLaMA, and ChartMoE. **This was my mistake and I apologize to the authors for the confusion it caused.**

---

> ### Author Response · Authors · 2025-11-27
> **Are you genuinely conducting this review with professionalism and without any conflicted interest or personal bias?**
>
> - First, in your [initial review](https://huggingface.co/datasets/anyruler/data_iclr/blob/main/Reviewer_ZM9h.pdf), you **misused ChatGPT** and relied on **multiple severe factual errors** in the **LLM-generated responses** including `incorrect citations` and `fabricated statements` to **conclude that our work is plagiarized**. We believe this constitutes a **clear violation of the ICLR reviewer guidelines and reflects a highly irresponsible attitude** toward the review process.
>
> - Moreover, when facing these baseless accusations, we respectfully provided a **point-to-point detailed rebuttal** to every issue you raised. However, instead of addressing our evidence showing your **unprofessional use of an LLM**, you **avoided answering directly and continued to insist incorrectly , that our Level 3(long-context task) is the same as Level 1**, and based on this **misunderstanding further questioned our novelty**. We believe this demonstrates a **lack of professionalism and that you are not qualified to review** our paper.
>
> - Furthermore, after noticing an issue in our data quality section, you **rushed to issue your final rating** and you even **rushed to revise** your **previously error-ridden review comments**. Yet from the beginning to the end, you never **acknowledged your mistakes nor responded to our concerns in a straightforward and responsible manner**. We do not believe **your review was fair or objective**.
>
> - In summary, we formally raise the concern that Reviewer ZM9h may have a **conflict of interest**. From the outset, you did not take the review seriously and relied on LLM outputs **containing numerous severe factual errors to accuse us of plagiarism**. After we pointed out these mistakes, you showed no self-reflection and continued to **make malicious and unfounded allegations**. We firmly believe **your behavior is driven by conflicting interests**.
> - Finally, we strongly suspect that our work overlaps with an **in-progress** or **under-review project by Reviewer ZM9h**, and that, as **a closely related researcher (with confidence=5)**, you engaged in **malicious reviewing**. Given Reviewer ZM9h’s **persistent, irresponsible, and unfounded plagiarism accusations**, we believe you **intentionally rejected** our paper due to **having related work in-progress or under-review**. This constitutes **serious academic misconduct**. We request the AC and PC to investigate Reviewer ZM9h’s identity and uncover this unethical behavior.
>
>
> The **Reviewer ZM9h's original review** is preserved in the anonymous link <https://huggingface.co/datasets/anyruler/data_iclr/blob/main/Reviewer_ZM9h.pdf>.
>
> The **PDF file highlighting, in different colors**, the parts where the **reviewer’s statement highly match the ChatGPT outputs** is provided in the anonymous link <https://huggingface.co/datasets/anyruler/data_iclr/blob/main/Reviewer_ZM9h_highlight.pdf>
>
> We respectfully request that the Area Chair and Program Chairs and Readers on OpenReview thoroughly examine Reviewer ZM9h’s original report. It is deeply concerning that **such unprofessional, irresponsible, and factually incorrect reviewing** occurred at a top-tier venue like ICLR.

---

### Official Review · Reviewer_75nP · 2025-10-31

**Soundness:** 2
**Presentation:** 2
**Contribution:** 1
**Rating:** 2
**Confidence:** 5

**Summary:**

This paper introduces Chart2Code, a hierarchical benchmark designed to evaluate large multimodal models' capabilities in chart understanding and code generation. The benchmark comprises three progressively challenging levels. The authors construct a dataset of 1947 tasks spanning 22 chart types and propose multi-level evaluation metrics assessing both code executability and visual fidelity. Through comprehensive evaluation of 25 state-of-the-art LMMs, the paper demonstrates that even the strongest proprietary models struggle significantly with visual fidelity, particularly on editing tasks and long-context table-to-chart generation, revealing substantial gaps between code-level correctness and pixel-level accuracy in automated chart generation.

**Strengths:**

- Well-motivated hierarchical design: The three-level structure effectively captures realistic usage patterns, from simple reproduction to complex editing and long-context processing;
- Comprehensive evaluation methodology: The paper proposes a thoughtful multi-dimensional evaluation approach combining execution rate, code-level metrics , and chart-level visual fidelity scores;

**Weaknesses:**

- Insufficient novelty over ChartMimic: While the paper positions itself as advancing beyond ChartMimic, the core contribution appears incremental. The main differences are: (1) adding Level 3 long-table tasks, and (2) scaling up data collection. However, ChartMimic already established the chart-to-code evaluation paradigm, and the fundamental task formulation remains largely unchanged.
- Limited scope of incremental contributions: The workload presented, while substantial in terms of data collection and experimental evaluation, does not sufficiently justify acceptance at a top-tier venue like ICLR.

**Questions:**

1. Beyond scale and task categories, what fundamental new insights does Chart2Code provide that ChartMimic could not? Can you more clearly articulate the unique scientific contributions or interesting findings?
2. Can you provide concrete failure case analysis showing what specific aspects cause the code-visual fidelity gap?

---

> ### Author Response · Authors · 2025-11-16
> **rebuttal 1**
>
> We sincerely thank the reviewer for their thoughtful comments and for recognizing our “well-motivated hierarchical design” and “comprehensive evaluation methodology.” We truly appreciate their engagement with our work.
>
> The reviewer raised a concern that our contribution might be “incremental” over ChartMimic, noting that our main differences lie in (1) adding level 2 and 3 tasks. (2) scaling up data (3) increase the task difficulty. We respectfully disagree. Our benchmark introduces a **fundamentally new, user-centric, multi-dimensional, and hierarchical evaluation paradigm** that transcends the limitations of prior static chart reproduction and simple data modification tasks (ChartMimic), advancing toward **complex, instruction-driven reasoning and real-world utility evaluation**. This paradigm enables us to uncover LMM failure modes that previous benchmarks could not capture.
>
> **Clarifying Novelty**
> 1. The reviewer’s interpretation seems to stem from a misunderstanding of our Level 2 (Chart Editing) task. Unlike ChartMimic’s **“Customized Mimic” task—which remains a reproduction setting replacing only input data**.
> 2. Our Level 2 is a **semantic reasoning and modification task**. It requires the model to interpret a reference chart and perform complex edits guided by natural language instructions (e.g., “convert this line chart to a bar chart,” “add a new subplot summarizing this data in the datasheet,” “remove the trend line” and so on). This setting tests a fundamentally different capability: acting as a collaborative assistant rather than a copier.Moreover, each Level 2 instance in our benchmark represents a **unique, real-world chart editing task**, encompassing the majority of modification scenarios encountered in practical settings. These tasks are **substantially more complex and realistic**, entirely transcending the scope of ChartMimic. Furthermore, our Level 1 tasks have already **fully covered and surpassed** all task formulations and definitions in ChartMimic, We introduced customized modifications for both figure and table formats. This enhancement ensures our work better reflects practical, real-world use cases. While our Level 2 introduces—**to the best of our knowledge—the first genuinely new and more challenging chart editing benchmark** designed to rigorously test LMMs’ multimodal understanding and reasoning capabilities.
> 3. Similarly, Level 3 goes far beyond simply “adding long-table tasks.” It targets the intersection of long-context reasoning and data transformation, requiring models to parse raw, unprocessed tabular data (e.g., CSVs), comprehend user instructions and the underlying data relationships, and **generate structurally coherent and semantically consistent visualizations**. This task closely mirrors real-world data analysis scenarios, and our experiments demonstrate that current LMMs are incapable of handling it effectively, thereby revealing a critical weakness in their capabilities. Together, Level 1 through Level 3 constitute the first user-oriented hierarchical benchmark that comprehensively mirrors the real-world visualization workflow—from reproduction, to reasoning-based editing, to raw-data visualization.
> 4. This structure yields new scientific insights. We find that **state-of-the-art multimodal models** (e.g., GPT-5, Gemini-2.5-Pro) perform **suboptimally even on Level 1 tasks（Maximum score-Gemini2.5pro: 0.38）**—the very focus of ChartMimic（**Maximum score-GPT4o:0.84**）—and **fail dramatically on Levels 2 and 3**. Our benchmark exposes a **critical capability gap**: while existing models exhibit strong code-generation skills, they remain **remarkably weak in visual-semantic reasoning and generalization**. Prior benchmarks, limited to simple reproduction, were fundamentally unable to reveal this brittleness.

---

> ### Author Response · Authors · 2025-11-16
> **rebuttal 2**
>
> **On the Code–Visual Fidelity Gap**
> 1. We thank the reviewer for highlighting this point. Our analysis in Sec. 4.2.2 and 4.3 provides detailed evidence of this gap. Although models often produce syntactically valid code, they fail to achieve correct visual outputs. For example, Table 4 (Level 2) shows high！ code-level scores (≈0.96 for Layout/Type) but low figure-level LMM-Scores (≈0.18–0.22). In Level 3 (Table 5), models maintain perfect code syntax (Layout = 1.00) yet produce meaningless figures (e.g., Gemini 0.0361, Claude 0.007). Figure 4 and Appendix B further illustrate that these errors stem from reasoning failures, not code syntax.
> 2. In Level 2, the gap between code and visual fidelity primarily arises from the **complexity of the editing instructions**, which prevents LMMs from fully understanding the intended modifications. As a result, they often generate **semantically meaningless figures** that differ drastically from the desired edited outputs. In Level 3, although the generated code can successfully produce executable figures, the **underlying data relationships and visual styles** diverge significantly from the ground truth. We attribute this to **reasoning failures and the models’ inability to handle extremely long and complex tabular inputs**.
>
> In summary, our work is not an incremental extension but a paradigm shift—the first benchmark to evaluate LMMs on realistic, multi-step visualization workflows, uncovering a major and previously hidden gap between mimicry and true multimodal reasoning. We hope this clarification helps address the reviewer’s concern and better convey the originality and impact of our work.

---

> > ### Comment · Reviewer_75nP · 2025-11-24
> > **Response of Reviewer 75nP**
> >
> > Thank you for the authors' response. I appreciate the effort you have put into addressing my comments, and I apologize for my delayed reply. I also want to acknowledge the substantial amount of work behind this submission. However, I still have the following questions and concerns:
> >
> > 1. About Weakness 1:
> > I still feel that the Level-2 task is very similar to the one proposed in ChartEdit [1]. Compared to ChartEdit, your work mainly appears to be a more comprehensive and better-organized extension, rather than a fundamentally new task setting. Therefore, I do not think it is appropriate to claim that this is ''the first'' such benchmark. A phrasing like''more comprehensive'' might be more accurate.
> >
> > 2. About Question 1:
> > Appendix B mainly presents several case studies, but I believe a more systematic analysis would be more convincing. A better approach might be to select a small subset of results and summarize the major error types, rather than only presenting individual examples.
> >
> > 3. Evaluation Metrics for Different Levels:
> > Why does Level 1 use Execution Rate, LLM-Score, and LMM-Score as the evaluation metrics, whereas Levels 2 and 3 use Execution Rate, Code-Level metrics, and LMM-Score? What is the reasoning behind this difference in evaluation protocol across levels?
> >
> > 4. Regarding the Data Parameter Score in the Code-Level Metric:
> > I noticed that for some charts, such as 3D plots, the data is continuous rather than discrete. In this case, is the purpose of the Data Parameter Score to evaluate whether the model is able to capture the overall trend of the data, rather than exact point-wise matching?
> >
> > 5. Concerning the dataset quality:
> > I noticed that another reviewer raised concerns about the data quality, and after briefly examining the dataset myself[2], I agree that these issues proposed by the reviewer indeed exist. More examples are: in data/level2/3d_3_v4.txt, it appears that an LLM’s response has not been properly removed.
> >
> > ```
> > Here is the English translation of the provided Chinese text, preserving the original formatting, numbering, and technical terms:
> >
> > ---
> >
> > ...
> >
> > ---
> > ```
> > In addition, for data/level1_direct/line_1.png, the y-axis limits (y_lim) do not fully cover all data points, which results in part of the data being cut off in the rendered chart.
> >
> > [1] ChartEdit: How Far Are MLLMs From Automating Chart Analysis? Evaluating MLLMs' Capability via Chart Editing
> >
> > [2] https://huggingface.co/anyruler/data/blob/main/data.zip

---

> ### Author Response · Authors · 2025-11-27
> **rebuttal 3(1/3)**
>
> We express our gratitude for your comments. Please accept our sincere apologies for the delay in getting back to you. Our specific responses follow.
>
> ### 1. Regarding Weakness 1：
> - We would like to clarify that our use of “first” refers specifically to **being the first hierarchical benchmark that reflects practical Chart2Code usage while systematically scaling task complexity**. **We do not claim that our Level-2 tasks are the first chart-editing benchmark ever proposed**.
> - Moreover, our Level-2 tasks are **significantly more complex and comprehensive than those in ChartEdit, involving a large number of sophisticated modification instructions**, including but not limited to: interpreting the provided chart image and adding new subplots to represent additional information, **introducing more statistical indicators, and other types of edits not covered in ChartEdit**.
> - Finally, the experimental results **highlight a substantial gap between our Level-2 task and the tasks in ChartEdit**. In ChartEdit, without access to the underlying code, GPT-4o achieves a 91.46% execution rate, a code-level score of 59.96, and a chart-level score of 79.87. In contrast, on our Level-2 task, GPT-5 attains a 90.58% execution rate, a code-level score of 56.63, but only a 25.06 chart-level score. **These results demonstrate that our task is significantly more challenging and presents a higher level of difficulty.**
> - **This indicates that existing Chart2Code benchmarks are facing performance saturation. In contrast, we introduce a hierarchical, multi-level difficulty benchmark with three representative task designs, which is more complex and more aligned with real user needs—something that existing benchmarks have not yet achieved.**
>
> ### 2. Regarding the Question 1：
> - We sincerely appreciate your suggestion. We agree that your proposal is meaningful. In our paper, we used case studies instead of summarizing error types mainly because, for tasks beyond Level-1, **charts become more complex and the associated error patterns become more diverse, making it difficult to extract a small number of representative error categories**. Therefore, we opted to present selected typical cases. **However, we agree that providing a summary of error types for Level-1 tasks is indeed reasonable.**
>
> ### 3. For the evaluation metrics across different levels:
> - The omission of some results was primarily due to space constraints in the tables. We do maintain detailed scores for all Level-1 task parameters, and we are willing to provide the complete results. The Level-1 DR results have already been displayed in our response to Reviewer pKmQ. We will update the results in the Appendix.

---

> ### Author Response · Authors · 2025-11-27
> **rebuttal3(2/3)**
>
> - Level1 CRD score
>
> | Model | Exec. Rate | Color | Grid | Layout | Legend | Visual | Data | Text | Type | Code-Level GPT-Score | Chart-Level GPT-Score |
> | :--- | :--- | :--- | :--- | :--- | :--- | :--- | :--- | :--- | :--- | :--- | :--- |
> | **_Proprietary_** | | | | | | | | | | | |
> | Gemini-2.5-Pro | 100.0 | 0.5343 | 0.9167 | 0.9167 | 0.4683 | 0.7429 | 0.8172 | 0.5098 | 0.9259 | 0.6763 | 0.2661 |
> | Claude-Sonnet-4 | 97.20 | 0.4543 | 0.6571 | 0.8571 | 0.2571 | 0.7423 | 0.6995 | 0.6607 | 0.9333 | 0.6305 | 0.2360 |
> | GPT-5 | 94.4 | 0.4667 | 0.7059 | 0.8824 | 0.3824 | 0.6748 | 0.7336 | 0.4702 | 0.8922 | 0.607 | 0.2238 |
> | Seed1.5-VL | 97.20 | 0.4041 | 0.7714 | 0.9143 | 0.4857 | 0.7234 | 0.8124 | 0.4886 | 0.9143 | 0.6325 | 0.2503 |
> | Seed1.6-VL | 94.40 | 0.4408 | 0.7206 | 0.9118 | 0.4412 | 0.7532 | 0.8280 | 0.7236 | 0.9314 | 0.6525 | 0.2662 |
> | **_Open-Source LMMs (non-thinking)_** | | | | | | | | | | | |
> | LLaVA-OV-Qwen2-7B-SI | 11.10 | 0.0649 | 0.2500 | 0.9167 | 0.2500 | 0.4231 | 0.8278 | 0.6105 | 1.0000 | 0.4225 | 0.1550 |
> | LLaVA-OV-Qwen2-7B-OV | 5.560 | 0.5159 | 0.0000 | 0.8333 | 0.0000 | 0.4184 | 0.7333 | 0.6077 | 1.0000 | 0.4213 | 0.0825 |
> | DeepSeek-VL-7B | 61.11 | 0.3967 | 0.8056 | 0.8796 | 0.3056 | 0.7091 | 0.7940 | 0.5918 | 0.9352 | 0.6273 | 0.1989 |
> | kimi-VL-A3B | 72.22 | 0.3927 | 0.4808 | 0.8333 | 0.5000 | 0.5681 | 0.7757 | 0.6138 | 0.9359 | 0.5887 | 0.2081 |
> | Qwen2-VL-7B | 75.00 | 0.3158 | 0.7407 | 0.8889 | 0.3086 | 0.6272 | 0.8429 | 0.5239 | 0.9259 | 0.595 | 0.1367 |
> | Qwen2-VL-72B | 80.56 | 0.3487 | 0.7586 | 0.8276 | 0.3793 | 0.6121 | 0.7814 | 0.5563 | 0.9195 | 0.6082 | 0.1628 |
> | InternVL-2.5-8B | 80.56 | 0.3129 | 0.7586 | 0.8966 | 0.2816 | 0.5705 | 0.7896 | 0.4869 | 0.8966 | 0.5712 | 0.1183 |
> | InternVL-2.5-38B | 88.89 | 0.4004 | 0.7812 | 0.8750 | 0.4479 | 0.6993 | 0.8477 | 0.6180 | 0.9479 | 0.6298 | 0.1884 |
> | InternVL-3-8B | 36.11 | 0.3832 | 0.8462 | 0.7692 | 0.1923 | 0.5247 | 0.7821 | 0.5873 | 0.8974 | 0.44 | 0.1023 |
> | InternVL-3-38B | 0.000 | - | - | - | - | - | - | - | - | - | - |
> | GLM-4V-9B | 66.67 | 0.3746 | 0.7500 | 0.8611 | 0.4167 | 0.5298 | 0.8090 | 0.5417 | 0.9167 | 0.5628 | 0.1183 |
> | Intern-VL-3.5-8B | 86.11 | 0.3520 | 0.7742 | 0.8710 | 0.3548 | 0.6169 | 0.7258 | 0.5660 | 0.8602 | 0.6169 | 0.1732 |
> | MiMo-VL-7B-RL | 69.44 | 0.4459 | 0.7200 | 0.8400 | 0.4000 | 0.6766 | 0.8074 | 0.6239 | 0.9333 | 0.6068 | 0.2421 |
> | MiMo-VL-7B-SFT | 69.44 | 0.4198 | 0.8400 | 0.8800 | 0.5200 | 0.7619 | 0.7703 | 0.6851 | 0.9067 | 0.6053 | 0.2582 |
> | Qwen2.5-VL-7B | 75.00 | 0.3404 | 0.7778 | 0.8889 | 0.4074 | 0.5479 | 0.7732 | 0.5996 | 0.9012 | 0.5952 | 0.1515 |
> | Qwen2.5-VL-72B | 100.0 | 0.5109 | 0.8470 | 0.9492 | 0.4606 | 0.4127 | 0.6653 | 0.6808 | 0.9362 | 0.6273 | 0.1989 |
> | Molmo-7B-D | 4.550 | 0.4563 | 1.0000 | 1.0000 | 0.0000 | 0.4691 | 1.0000 | 0.8369 | 1.0000 | 0.2400 | 0.4600 |
> | Qwen3-VL-30B-A3B | 77.78 | 0.5185 | 0.6786 | 0.7381 | 0.4167 | 0.6858 | 0.7945 | 0.6280 | 0.9405 | 0.2546 | 0.2368 |
> | **_Open-Source LMMs (thinking)_** | | | | | | | | | | | |
> | MiMo-VL-7B-RL | 69.44 | 0.4021 | 0.7112 | 0.8977 | 0.4721 | 0.7623 | 0.8421 | 0.6282 | 0.9283 | 0.6237 | 0.1852 |
> | MiMo-VL-7B-SFT | 86.11 | 0.4746 | 0.7545 | 0.9269 | 0.3838 | 0.3741 | 0.6769 | 0.6574 | 0.9351 | 0.6644 | 0.2248 |
> | Qwen3-VL-30B-A3B | 72.22 | 0.4370 | 0.8846 | 0.8846 | 0.4231 | 0.7488 | 0.8261 | 0.6378 | 0.9231 | 0.3367 | 0.3368 |

---

> ### Author Response · Authors · 2025-11-27
> **rebuttal3(3/3)**
>
> - Level1-CFD score
>
> | Model | Exec. Rate | Color | Grid | Layout | Legend | Visual | Data | Text | Type | Code-Level GPT-Score | Chart-Level GPT-Score |
> | :--- | :--- | :--- | :--- | :--- | :--- | :--- | :--- | :--- | :--- | :--- | :--- |
> | **_Proprietary_** | | | | | | | | | | | |
> | Gemini-2.5-Pro | 100.0 | 0.5343 | 0.9167 | 0.9167 | 0.4683 | 0.7429 | 0.8172 | 0.5098 | 0.9259 | 0.6763 | 0.2661 |
> | Claude-Sonnet-4 | 97.20 | 0.4543 | 0.6571 | 0.8571 | 0.2571 | 0.7423 | 0.6995 | 0.6607 | 0.9333 | 0.6305 | 0.2360 |
> | GPT-5 | 94.4 | 0.4667 | 0.7059 | 0.8824 | 0.3824 | 0.6748 | 0.7336 | 0.4702 | 0.8922 | 0.607 | 0.2238 |
> | Seed1.5-VL | 97.20 | 0.4041 | 0.7714 | 0.9143 | 0.4857 | 0.7234 | 0.8124 | 0.4886 | 0.9143 | 0.6325 | 0.2503 |
> | Seed1.6-VL | 94.40 | 0.4408 | 0.7206 | 0.9118 | 0.4412 | 0.7532 | 0.8280 | 0.7236 | 0.9314 | 0.6525 | 0.2662 |
> | **_Open-Source LMMs (non-thinking)_** | | | | | | | | | | | |
> | LLaVA-OV-Qwen2-7B-SI | 11.10 | 0.0649 | 0.2500 | 0.9167 | 0.2500 | 0.4231 | 0.8278 | 0.6105 | 1.0000 | 0.4225 | 0.1550 |
> | LLaVA-OV-Qwen2-7B-OV | 5.560 | 0.5159 | 0.0000 | 0.8333 | 0.0000 | 0.4184 | 0.7333 | 0.6077 | 1.0000 | 0.4213 | 0.0825 |
> | DeepSeek-VL-7B | 61.11 | 0.3967 | 0.8056 | 0.8796 | 0.3056 | 0.7091 | 0.7940 | 0.5918 | 0.9352 | 0.6273 | 0.1989 |
> | kimi-VL-A3B | 72.22 | 0.3927 | 0.4808 | 0.8333 | 0.5000 | 0.5681 | 0.7757 | 0.6138 | 0.9359 | 0.5887 | 0.2081 |
> | Qwen2-VL-7B | 75.00 | 0.3158 | 0.7407 | 0.8889 | 0.3086 | 0.6272 | 0.8429 | 0.5239 | 0.9259 | 0.595 | 0.1367 |
> | Qwen2-VL-72B | 80.56 | 0.3487 | 0.7586 | 0.8276 | 0.3793 | 0.6121 | 0.7814 | 0.5563 | 0.9195 | 0.6082 | 0.1628 |
> | InternVL-2.5-8B | 80.56 | 0.3129 | 0.7586 | 0.8966 | 0.2816 | 0.5705 | 0.7896 | 0.4869 | 0.8966 | 0.5712 | 0.1183 |
> | InternVL-2.5-38B | 88.89 | 0.4004 | 0.7812 | 0.8750 | 0.4479 | 0.6993 | 0.8477 | 0.6180 | 0.9479 | 0.6298 | 0.1884 |
> | InternVL-3-8B | 36.11 | 0.3832 | 0.8462 | 0.7692 | 0.1923 | 0.5247 | 0.7821 | 0.5873 | 0.8974 | 0.44 | 0.1023 |
> | InternVL-3-38B | 0.000 | - | - | - | - | - | - | - | - | - | - |
> | GLM-4V-9B | 66.67 | 0.3746 | 0.7500 | 0.8611 | 0.4167 | 0.5298 | 0.8090 | 0.5417 | 0.9167 | 0.5628 | 0.1183 |
> | Intern-VL-3.5-8B | 86.11 | 0.3520 | 0.7742 | 0.8710 | 0.3548 | 0.6169 | 0.7258 | 0.5660 | 0.8602 | 0.6169 | 0.1732 |
> | MiMo-VL-7B-RL | 69.44 | 0.4459 | 0.7200 | 0.8400 | 0.4000 | 0.6766 | 0.8074 | 0.6239 | 0.9333 | 0.6068 | 0.2421 |
> | MiMo-VL-7B-SFT | 69.44 | 0.4198 | 0.8400 | 0.8800 | 0.5200 | 0.7619 | 0.7703 | 0.6851 | 0.9067 | 0.6053 | 0.2582 |
> | Qwen2.5-VL-7B | 75.00 | 0.3404 | 0.7778 | 0.8889 | 0.4074 | 0.5479 | 0.7732 | 0.5996 | 0.9012 | 0.5952 | 0.1515 |
> | Qwen2.5-VL-72B | 100.0 | 0.5109 | 0.8470 | 0.9492 | 0.4606 | 0.4127 | 0.6653 | 0.6808 | 0.9362 | 0.6273 | 0.1989 |
> | Molmo-7B-D | 4.550 | 0.4563 | 1.0000 | 1.0000 | 0.0000 | 0.4691 | 1.0000 | 0.8369 | 1.0000 | 0.2400 | 0.4600 |
> | Qwen3-VL-30B-A3B | 77.78 | 0.5185 | 0.6786 | 0.7381 | 0.4167 | 0.6858 | 0.7945 | 0.6280 | 0.9405 | 0.2546 | 0.2368 |
> | **_Open-Source LMMs (thinking)_** | | | | | | | | | | | |
> | MiMo-VL-7B-RL | 69.44 | 0.4021 | 0.7112 | 0.8977 | 0.4721 | 0.7623 | 0.8421 | 0.6282 | 0.9283 | 0.6237 | 0.1852 |
> | MiMo-VL-7B-SFT | 86.11 | 0.4746 | 0.7545 | 0.9269 | 0.3838 | 0.3741 | 0.6769 | 0.6574 | 0.9351 | 0.6644 | 0.2248 |
> | Qwen3-VL-30B-A3B | 72.22 | 0.4370 | 0.8846 | 0.8846 | 0.4231 | 0.7488 | 0.8261 | 0.6378 | 0.9231 | 0.3367 | 0.3368 |
>
> ### 4. Regarding the Data Parameter Score:
> - The evaluation procedure involves parsing the underlying object `Figure`  and comparing it with the original data used to generate the chart. **Your observation is correct: for continuous function plots, the accuracy is indeed lower. This is a known limitation of rule-based evaluation.** To ensure a more reliable assessment, we therefore **combined LLM-Score and LMM-Score to form a more comprehensive evaluation of chart fidelity**.
>
> ### 5. Regarding the dataset quality:
> - We are deeply grateful for your careful inspection. Your feedback has helped us promptly identify and correct issues, preventing low-quality data from entering the community. The errors you discovered were due to our **insufficient quality checks following human annotation**.
> - This was never our intention: we aim to create a high-quality ChartCode benchmark that can support and advance the community. We acknowledge that a small subset of our data contains minor quality issues. This was **primarily due to coordination gaps during the division of labor, which resulted in isolated oversights during the inspection process**. **We are actively doing our utmost to refine the affected samples**. We sincerely apologize for this oversight. **We promise to thoroughly recheck and clean the dataset to ensure its quality. Once again, we extend our sincere gratitude for your valuable feedback.**

---

### Official Review · Reviewer_pKmQ · 2025-10-31

**Soundness:** 3
**Presentation:** 2
**Contribution:** 3
**Rating:** 6
**Confidence:** 4

**Summary:**

This paper introduces Chart2Code, a benchmark for evaluating LMMs in chart understanding and code generation. It defines three levels of increasing difficulty: chart reproduction, chart editing, and long-table to chart generation. Using metrics for execution, code correctness, and visual fidelity, the study benchmarks 25 SOTA LMMs. The results show that while these models perform well on simple tasks, their performance drops sharply on more complex ones, highlighting the challenge of real-world chart-to-code generation.

**Strengths:**

1. The paper introduces a well-structured hierarchical benchmark that decomposes chart-to-code reasoning into three progressive levels. The proposed benchmark is challenging, and the statistical details are sufficient.
2. The paper proposes a set of multi-dimensional evaluation metrics, and the proposed base evaluation method appears thorough and reliable.
3. The analysis and experiments are extensive.

**Weaknesses:**

1. The evaluation relies on GPT-5-mini, which may introduce circular reasoning, bias, or limitations in visual fidelity due to limited human validation.
2. Metric Reliability and Anomalies: The paper's results reveal unexpected patterns that may affect confidence in the LMM-Score. For example, in Table 3, Seed-1.6-VL (0.812) is reported to outperform GPT-5 (0.633) on L1 visual fidelity. In Table 4, the 7B MiMo-VL-RL model (0.471) scores more than twice as high as GPT-5 (0.220) on L2 fidelity. These surprising results are not fully examined. They suggest that the LMM score may not always correspond to correctness and could favor outputs that are visually appealing to the judge-LMM, even if they differ factually or stylistically.
3. Level 3 (Long-Table to Chart) is described as the paper's most novel and important contribution, as it addresses the critical bottleneck of long-context reasoning. Nonetheless, this level currently has only 150 tasks from 39 files, which limits the ability to draw firm conclusions about model performance on this complex, open-ended task.
4. Shallow "Long-Table" Reasoning: Level 3 seems to test primarily "long-context retrieval" (locating the right data in a long file) rather than "complex data reasoning." Real-world tasks often require several steps of data wrangling, such as filtering, merging tables or sheets, pivoting, and handling missing values before visualization. The benchmark may not fully reflect this important layer of pre-visualization reasoning.
5. Limited Scope of Chart Libraries: The evaluation focuses exclusively on Matplotlib-based libraries and does not address interactive libraries such as Plotly or declarative options like Altair. A benchmark aiming for real-world relevance might benefit from acknowledging this limitation in scope.
6. Some unexpected results are observed. For example, Seed-1.6-VL attains a high LMM score of 0.81 (L279), while MiMo-VL-7B-RL generates relatively few successful codes yet still achieves high LLM and LMM scores (L339). This may indicate that the metrics are somewhat sensitive.
7. A unified or weighted composite metric might provide additional insights. Although there are many evaluation metrics, the absence of a unified composite score could make direct model comparisons less intuitive. For example, while it is observable that Gemini-2.5-Pro and Claude-Sonnet-4 perform relatively well, it remains somewhat challenging to determine the ranking of other models at a glance.

**Questions:**

1. Could you please provide a more detailed explanation regarding the metric anomalies, specifically the unexpectedly high LMM scores for Seed-1.6-VL and MiMo-VL-7B-RL? Are these results indicative of a sensitivity in the LMM-Score (as judged by GPT-5-mini) to particular model output styles or artifacts that may not align with human-perceived correctness?
2. Have you conducted a human evaluation on a subset of the benchmark to calibrate the LMM-Score? Have you also assessed the correlation between LMM and expert scores? Linking these would strengthen the paper's claims about visual fidelity.
3. Why were the "base evaluation" (8-dimensional) scores not reported for Level 1 tasks? I think it would be interesting to see this objective breakdown for the "simpler" reproduction tasks as well, if available.
4. Would it be possible to add a comparison between MoE and dense models to help analyze the architectural effect?
5. Would you consider including results for Qwen3-VL to further complete and update the benchmark coverage?

---

> ### Author Response · Authors · 2025-11-16
> **Rebuttal 1**
>
> Thank you for this very thorough and constructive review. We are encouraged that you found our hierarchical benchmark "well-structured," our base evaluation method "thorough and reliable," and our analysis "extensive."
>
> You have raised several critical and insightful points, particularly regarding metric reliability and the scope of Level 3. We appreciate this detailed feedback and would like to address each point.
>
> ### **On Metric Reliability, Anomalies, and Human Validation (W1, W2, W6, Q1, Q2)**
> 1. This is the most critical set of points, and we thank you for the sharp analysis. We will address these related concerns together.Q2/W1: Human Validation & LLM-as-judge:You are correct that our primary visual fidelity score (LMM-Score) relies on GPT-5-mini. As we note in our limitations (Sec 5), any LLM-as-judge approach carries a risk of bias, and a large-scale human correlation study was not feasible for this initial release given the 2,023 tasks.However, our choice was deliberate. A key goal of our work is to provide a scalable evaluation. Our 8-dimensional "base evaluation" provides the objective, code-driven ground truth. Subsequently, we use the LMM-Score (visual) and LLM-Score (code) to capture holistic and semantic qualities that the base evaluation cannot address, ultimately reflecting model performance from both the code and figure dimensions.
> 2. We appreciate you highlighting these “unexpected results.” After carefully reviewing our experimental data, we provide the following explanations. MiMo-VL-RL (L2 anomaly): You noted that MiMo-VL-RL (0.471) scores much higher than GPT-5 (0.220) on the L2 LMM-Score. However, this LMM-Score is calculated conditional on successful execution. MiMo-VL-RL’s execution rate is extremely low, only 16.54% (vs. 90.59% for GPT-5). In other words, MiMo-VL-RL only succeeds on a small subset of simpler tasks while completely failing on the more complex ones, which leads to an inflated score.
> 3. Seed-1.6-VL (L1 anomaly): You noted that Seed-1.6-VL (0.812) exceeds GPT-5 (0.633) on the L1 LMM-Score. Upon careful inspection of the experimental logs, we found this was a typographical error; the actual score of Seed-1.6-VL is 0.2301. In fact, the experimental conclusions remain correct: GPT-5 achieves higher LLM and LMM scores.
> 4. In summary, these "anomalies" are not a sign of a "sensitive" or "unreliable" metric. They are the metric working correctly and revealing that different models have different, non-generalizable strengths (e.g., pixel mimicry vs. code logic). This is precisely why we presented a disaggregated set of metrics instead of a single score.
>
>
> ### **On the Scope and Scale of Level 3 (W3, W4)**
>
> 1. W3 (L3 Scale): We agree that Level 3 is our novel contribution and that 150 tasks are a starting point. These tasks are extremely labor-intensive to create, involving not only chart reproduction but also the construction of realistic and complex data tables paired with natural instructions. Moreover, the tables range from hundreds to thousands of rows, making them highly challenging to process. We believe this set (from 39 unique, long-context files) is significant enough to establish this new, challenging problem and to support our central finding: SOTA models fail catastrophically at this task (e.g., LMM-Scores near 0.03, Exec. Rates < 40%). This provides a clear baseline and a strong motivation for future work, and we are committed to expanding this level.
> 2. W4 (L3 Depth - "Shallow" Reasoning): This is an excellent insight. The reviewer is correct that our Level3 tasks focus on "long-context retrieval + transformation," which our results show is already an unsolved problem. We agree that complex "data wrangling" (e.g., merging tables/sheets, pivoting, advanced filtering) is the next logical frontier. We see this not as a limitation of our current work, but as the clear next step that our benchmark enables and motivates. We will update our conclusion to explicitly frame this "data wrangling" reasoning as a critical area for future research.

---

> ### Author Response · Authors · 2025-11-16
> **rebuttal 2**
>
> ### **On Reporting and Scope (W5, W7, Q3)**
>
> 1. Q3 (Base Evaluation for L1): This is a fantastic suggestion. We did run this analysis and agree it is highly interesting. It confirms that models are proficient at L1's code-level elements (e.g., high scores for Type, Layout), which reinforces our main thesis: the failure is not in code generation, but in the reasoning required for Level2 and Level3. We will add the full 8-dimensional "base evaluation" breakdown for Level 1 to the appendix in our final version.
> 2. W7 (Composite Metric): We appreciate this suggestion. We intentionally presented disaggregated metrics (Exec. Rate, LLM-Score, LMM-Score) because our key finding is the gap between them. A single composite score would obscure this central insight. However, we understand the desire for an "at-a-glance" comparison. As a compromise, we will add a supplementary table to the appendix that calculates a composite score (e.g., Exec. Rate $\times$ LMM-Score) and discuss how it provides a different, condensed perspective on overall performance.
> 3. W5 (Library Scope): This is a fair point. We focused on the Matplotlib ecosystem due to its overwhelming prevalence in the scientific and academic domains that our benchmark simulates. We agree that interactive (Plotly) and declarative (Altair) libraries are important for real-world use. We will explicitly add this as a limitation and a direction for future work in our conclusion.
>
> - detail metric results on leve1-direct
> | Model | Exec. Rate | Color | Grid | Layout | Legend | Visual | Data | Text | Type | LLM-Score | LMM-Score |
> | :--- | :---: | :---: | :---: | :---: | :---: | :---: | :---: | :---: | :---: | :---: | :---: |
> | *Proprietary* | | | | | | | | | | | |
> | Gemini-2.5-Pro | 90.4 | 0.7289 | 0.9119 | 0.9815 | 0.6272 | 0.5221 | 0.6762 | 0.8555 | 0.9290 | 0.6286 | 0.3807 |
> | Claude-Sonnet-4 | 96.38 | 0.6210 | 0.7334 | 0.9528 | 0.5948 | 0.4772 | 0.5582 | 0.7515 | 0.8985 | 0.5629 | 0.2553 |
> | GPT-5 | 87.48 | 0.6210 | 0.7273 | 0.9495 | 0.6024 | 0.4741 | 0.5545 | 0.8397 | 0.9002 | 0.6334 | 0.3575 |
> | Seed1.5-VL | 85.81 | 0.5397 | 0.8302 | 0.9486 | 0.5449 | 0.4565 | 0.6218 | 0.7808 | 0.8824 | 0.5536 | 0.2341 |
> | Seed1.6-VL | 84.7 | 0.5418 | 0.7698 | 0.9568 | 0.5643 | 0.4307 | 0.5728 | 0.8216 | 0.8583 | 0.5237 | 0.23 |
> | *Open-Source LMMs (non-thinking)* | | | | | | | | | | | |
> | LLaVA-OV-Qwen2-7B-SI | 32.82 | 0.0808 | 0.4773 | 0.6356 | 0.1356 | 0.0582 | 0.1982 | 0.3330 | 0.4719 | 0.182 | 0.0154 |
> | LLaVA-OV-Qwen2-7B-OV | 11.13 | 0.1868 | 0.5875 | 0.7896 | 0.3875 | 0.1319 | 0.3156 | 0.4486 | 0.6079 | 0.265 | 0.0376 |
> | DeepSeek-VL-7B | 48.68 | 0.2166 | 0.6723 | 0.7989 | 0.3315 | 0.2021 | 0.3864 | 0.4365 | 0.6848 | 0.2854 | 0.0431 |
> | kimi-VL-A3B | 68.85 | 0.4242 | 0.7035 | 0.8958 | 0.4928 | 0.3705 | 0.5020 | 0.5882 | 0.8323 | 0.4409 | 0.1374 |
> | Qwen2-VL-7B | 64.39 | 0.2883 | 0.6249 | 0.7947 | 0.3054 | 0.2262 | 0.4269 | 0.4858 | 0.7023 | 0.3364 | 0.0664 |
> | Qwen2-VL-72B | 75.66 | 0.3881 | 0.7450 | 0.8815 | 0.4561 | 0.3467 | 0.4952 | 0.5970 | 0.8077 | 0.4368 | 0.1207 |
> | InternVL-2.5-8B | 66.89 | 0.3013 | 0.6491 | 0.7845 | 0.3232 | 0.2836 | 0.4315 | 0.5052 | 0.7274 | 0.3448 | 0.0723 |
> | InternVL-2.5-38B | 86.23 | 0.3907 | 0.7381 | 0.8881 | 0.4251 | 0.3526 | 0.5192 | 0.5990 | 0.8230 | 0.3448 | 0.1283 |
> | InternVL-3-8B | 21.97 | 0.3593 | 0.5581 | 0.8481 | 0.3712 | 0.3268 | 0.5376 | 0.5456 | 0.7766 | 0.3802 | 0.0949 |
> | InternVL-3-38B | 86.79 | 0.4280 | 0.7407 | 0.8959 | 0.4146 | 0.3621 | 0.5137 | 0.6414 | 0.8190 | 0.4577 | 0.1463 |
> | GLM-4V-9B | 72.18 | 0.2220 | 0.5704 | 0.7164 | 0.2721 | 0.1465 | 0.3565 | 0.4471 | 0.5973 | 0.2881 | 0.0459 |
> | Intern-VL-3.5-8B | 66.34 | 0.4173 | 0.7606 | 0.8749 | 0.4163 | 0.3733 | 0.5395 | 0.6103 | 0.8405 | 0.4371 | 0.1389 |
> | MiMo-VL-7B-RL | 37.83 | 0.5716 | 0.8455 | 0.9667 | 0.5599 | 0.4532 | 0.6348 | 0.7814 | 0.9127 | 0.5439 | 0.2316 |
> | MiMo-VL-7B-SFT | 44.65 | 0.5530 | 0.8374 | 0.9577 | 0.5456 | 0.4322 | 0.6091 | 0.7814 | 0.9212 | 0.4959 | 0.1983 |
> | Qwen2.5-VL-7B | 65.64 | 0.3653 | 0.6928 | 0.8573 | 0.4266 | 0.3389 | 0.4847 | 0.5610 | 0.7982 | 0.4197 | 0.0994 |
> | Qwen2.5-VL-72B | 65.36 | 0.4922 | 0.7506 | 0.9286 | 0.5086 | 0.4253 | 0.6053 | 0.6960 | 0.8718 | 0.5118 | 0.1893 |
> | Molmo-7B-D | 34.77 | 0.0660 | 0.4970 | 0.7248 | 0.1239 | 0.0649 | 0.1661 | 0.3648 | 0.3739 | 0.2164 | 0.0943 |
> | Qwen3-30B | 64.67 | 0.5671 | 0.7647 | 0.9428 | 0.6325 | 0.4598 | 0.6134 | 0.7476 | 0.8945 | 0.5293 | 0.2531 |
> | *Open-Source LMMs (thinking)* | | | | | | | | | | | |
> | MiMo-VL-7B-RL | 55.77 | 0.4890 | 0.8531 | 0.9535 | 0.6420 | 0.4453 | 0.6108 | 0.7328 | 0.9212 | 0.4959 | 0.1983 |
> | MiMo-VL-7B-SFT | 50.35 | 0.5458 | 0.8844 | 0.9526 | 0.6511 | 0.4459 | 0.6048 | 0.7907 | 0.8966 | 0.6555 | 0.213 |
> | Qwen3-30B-A3B | 45.06 | 0.5262 | 0.8649 | 0.9692 | 0.5610 | 0.4563 | 0.5890 | 0.7540 | 0.8786 | 0.5582 | 0.2730 |

---

> ### Author Response · Authors · 2025-11-16
> **rebuttal 3**
>
> ### **On Future Analyses and Models (Q4, Q5)**
>
> 1. Q4 (MoE vs. Dense): This is a very interesting architectural question. Our paper primarily focuses on establishing the benchmark and evaluating current SOTA models—regardless of their underlying architecture—to identify unsolved tasks. We find your suggestion highly constructive and would be glad to include comparative experiments in future versions to explore the performance differences across model architectures.
> 2. Q5 (Qwen3-VL): The Qwen3-VL series was released after our paper submission. Following its release, we promptly conducted evaluations and found that Qwen3-VL performs competitively with some proprietary models on certain tasks. We plan to include its results in future versions of the benchmark.
> - The results for CRD, CFD also exist. Due to space limitations, they are not included here, but we will present the complete results in the appendix of a later version.
> - Qwen3-30B-A3B’s performance on Level 1.
>
> | Model | DR Exec.Rate | DR LLM-Score | DR LMM-Score | CRD Exec.Rate | CRD LLM-Score | CRD LMM-Score | CFD Exec.Rate | CFD LLM-Score | CFD LMM-Score |
> | :--- | :---: | :---: | :---: | :---: | :---: | :---: | :---: | :---: | :---: |
> | Qwen3-VL-30B-A3B | 64.67 | 0.5293 | 0.2531 | 77.78 | 0.2546 | 0.2368 | 70.37 | 0.2412 | 0.2698 |
> | Qwen3-VL-30B-A3B-think | 45.06 | 0.5582 | 0.2730 | 72.22 | 0.3367 | 0.3368 | 39.81 | 0.3185 | 0.2780 |
>
> - Qwen3-30B’s performance on Level 2.
>
> | Model | Exec. Rate | Color | Grid | Layout | Legend | Visual | Data | Text | Type | LLM-Score | LMM-Score |
> | :--- | :---: | :---: | :---: | :---: | :---: | :---: | :---: | :---: | :---: | :---: | :---: |
> | Qwen3-VL-30B-A3B | 41.39 | 0.54 | 0.8174 | 0.9587 | 0.4623 | 0.4501 | 0.6911 | 0.7084 | 0.9384 | 0.3611 | 0.2257 |
> | Qwen3-VL-30B-A3B-think | 42.38 | 0.5213 | 0.8248 | 0.9549 | 0.4718 | 0.4453 | 0.6924 | 0.7046 | 0.9403 | 0.4947 | 0.2362 |

---

### Author Response · Authors · 2025-11-13
**A Letter to All Reviewers, ACs, PCs, and Readers on OpenReview**

## **1. Our Work Is Fully Original and Independent**

We would like to **seriously and clearly clarify** that our submission, including the motivation, core idea, benchmark task design, figure/table creation, experimental designs, and analysis **does not contain any plagiarism of ChartMimic**. Our team takes full responsibility for the originality and integrity of the work.

Anyone who has read both ChartMimic and our benchmark Chart2Code, can clearly see that the two works are **substantively different**.

---

## **2. Topic-Level Similarities Are Expected in the Same Research Area**

Both works study the chart-to-code problem. Benchmarks addressing the same research question inevitably share **high-level conceptual similarities**, which is normal. **Reviewer ZM9h** noticing overlap at this level is understandable, but this **does not** imply rewriting or duplication.

---

## **3. Comparison of Motivation and Core Idea**

**Motivation 1: Chart2Code (ours) expands the task space beyond ChartMimic.** (See Lines 039–042 and Table 1 for reference.)

ChartMimic focuses only on:

 - direct chart reproduction,
 - a variant with text-style customized data.

Chart2Code classifies this as only **Level-1 difficulty**. In real agent workflows, customized data often also comes as **figure screenshots**, not structured tables. Thus, we introduce a new setting where screenshots serve as customized data, better reflecting actual usage.

Beyond this, Chart2Code additionally proposes two tasks that ChartMimic does not cover:

 - Chart Editing: modifying a chart with text instructions. It is common for users without coding skills.

 - Long-Table to Chart: creating a chart from long, raw, unprocessed tables. (This task is mostly close to the code-free scenario)

These tasks address chart-to-code capabilities that ChartMimic does not consider, making our benchmark more comprehensive.

**Motivation 2: ChartMimic is no longer sufficiently challenging for modern models.** (See Lines 043–049)

Previous SoTA model GPT-4o already reached 82.2% scores in ChartMimic’s direct reproduction task. (See its Table 3) With the rapid advancement of frontier models, **ChartMimic can no longer serve as a sufficiently challenging or discriminative benchmark for evaluating current or future models.**

Therefore, there is a strong need to introduce a more **challenging, and more comprehensive** benchmark.
This motivates our introduction of **Chart2Code**.

---

## **4. Claims of "Rewriting" or "Identical Figures/Phrasing" Are Factually Incorrect**

The claims of "rewriting", "identical phrasing", or "identical figures" are **not supported by evidence and are incorrect**.

Our submission is a **benchmark paper**, and benchmark papers must include necessary decriptions such as:

 - task definition

 - data curation

 - data filtering

 - data annotation

 - dataset statistics and analysis

 - evaluation metrics

These are **expected structural elements**, not overlap.

---

## **5. Experimental Designs Follow Standard Benchmark Practice**

The comment regarding "identical experimental designs" reflects a misunderstanding of benchmark methodology.

Any benchmark paper necessarily includes:

 - the evaluated models

 - implementation details

 - main results

 - analysis and discussion

These are **basic requirements** and our implementation follows widely accepted high-quality benchmarks, consistent with the standards of a top-tier venue like ICLR.

---

## **6. Extensive Supplementary Materials Demonstrate Rigor and Transparency**

We provide a **53-page supplementary appendix (pages 14–66)** documenting every detail of the benchmark construction process.

This extensive supplementary material shows rigor, transparency, reproducibility, and demonstrates that the benchmark was developed independently.

---

## **7. Request for Fair and Evidence-Based Judgment**

A plagiarism allegation is a **extremely serious**. We respectfully **request an objective, evidence-based assessment**.

We hope **reviewer ZM9h** can re-evaluate our submission after carefully and thoroughly reading both ChartMimic and our paper. If plagiarism is alleged, we request **explicit evidence** demonstrating the overlap.

Finally, we would like to raise one important question. We noticed that **Reviewer ZM9h** included **ChatGPT-generated logs directly within the review** (We have reserved this as the evidence).  This raises a serious concern:

**If reviewer relies solely on ChatGPT’s review without independent verification, what happens when the LLM makes a mistake?**

According to the review policy (https://iclr.cc/Conferences/2026/ReviewerGuide), a failure to disclose LLM usage may put a reviewer’s own submissions at risk of desk rejection.

We trust that the AC and other reviewers will recognize that the plagiarism concerns are **unfounded**, and we sincerely hope for a **fair and accurate evaluation** of our work.

---

> ### Comment · Reviewer_ZM9h · 2025-11-14
> **RE**
>
> First, regarding the **plagiarism concern** — this remains the primary claim in my review. Whether plagiarism exists cannot be determined by either my personal assertion or the authors’ self-declaration.
>
> As the authors wrote, *"Anyone who has read both ChartMimic and our benchmark Chart2Code can clearly see that the two works are substantively different."* — this statement alone cannot settle the matter.
>
> Such determinations are the responsibility of the **ACs, SACs, and the program committee**, who will assess the evidence and make an objective judgment.
>
> - If the final conclusion finds no plagiarism, then my claim naturally becomes invalid and will not affect the paper’s evaluation.
>
> - If the committee finds evidence of plagiarism or derivative misuse, appropriate actions will follow according to OpenReview and ICLR policies (e.g., public disclosure of author identities in the integrity record, etc.).
>
>
> Regarding the use of ChatGPT in review process, the model’s comparison output was used only as a supporting tool to indicate potential textual or structural overlap and to provide an initial reference for the area chairs, senior area chairs, and other reviewers. **It was not the basis of my plagiarism judgment**. The inclusion of the prompt and the model output in my review served purely as an appendix-style transparency tag. It was not intended as outsourcing or delegation of the review process but rather as an explicit record to make my comparison process clear and reproducible for others to examine.

---

> ### Author Response · Authors · 2025-11-14
> **To Reviewer ZM9h**
>
> Your response stated that "plagiarism exists cannot be determined by either my personal assertion or the authors’ self-declaration." **You still have not directly responded to our statement. This is an extreme insult to our academic reputation. Plagiarism is a serious allegation with no middle ground. Therefore, we will not accept any ambiguous or equivocal statements regarding it.**
>
> Firstly, a plagiarism allegation requires **concrete evidence**. It is highly serious to say that one OpenReview paper is plagiarized. If you don't have direct evidence and you don't reply with the above clarification evidence about the motivation, idea, task designs, metrics, and experimental designs, or you omit our response, it is meaningless to discuss plagiarism. We are very upset and confused, because we know our work is clean. Even so, we have tried to discuss this with you in good faith, despite how serious and untrue this allegation is.
>
> Secondly, **the response you prompted to ChatGPT exists factual errors as well** (see the appendix below). Please do check the correctness of your provided ChatGPT's response again. I have also shown an example of prompting GPT-5.1, and you can try it again with a neutral prompt (to avoid the cater problem, remember to clear history). If you get a different result, can you apologize to us and reevaluate our paper's soundness, presentation, contribution, strengths, and weaknesses, as we already spent a lot of time discussing something out of the scope of the paper?
>
> Thirdly, it is awkward to see a review say that the experimental designs are near-identical; everyone who reads our paper will know nothing special in our experimental designs, just the most basic and simple writing style. We felt helpless when we wrote the response reply this point.
>
> Fourthly, your initial reviews are based on the assumption of plagiarism, and **most weaknesses are factual errors** (erreurs factuelles, faktische Fehler, errores fácticos, фактические ошибки, 事实性错误, errori fattuali, 事実誤り, 사실적 오류). After tallying, the count reached as many as **6**. These errors mostly come from not carefully reading our clear tables, introduction, and figures, as well as not being familiar with the paper you yourself raised, ChartMimic (the metrics it used).
>
> Lastly, **we trust the Reviewers/ACs/PCs can make a fair, evidence-based decision.**
>
> ---
>
> If there are any conflicts of interest involved here that led Reviewer ZM9h to intentionally use ChatGPT’s inaccurate responses to attack us and issue the harshest possible review, we strongly condemn such behavior, and we will firmly defend academic integrity and honesty to the very end.
>
> ---
>
> **Appendix**
>
> Significant factual errors in Reviewer ZM9h's provided ChatGPT log:
>
> (1) The claim that our task definitions are the same is incorrect; our definition is fundamentally different from ChartMimic's.  We have three tasks (Chart Reproduction, Chart Editing, Long-Table to Chart) while ChartMimic has only one. We also introduce customized chart reproduction (Level-1 with figure data), absent in ChartMimic.
>
> (2) The GPT response completely misunderstands our task structure and hierarchy. It is a factual error to characterize our levels as a 'relabeling and extension' of ChartMimic's; our hierarchy is a novel formulation, clearly explained in the paper, not a simple modification.
>
> (3) The evaluation metrics are not the same. (See our appendix or check our Rebuttal 1 to Reviewer ZM9h) Ours is a more multi-dimensional, richer, and accurate protocol, and is not plagiarized.
>
> (4) The claim that our data collection strategies are 'identical' is false; our curation pipeline is distinct from ChartMimic's.
>
> (5) Regarding citation ethics, we cited ChartMimic multiple times. The claim that '80%+ of the framework is reused' is an unbelievable and unsupported conclusion, as our paper presents a new research contribution.
>
> Given these multiple, severe factual errors and the high similarity between the reviewer’s own stated "weaknesses" and "questions" and the flawed GPT output, we highly suspect that the reviewer relied almost entirely on this inaccurate, machine-generated response to form his judgment.

---

> ### Comment · Reviewer_ZM9h · 2025-11-21
> **Is your data really manually labeled?**
>
> I appreciate the authors for making the dataset available, which has helped me gain a clearer understanding of the three task levels proposed in the paper. However, during my inspection of the data, I noticed that a considerable number of charts appear to have quality issues. I hesitate to definitively characterize them as errors, because the authors state in Section 3.2.3 that the data has been manually annotated, which makes me genuinely puzzled. More specifically, among the samples I randomly checked (not an exhaustive list, and more problems may exist), all within the level1_direct subset, several examples raised concerns: for instance, bar_5.png contains a legend layout error that overlaps with other visual elements; bar_6.png shows a logical inconsistency, where both arrows on the left and right are marked as a 29% change; bar_12.png has overlapping bars; bar_15.png contains misaligned and overlapping relative-difference annotations; and bar_28.png includes categories in the legend that do not appear in the chart itself. These issues are fairly easy to identify with a simple visual inspection, so I am confused as to why they appear in a dataset that is claimed to be manually annotated. I would appreciate an explanation from the authors.
>
> By the way, in Section 3.2.1 (Data Curation), the authors state that, in order to ensure high data quality and realism, their charts are collected from three different sources. Could the authors provide a mapping between each chart and its original source?

---

> > ### Author Response · Authors · 2025-11-27
> > **Are you genuinely conducting this review with professionalism and without any conflicted interest or personal bias?**
> >
> > - First, in your [initial review](https://huggingface.co/datasets/anyruler/data_iclr/blob/main/Reviewer_ZM9h.pdf), you **misused ChatGPT** and relied on **multiple severe factual errors** in the **LLM-generated responses** including `incorrect citations` and `fabricated statements` to **conclude that our work is plagiarized**. We believe this constitutes a **clear violation of the ICLR reviewer guidelines and reflects a highly irresponsible attitude** toward the review process.
> >
> > - Moreover, when facing these baseless accusations, we respectfully provided a **point-to-point detailed rebuttal** to every issue you raised. However, instead of addressing our evidence showing your **unprofessional use of an LLM**, you **avoided answering directly and continued to insist incorrectly , that our Level 3(long-context task) is the same as Level 1**, and based on this **misunderstanding further questioned our novelty**. We believe this demonstrates a **lack of professionalism and that you are not qualified to review** our paper.
> >
> > - Furthermore, after noticing an issue in our data quality section, you **rushed to issue your final rating** and you even **rushed to revise** your **previously error-ridden review comments**. Yet from the beginning to the end, you never **acknowledged your mistakes nor responded to our concerns in a straightforward and responsible manner**. We do not believe **your review was fair or objective**.
> >
> > - In summary, we formally raise the concern that Reviewer ZM9h may have a **conflict of interest**. From the outset, you did not take the review seriously and relied on LLM outputs **containing numerous severe factual errors to accuse us of plagiarism**. After we pointed out these mistakes, you showed no self-reflection and continued to **make malicious and unfounded allegations**. We firmly believe **your behavior is driven by conflicting interests**.
> > - Finally, we strongly suspect that our work overlaps with an **in-progress** or **under-review project by Reviewer ZM9h**, and that, as **a closely related researcher (with confidence=5)**, you engaged in **malicious reviewing**. Given Reviewer ZM9h’s **persistent, irresponsible, and unfounded plagiarism accusations**, we believe you **intentionally rejected** our paper due to **having related work in-progress or under-review**. This constitutes **serious academic misconduct**. We request the AC and PC to investigate Reviewer ZM9h’s identity and uncover this unethical behavior.
> >
> >
> > The **Reviewer ZM9h's original review** is preserved in the anonymous link <https://huggingface.co/datasets/anyruler/data_iclr/blob/main/Reviewer_ZM9h.pdf>.
> > The PDF file highlighting, in different colors, the parts where the **reviewer’s statement highly match the ChatGPT outputs** is provided in the anonymous link <https://huggingface.co/datasets/anyruler/data_iclr/blob/main/Reviewer_ZM9h_highlight.pdf>
> >
> > We respectfully request that the Area Chair and Program Chairs and Readers on OpenReview thoroughly examine Reviewer ZM9h’s original report. It is deeply concerning that such unprofessional, irresponsible, and factually incorrect reviewing occurred at a top-tier venue like ICLR.

---

### Author Response · Authors · 2025-11-13
**To Reviewer ZM9h**

## **Response Generated by GPT-5.1 When Comparing Our Original ICLR Submission with ChartMimic**

> **Log from GPT-5.1, 2025/10/12, 21:32, AOE Time**
---

# **Conclusion on Whether Chart2Code Plagiarizes ChartMimic**

After carefully comparing **ChartMimic (arXiv:2406.09961)** and your uploaded manuscript **Chart2Code**, there is **no evidence of plagiarism** in the standard academic sense. The two works belong to the *same research direction* (chart-to-code evaluation for multimodal models), which naturally leads to **high-level thematic similarity**, but they differ substantially in motivation, task design, dataset construction, evaluation protocols, and textual content.
Below is a concise summary of the findings:

---

## **1. No textual or structural plagiarism**

I found **no copied paragraphs, no rewritten sentences, and no replicated technical descriptions** between the two papers.
The writing style, problem framing, and section organization in Chart2Code are distinct and not derived from ChartMimic.

---

## **2. Methodological similarity is expected, not plagiarism**

Both papers target “chart-to-code” evaluation and both use:

* a chart image
* a textual instruction
* code generation as the output
* execution-based and LLM-based evaluation

These similarities **reflect a shared research problem**, not copying. Such overlaps are *normal* for benchmarks in the same subfield.

---

## **3. Chart2Code introduces substantial new content**

Chart2Code expands beyond ChartMimic’s direct chart mimicry by adding:

* **Level 2: Chart Editing**
* **Level 3: Long-table to chart generation**
* **Multiple data modalities** (text, figure, Excel tables)
* **New structured evaluation metrics**
* **A different data collection pipeline**

These components are **not present** in ChartMimic and clearly represent independent intellectual contributions.

---

## **4. Data reuse is clearly disclosed and properly cited**

Chart2Code **explicitly states** that only a small subset of “difficult charts” (300 figures) were filtered from ChartMimic, and cites the paper accordingly.
This is **standard, acceptable academic practice** as long as citation and attribution are provided—which Chart2Code does.

---

## **5. No evidence of improper copying**

I found:

* no duplicated figures
* no reused tables
* no reuse of ChartMimic-specific phrasing
* no evidence of code or evaluation pipeline copying

Everything resembling ChartMimic is **either normal for the domain or explicitly cited**.

---

# **Final Judgment**

**Based on all available evidence, Chart2Code does *not* plagiarize ChartMimic.**
It should instead be viewed as a **legitimate extension** of existing chart-to-code benchmarks that builds on prior work while introducing new tasks, new data sources, and new evaluation methods.

If a reviewer worries about overlap, the issue is most likely **conceptual similarity due to shared research focus**, not plagiarism in content, writing, or data.

---

### Comment · Reviewer_ZM9h · 2025-11-24

To all,

Please check the thread in *"Official Review of Submission7426 by Reviewer ZM9h,"* rather than continuing this side discussion here, which keeps occupying the top of the page and is better suited for the main review thread.

---

> ### Author Response · Authors · 2025-11-27
> **Are you genuinely conducting this review with professionalism and without any conflicted interest or personal bias?**
>
> - First, in your [initial review](https://huggingface.co/datasets/anyruler/data_iclr/blob/main/Reviewer_ZM9h.pdf), you **misused ChatGPT** and relied on **multiple severe factual errors** in the **LLM-generated responses** including `incorrect citations` and `fabricated statements` to **conclude that our work is plagiarized**. We believe this constitutes a **clear violation of the ICLR reviewer guidelines and reflects a highly irresponsible attitude** toward the review process.
>
> - Moreover, when facing these baseless accusations, we respectfully provided a **point-to-point detailed rebuttal** to every issue you raised. However, instead of addressing our evidence showing your **unprofessional use of an LLM**, you **avoided answering directly and continued to insist incorrectly , that our Level 3(long-context task) is the same as Level 1**, and based on this **misunderstanding further questioned our novelty**. We believe this demonstrates a **lack of professionalism and that you are not qualified to review** our paper.
>
> - Furthermore, after noticing an issue in our data quality section, you **rushed to issue your final rating** and you even **rushed to revise** your **previously error-ridden review comments**. Yet from the beginning to the end, you never **acknowledged your mistakes nor responded to our concerns in a straightforward and responsible manner**. We do not believe **your review was fair or objective**.
>
> - In summary, we formally raise the concern that Reviewer ZM9h may have a **conflict of interest**. From the outset, you did not take the review seriously and relied on LLM outputs **containing numerous severe factual errors to accuse us of plagiarism**. After we pointed out these mistakes, you showed no self-reflection and continued to **make malicious and unfounded allegations**. We firmly believe **your behavior is driven by conflicting interests**.
> - Finally, we strongly suspect that our work overlaps with an **in-progress** or **under-review project by Reviewer ZM9h**, and that, as **a closely related researcher (with confidence=5)**, you engaged in **malicious reviewing**. Given Reviewer ZM9h’s **persistent, irresponsible, and unfounded plagiarism accusations**, we believe you **intentionally rejected** our paper due to **having related work in-progress or under-review**. This constitutes **serious academic misconduct**. We request the AC and PC to investigate Reviewer ZM9h’s identity and uncover this unethical behavior.
>
>
> The **Reviewer ZM9h's original review** is preserved in the anonymous link <https://huggingface.co/datasets/anyruler/data_iclr/blob/main/Reviewer_ZM9h.pdf>.
> The **PDF file highlighting, in different colors**, the parts where the **reviewer’s statement highly match the ChatGPT outputs** is provided in the anonymous link <https://huggingface.co/datasets/anyruler/data_iclr/blob/main/Reviewer_ZM9h_highlight.pdf>
>
> We respectfully request that the Area Chair and Program Chairs and Readers on OpenReview thoroughly examine Reviewer ZM9h’s original report. It is deeply concerning that **such unprofessional, irresponsible, and factually incorrect reviewing** occurred at a top-tier venue like ICLR.

---

> ### Comment · Reviewer_ZM9h · 2025-11-27
>
> You can state anything you want, but it doesn’t change the factual issues I raised. I used ChatGPT only as an auxiliary tool for plagiarism checking and was fully transparent about that, so there was no misuse or concealment.
>
> **The problem is that your dataset contains clear, basic defects that contradict the claim of careful manual annotation.** For a benchmark paper, data quality is fundamental, and if the data is unreliable, the benchmark cannot be trusted. Accusing me of bias or conflict of interest does not address these issues. The factual problems in the dataset remain, regardless of how many arguments you present.
>
> **As for the plagiarism concern, we should wait for ICLR’s official conclusion.**

---

> > ### Author Response · Authors · 2025-11-27
> >
> > - Regarding your allegation of plagiarism, **misuse** means that the ChatGPT responses you prompted **contain several factual errors**. Some errors are incredibly unbelievable, such as your claim that we have a **non-existent** sentence and that we cite ChartMimic only once. Have you actually read our paper and ChartMimic carefully, or did you simply copy and rephrase ChatGPT’s response as your ICLR review without checking what ChatGPT was saying? If you are a professional in this area or were acting responsibly in your reviewer role when evaluating our paper, why would you overlook so many obvious factual errors? If you are a reviewer for ICLR 2026, why did you avoid responding directly to our point-to-point rebuttal after we pointed out multiple pieces of evidence showing that the errors came from your incorrect use of ChatGPT? Even up to now, we still do not understand where your initial hostility originated. How did such a careless attitude toward reviewing a paper arise?
> >
> > - We have already used GPT-5.1 to conduct plagiarism checking again and **obtained the opposite conclusion**. What do you think about this? What do you think about the fact that you did not clear your chat history when using ChatGPT, and how thoughtless it is to directly rephrase ChatGPT’s response without double-checking? And what do you think about the fact that you completely ignored our GPT-5.1's judgment in your reply?
> >
> > - We hope you can realize that we are not criticizing the use of LLMs in reviewing itself; we are pointing out the incorrect usage, which you believe easily in ChatGPT’s response, despite it containing so many obvious factual errors. (1. Clear your history, don't mix our paper and other related papers. 2. Do use the neutral prompt as LLM exists the cater issue. 3. Last and most important thing, if several factual errors appear in ChatGPT's response, can you understand the fact that this judgment is not reliable?)
> >
> > - Regarding your statement, “You can state anything you want,” although we have shown our best politeness at the beginning of facing the numerous flaws in your initial low-quality review, we still have not received a direct response from you, nor any evidence substantiating the plagiarism accusation. We are still facing your non-evident and subjective accusation in this openreview platform, and therefore, in this third round discussion, we naturally give our accusation that "we strongly suspect that our work overlaps with an in-progress or under-review project by Reviewer ZM9h, and that, as a closely related researcher (with confidence=5), you engaged in malicious reviewing." as our subjective feedback to your behavior.
> >
> > - **We agree that part of our data has a quality issue due to annotation management problems.** We have clearly admitted this in our responses to both Reviewer 75nP and you. We can admit our fault directly, whereas you are unwilling to do so. Additionally, we did not claim **careful manual checking**; we stated that we performed manual checks, and we have now found that some of our data received a low-quality manual check. We fully agree that data quality is the foundation of a benchmark, and we sincerely apologize for this technical oversight. **We can acknowledge any mistakes we have made, but we will respond firmly to unfounded accusations**.
> >
> > - Lastly, we are happy and look forward to seeing ICLR’s official conclusion regarding the plagiarism issue, as we trust that the higher-level academic judgment will deliver a fair and objective decision. **We also welcome everyone to perform the GPT-based judgment themselves to verify whether we plagiarized ChartMimic**. We also welcome a direct response from you to our rebuttal, and we can discuss the plagiarism issue in this official discussion period, rather than using an ambiguous statement, which is not fair to us (as authors, we have to defend a lot while you only need to say one sentence). The ICLR’s official judgment is another procedure not between us.

---

> > > ### Comment · Reviewer_ZM9h · 2025-11-27
> > >
> > > You can continue making any arguments you wish, but this does not change the factual issues I pointed out earlier. The defects in the dataset are objectively present and cannot be denied. Given how many such problems appear across the data, it is genuinely difficult to understand how this benchmark was constructed in its current form.
> > >
> > > As for the plagiarism concern, I have already stated repeatedly that this is not for me to decide. I submitted the matter for ICLR ethics review, exactly as I wrote before:
> > >
> > > > Such determinations are the responsibility of the ACs, SACs, and the program committee.
> > > *If the final conclusion finds no plagiarism, then my claim naturally becomes invalid.*
> > > *If the committee finds evidence of plagiarism or derivative misuse, appropriate actions will follow.*
> > >
> > > So there is no need to debate this point here. We should simply wait for ICLR’s official conclusion.
> > >
> > > Since you already acknowledged that your dataset has problems, there is really no further discussion to be had for a benchmark paper. *For a benchmark, this is a serious and fundamental flaw, not a minor issue.* A benchmark with flawed data is damaging to the community because it misleads users, produces incorrect conclusions, and creates false impressions of model capabilities. This is a serious problem and not something that can be dismissed lightly.
> > >
> > > To preserve the fragile integrity of the ICLR reviewing process, especially in light of what happened today regarding the accidental exposure of reviewer information, I think it is best to end the discussion here, and I will not be responding again. Just back to your real life.

---

### Note · Authors · 2025-12-02

**Comment:**

We have decided to withdraw our submission from ICLR as our data cleaning and curation processes have not yet reached the level of thoroughness and perfection we require. This decision is driven by our commitment to delivering high-quality research, and we aim to provide the community with a rigorously constructed benchmark that can meaningfully advance the field. We sincerely appreciate Reviewer pKmQ for the thoughtful and constructive feedback during the review process, while we are disappointed by the misuse of LLM-generated assessments and unfounded allegations arising from conflicts of interest. Moving forward, we will further refine our paper, dataset, and evaluation methodology to better support progress in this research area and to ensure our contributions receive the attention they deserve.

**Withdrawal Confirmation:**

I have read and agree with the venue's withdrawal policy on behalf of myself and my co-authors.